# Optineurin links Hace1-dependent Rac ubiquitylation to integrin-mediated mechanotransduction to control bacterial invasion and cell division

Serena Petracchini[1,16], Daniel Hamaoui[2,3,16], Anne Doye[2,3], Atef Asnacios [4], Florian Fage[4], Elisa Vitiello[5], Martial Balland [5], Sebastien Janel [6], Frank Lafont[6], Mukund Gupta[7], Benoit Ladoux [7], Jerôme Gilleron[8], Teresa M. Maia [9,10,11], Francis Impens [9,10,11], Laurent Gagnoux-Palacios[12], Mads Daugaard [13,14], Poul H. Sorensen[15], Emmanuel Lemichez[1,2,3,17] ✉ & Amel Mettouchi [1,2,3,17] ✉

Extracellular matrix (ECM) elasticity is perceived by cells via focal adhesion structures, which transduce mechanical cues into chemical signalling to conform cell behavior. Although the contribution of ECM compliance to the control of cell migration or division is extensively studied, little is reported regarding infectious processes. We study this phenomenon with the extraintestinal *Escherichia coli* pathogen UTI89. We show that UTI89 takes advantage, via its CNF1 toxin, of integrin mechanoactivation to trigger its invasion into cells. We identify the HACE1 E3 ligase-interacting protein Optineurin (OPTN) as a protein regulated by ECM stiffness. Functional analysis establishes a role of OPTN in bacterial invasion and integrin mechanical coupling and for stimulation of HACE1 E3 ligase activity towards the Rac1 GTPase. Consistent with a role of OPTN in cell mechanics, OPTN knockdown cells display defective integrin-mediated traction force buildup, associated with limited cellular invasion by UTI89. Nevertheless, OPTN knockdown cells display strong mechanochemical adhesion signalling, enhanced Rac1 activation and increased cyclin D1 translation, together with enhanced cell proliferation independent of ECM stiffness. Together, our data ascribe a new function to OPTN in mechanobiology.

Many bacterial pathogens use cell invasion as a critical step for host colonization and further dissemination or for persistence. Uropathogenic strains of *Escherichia coli* (UPEC) are a leading cause of urinary tract infections (UTIs) and bacteraemia, and are also frequently responsible for meningitis in neonates[1,2]. Expression of chaperone-usher pathway type I pili tipped with the adhesin FimH is important for colonization, invasion and persistence of UPEC in the mouse bladder and to form persistent reservoirs in the gastrointestinal tract[2,3].

Mechanistically, UPEC utilizes FimH to bind to glycosylated integrin receptors and get internalized into host cells[4]. This invasion process also involves Rho GTPase members, notably Rac1, to drive actin cytoskeleton rearrangement, leading to the zippering of the plasma membrane around bacteria and engulfment[2,5,6]. Type 1 pili are sophisticated mechanoresponsive attachment appendages that allow bacteria to resist urinary flow and stay attached to tissues due to the reinforcement of their surface binding via a catch-bond mechanism of

---

the FimH adhesin and the elastic spring structure of the FimA pilus rod[7–10]. UPEC encounter different niches associated with pathology, going from the bladder and kidney or even reaching the bloodstream and endothelia[2]. These different niches encountered by UPEC in the body harbor extremely different tissue mechanical properties, with, for example, an average elastic modulus of the gastrointestinal tissues three-fold higher than that of bladder tissue and 100-fold higher than that of kidney[11–13]. How tissue mechanics influence the outcome of UPEC infection is a poorly explored question. Numerous factors from bacterial pathogens exploit eukaryotic Rho GTPase signaling to invade and proliferate within their hosts[14,15]. The highly prevalent cytotoxic necrotizing factor 1 (CNF1) toxin from UPEC represents an example of Rho GTPase activating factor that triggers actin-based membrane deformations for host cell invasion[6,16,17]. Whether and how FimH and CNF1 work in a concerted or redundant fashion to enhance host cell invasion by UPEC remains to be defined.

Rac1 is a classic switch GTPase that undergoes a guanine nucleotide-based spatiotemporal cycle. Rac1 alternates between a GDP-bound "OFF" state and a GTP-bound "ON" state that interacts with effectors to trigger downstream signaling[18,19]. Emerging studies have identified the crucial role of a degradative pathway via ubiquitin and proteasome systems (UPS) in the control of Rac1 activity[16,20–22]. This pathway involves the HECT domain and ankyrin repeats containing E3 ubiquitin ligase (HACE1), which associates selectively with active Rac1 to catalyze its polyubiquitylation and UPS-mediated degradation[20,22]. Control of Rac1 activity via the UPS is crucial during epithelial infection by CNF1-producing UPEC and for balanced reactive oxygen species production by Rac1-dependent NADPH oxidases[23]. The discovery that Rac1 activity is influenced by the stiffness of the extracellular matrix (ECM) or mechanical tension and stretch[24–26] has raised major questions concerning signaling pathways connecting Rac1 to mechanotransduction. While the application of force has been shown to activate GEFH1 and LARG, two RhoA GEFs[27,28], no such situation has been described thus far for the Rac1 GTPase. Rac1 clearly regulates cell proliferative behavior linked to ECM compliance[29], and while downstream signaling has been well characterized, how Rac1 senses and adapts its level of activity to ECM compliance remains to be elucidated.

One major cellular component in cell mechanosensing and adaptation is the actin cytoskeleton, as it is the source of internally generated force[30]. This force is further transmitted through adhesion receptors to generate traction on the ECM, and reciprocally, adhesion receptors will strengthen their binding to the ECM and reinforce the composition of the adhesion structures to resist forces[31]. Rho family GTPases were established decades ago as critical regulators of actin cytoskeleton dynamics and fate[32]. Indeed, RhoA promotes the polymerization of actin into linear filaments and stimulates the activation of Myosin II to assemble contractile actomyosin filaments[33]. Rac1 promotes the assembly of a cortical actin meshwork, leading to membrane protrusions via WAVE-driven activation of the actin polymerizing complex Arp2/3[34]. Focal adhesion (FA) formation and actin cytoskeleton polymerization and remodeling are tightly coupled. Indeed, by activating dendritic actin polymerization, Rac1 triggers the recruitment of integrins and actin-binding proteins into nascent adhesions that are formed in the lamellipodium zone. Hence, Arp2/3 shows affinity for FAK and Talin[35]. The protrusive force generated by actin polymerization on the membrane creates an actin retrograde flow in the cell body-proximal part of the protrusion, the lamella zone, associated with actin depolymerization and nascent adhesion dispersal. In the presence of myosin- and alpha-actinin-mediated actin crosslinking, a fraction of nascent adhesions can be strengthened at the lamellipodia-lamella boundary[36–38] to generate focal complexes and then mature further into FAs with the generation of RhoA-dependent actomyosin contractile stress fibers. FAs will experience tension-induced changes in their composition and signaling activity[31,39–41]. In this way, mechanical cues originating from the ECM

are converted into biochemical signaling, which will control cellular responses such as migration or proliferation. Thus, Rac1 influences the mechanical response/adaptation of cells via the aforementioned functions. It is now important to define how to position Rac-mediated control of cell functions in the context of mechanosensing, given that ECM stiffening and abnormal responses to mechanical cues of the ECM are associated with a large spectrum of human diseases, including cancer and chronic inflammation[42–46], and that Rac1 plays a critical role in the control of cell-ECM adhesion, transcriptional responses, and cell division[47–49] and is frequently highjacked by invasive pathogens during infectious processes[50–54]. In this work we show that ECM mechanical properties impact Rac1-GTP ubiquitylation and thereby tune Rac1 level of activity in cells. An increase of ECM stiffness leads to elevation in the protein level of Optineurin, a HACE1 partner, which then stimulates Hace1-dependent ubiquitylation of Rac1. Consequently, The knockdown of OPTN leads to enhanced Rac1-dependent cellular responses as increased integrin adhesion, cyclin D1 translation and cell division. Studying *E. coli* UTI89 invasive pathogen, we show that ECM mechanical properties modulate CNF1 toxin-dependent invasion of cells by the pathogen in cooperation with the integrin-receptor FimH adhesin of type 1 pili. This requires integrin mechanoactivation and Rac1 activity. In this model of host cell invasion, we reveal a function of OPTN in the control of bacteria internalization via modulation of Rac1 activity and a mechanical coupling of stimulated integrins to the actomyosin network to generate traction forces. Together, our data ascribe a new function to OPTN in mechanobiology.

## Results
### Extracellular matrix compliance tunes cellular invasion by CNF1-producing uropathogenic *E. coli* in a FimH–integrin mechanoactivation-dependent mechanism

UPEC are a good model to assess the contribution of changes in the mechanical properties of tissues in infection, considering that they use both integrins as receptors and toxins to manipulate cytoskeletal dynamics via host small GTPases. To assess the effect of ECM elasticity on the infection process, we cultured primary HUVEC cells on fibronectin-coated hydrogels with different elastic moduli. We monitored UPEC adhesion to cells (after 30 min of contact) as well as the efficiency of cell invasion at early time points of infection (another 30 min after bacteria contact with cells) using the gentamicin protection assay. For this purpose, we used a UTI89-derived strain deleted from the *cnf1 toxin*-encoding gene (hereafter referred to as "Ec", see materials) that we complemented when indicated with a defined concentration of recombinant CNF1 toxin. This approach allows to quantitatively address key initial events. At a constant cell density and MOI, we measured a slight monotonic stiffness-dependent increase in bacterial invasion (Fig. 1a), which was further increased by the presence of CNF1 toxin to reach the highest value for cells plated on 50 kPa ECM. Such an observation indicates that efficient bacterial invasion relies on a toxin function that is limited by a cellular event or factor controlled by ECM stiffening. ECM compliance had no impact on bacterial adhesion to the host cell surface (Supplementary Fig. 1A). We verified that CNF1 activates Rac1 equally well in cells cultured on ECM of different stiffnesses (Supplementary Fig. 1B). We also confirmed that adhesion of Ec was similar in the presence or absence of CNF1 (Supplementary Fig. 1A), ruling out an effect of the toxin in the bacterial adhesion step, as expected[6]. Deletion of the FimH adhesin, which tips type 1 fimbriae and binds to mannosylated beta1 integrins[4], abrogated the ECM stiffness-dependent invasion of bacteria (Fig. 1a). In this context, exogenous complementation by CNF1 only led to a marginal gain in invasion, which was similar for different ECM stiffnesses. We reasoned that CNF1 might cooperate with FimH to allow integrin mechanical coupling to the cytoskeleton and efficient active bacterial invasion. Integrins belong to a key family of mechanoreceptors that convert mechanical cues into chemical signaling. Key aspects of

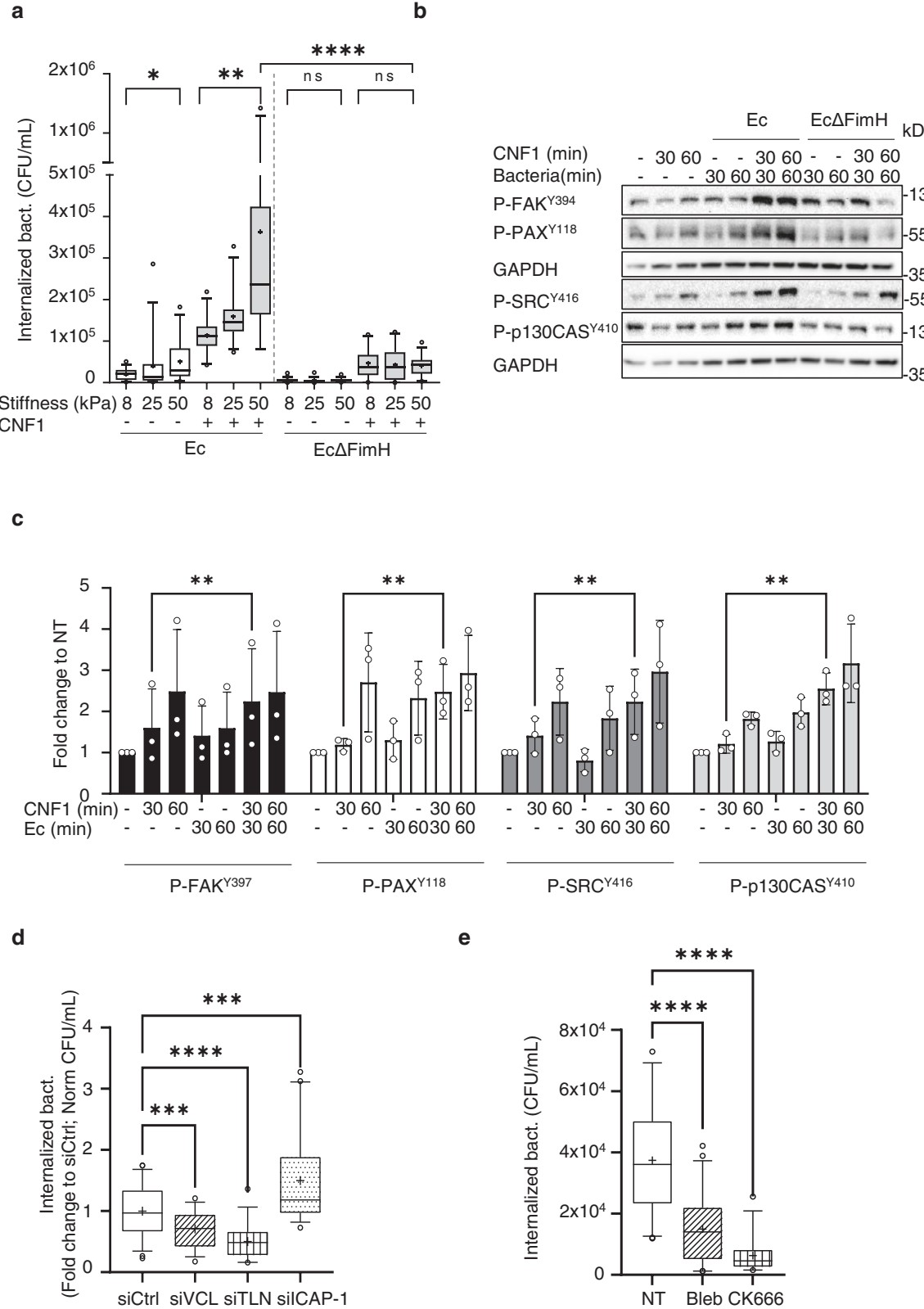

tensional sensing and tension-induced signaling within FAs involve the FAK-Src tyrosine-kinase module, which phosphorylates substrates in the FA upon stretch-dependent unfolding and unmasking of sites[40,41,55,56]. We analyzed the activity of these kinases in toxin-treated and infected cells by monitoring their autophosphorylation and the phosphorylation of paxillin and p130CAS as critical targets. Consistent with our hypothesis, we found that CNF1 activates integrin

mechanosignalling, as measured by the phosphorylation levels of FAK-Y[397], Src-Y[416], Paxillin-Y[118] and p130-CAS-Y[410]. CNF1 activates integrin mechanosignalling at 30 min, with a maximum reached at 60 min of the assay (Fig. 1b, c and Supplementary Fig. 1C). This signaling was accelerated or reinforced to reach the maximal value as early as 30 min in the presence of FimH-expressing bacteria (Fig. 1b, c), while bacteria themselves were less efficient at triggering integrin mechanosignalling

**Fig. 1 | ECM stiffness and integrin mechanoactivation control cnf1-producing uropathogenic *E. coli* invasion. a** Internalized Ec or FimH-deleted Ec (EcΔFimH) 30 min after gentamicin treatment in HUVEC cultured on ECM of increasing stiffness (8, 25, 50 kPa) in complete medium. CNF1 was added at 1 nM where indicated (+). An MOI of 100 was used. Boxplots displays absolute number of CFU/ml, $n = 3$ independent experiments and at least three replicates per condition. One-way ANOVA with Dunnett's correction test for multiple comparisons: ns, $p > 0.999$ not significant; *$p = 0.0147$; **$p = 0.0028$; ****$p \leq 0.0001$. **b** Representative immunoblots of phospho-protein levels for phospho-FAK(tyr-397), phospho-Paxillin(tyr-118), phospho-Src (tyr-416) and phospho-p130CAS(tyr-410) in response to HUVEC infection for 30 or 60 min by Ec or EcΔFimH (MOI 100), in presence of 1 nM CNF1 toxin when indicated. GAPDH was used as loading control. **c** Quantification of phospho-protein levels for phospho-FAK(tyr-397), phospho-Paxillin(tyr-118), phospho-Src(tyr-416) and phospho-p130CAS(tyr-410) relative to levels of GAPDH and normalized to the non-treated condition. Bars represent means ± SD, n3

independent experiments (single dots). Two-way ANOVA with Tukey's correction test for multiple comparisons: **$p = 0.0016$. Expression of FAK, PAX, Src and p130CAS are not modified in our experimental conditions. **d** Quantification of internalized Ec in HUVECs 30 min after gentamicin treatment in HUVECs knocked down for Vinculin(siVCL), Talin(siTLN) or Icap-1(siICAP), in presence of 1 nM CNF1. Boxplots display the quantification of CFU/ml relative to the control condition (siCTRL) from $n = 3$(siVCL, siTLN) and $n = 5$(siICAP) independent experiments and three replicates per condition. One-way ANOVA with Dunnett's correction test for multiple comparisons: ***, siCtrl-siICAP1 $p = 0.0005$, siCtrl-siVCL $p = 0.0007$; ****, siCtrl-siTLN $p \leq 0.0001$. **e** Quantification of internalized Ec in HUVECs either non-treated or treated with blebbistatin (20 μM) or CK666 (250 μM) 30 min after gentamicin treatment. Boxplots displays absolute number of CFU/ml, $n = 4$ independent experiment and three replicates per condition. One-way ANOVA with Dunnett's correction test for multiple comparisons: ****$p \leq 0.0001$.

at this time point. The marked decrease of FAK and Src signaling upon infection with FimH-deleted bacteria showed that the presence of FimH-tipped pili at the surface of bacteria fostered integrin mechanosignalling activation in the presence of CNF1 specifically (Ec Δ FimH, Fig. 1b right lanes and Supplementary Fig. 1D). These data establish the underrated role of CNF1 in FimH-dependent integrin functional engagement and in the ensuing downstream signaling for efficient bacterial internalization. As expected, knocking down beta1 integrins in cells led to impairment of bacterial internalization (Supplementary Figs. 1E and 4) and loss of mechanosignalling (Supplementary Fig. 1F).

To further confirm that integrin mechanoactivation is required for UPEC invasion induced by CNF1, we modulated intrinsic regulators of integrin mechanics (Fig. 1d). Knockdown of either Vinculin or Talin, which are necessary to connect integrins to the actin cytoskeleton and stimulate integrin conformational activation to transmit force[39,57], lead to reduced bacterial invasion (Fig. 1d and Supplementary Fig. 1G). Interestingly, the knockdown of ICAP-1, a negative regulator of integrins that competes with Talin for beta1 cytoplasmic tail binding, led to an improvement in bacterial internalization in response to CNF1 (Fig. 1d). The knockdown of ICAP-1 has been shown to increase integrin activation and leads to enhanced beta1 integrin-dependent contractile forces in these cells[58]. Therefore, we reasoned that CNF1 and FimH work in concert to trigger higher mechanoactivation of integrins in the context of ICAP-1 loss, an event preempted during CNF1-mediated bacterial invasion. Consistent with this finding, treatment of cells with blebbistatin, an inhibitor of actomyosin contractility, or with CK-666, an inhibitor of Arp2/3-mediated actin polymerization downstream of Rac1, impaired bacterial invasion into cells (Fig. 1e) while having no impact on bacterial attachment to cells (Supplementary Fig. 1I) nor bacterial growth (Supplementary Fig. 1J). The knockdown of Talin, Vinculin or ICAP-1 had no effect on bacterial attachment to cells (Supplementary Fig. 1H), consistent with our finding that ECM stiffness had no impact on bacterial attachment (Supplementary Fig. 1A).

Altogether, our data demonstrate that ECM elasticity controls host cell susceptibility to *E. coli* invasion by influencing binomial FimH and CNF1-dependent processes.

## Proteomics analysis of the cellular adaptation to ECM stiffness and CNF1 toxin activity

To understand how host cells adapt their proteome to ECM stiffness to eventually control bacterial invasion, we performed label-free, quantitative proteomics analysis of HUVECs cultured on fibronectin ECM of different stiffnesses (low 1 kPa, 4 kPa, high 50 kPa). To study the impact of CNF1 toxin on this cellular response, we cultured cells overnight and then assigned them to either the untreated (NT) or CNF1-treated group (CNF1) upon treatment for an additional 2 h with the toxin (Supplementary Fig. 2A). Protein lysates from four replicates per condition were processed for liquid chromatography-tandem mass spectrometry (LC-MS/MS) analysis. A total of 3515 protein groups were identified, and

1804 were reliably quantified (protein groups with at least three valid LFQ intensity values in one of the experimental conditions) (listed in Supplementary Data 1 and 2). A principal component analysis was performed on the replicate samples using all quantified proteins as variables (Supplementary Fig. 2B). PC1 on the *x*-axis explains 62.7% of the data variance, indicating good segregation of the samples regarding experimental conditions, except with one replicate of two independent conditions that were nevertheless kept in the analysis. Differences in protein abundance between the groups were visualized in a heatmap after non-supervised hierarchical clustering of *z*-scored protein LFQ intensities (Fig. 2a and Supplementary Data 3). Global analysis of quantitative changes in the proteome (Supplementary Fig. 2C) showed dominant trends toward increases in protein expression as a function of increases in ECM stiffness (Fig. 2a and Supplementary Fig. 2C), with 652 and 771 proteins showing upregulated expression while 79 and 78 proteins showed downregulated expression in the NT and CNF1 groups, respectively. We examined the distribution of differentially expressed proteins in each comparison and found that the quantitative impact of CNF1 was the most obvious in the 4 kPa vs. 1 kPa upregulated groups, with differences being tempered in other groups (Supplementary Fig. 2C). The heatmap shows a gradual response of cells to ECM stiffness and clustering in two groups: proteins highly expressed in cells cultured on low stiffness ECM and whose expression decreases monotonically (cluster 1) and proteins with low expression in cells cultured on low stiffness ECM and whose expression increases monotonically with ECM rigidity, the major cluster (cluster 2) (Fig. 2a). We thus performed a functional enrichment analysis using DAVID on proteins with significantly upregulated expression in any of the pairwise comparisons (50 vs. 1, 4 vs. 1 or 50 vs. 4) in either the non-treated or CNF1-treated group (Fig. 2b and Supplementary Data 4). We found common terms shared between the NT- and CNF1-treated conditions (keywords: SH3 domain, cell adhesion, metal-binding, LIM domain, zinc), while the CNF1 analysis also identified terms associated with metabolism (aminoacyl-tRNA biosynthesis, fatty-acid elongation) and innate immune responses (natural killer cell-mediated cytotoxicity).

The CNF1 toxin induces Rac1 permanent activation, an event sensed by the HACE1 E3 ligase that polyubiquitylates the active form of Rac1 as a function of its strength of activation for subsequent proteasomal degradation[16,22]. In the search for potential partners of the Rac1-HACE1 system regulated by ECM stiffness, we used Cytoscape to build the interaction network corresponding to the common enhanced keywords of NT (yellow nodes) and CNF1 (yellow nodes with orange edges) conditions in our enrichment analysis, which we further merged with the HACE1 network (light-blue nodes) retrieved from the IntAct database (Fig. 2c). Interestingly, we found optineurin (OPTN), a previously described HACE1-interacting partner[59] belonging to metal-binding and zinc upregulated keywords. OPTN also has connections to the cell adhesion term via an interaction with Vinculin as determined from a proteome-scale two-hybrid screen to search for proteins

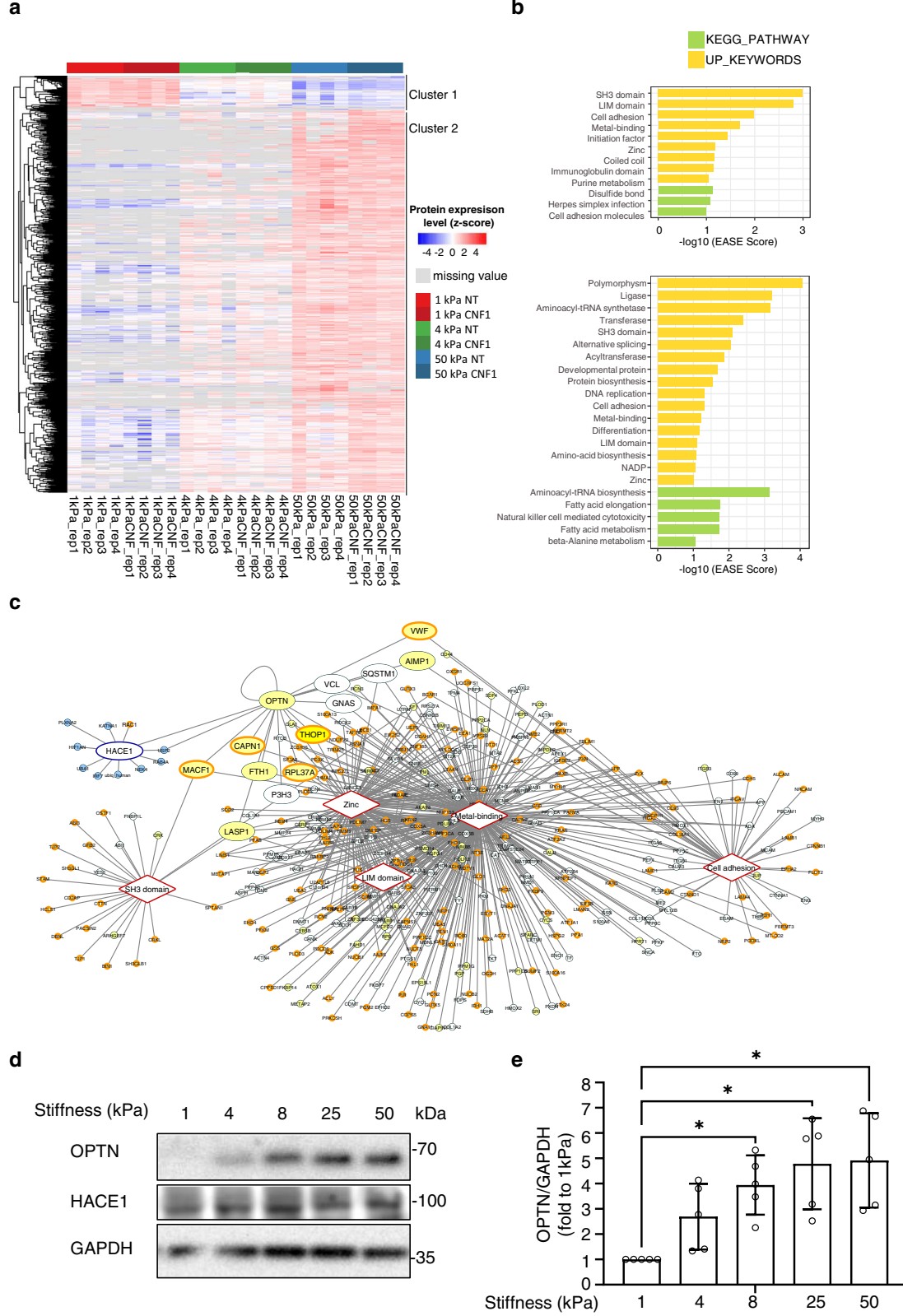

involved in muscular dystrophies[60]. Moreover, OPTN localization to FAs was described in podocytes[61]. OPTN belongs to the group of proteins with differentially upregulated expression between 1 and 50 kPa in our dataset (Supplementary Data 3, cluster2). Therefore, OPTN represents a good candidate.

We verified the expression of OPTN as a function of ECM compliance by western blots. For this, fibronectin was coated on hydrogels of increasing stiffness, spanning elastic modulus values from 1 to 50 kPa, and OPTN levels were monitored by western blots following 15 h of adhesion (Fig. 2d, e). The OPTN levels increased gradually with the ECM stiffness. In contrast, the expression of HACE1 was constant at the different ECM compliances (Fig. 2d and Supplementary Fig. 2D). Given that OPTN expression can be transcriptionally induced by TNF-α and IFN-γ[62], we assessed whether stiffness-mediated induction of

**Fig. 2 | Proteome analysis of extracellular matrix stiffness-induced changes unveils regulation of OPTN expression. a** Heatmap plot of the Z-scored LFQ intensities from significantly regulated proteins after non-supervised hierarchical clustering. The list of proteins can be found in Supplementary Data 3. **b** Graphical representation of the functional enrichment analysis performed using DAVID for the list of significantly upregulated proteins. A 0.1 EASE score cut-off allowed to gather a group of enriched UP_KEYWORDS (in yellow) and KEGG pathways (in green). Results from non-CNF1-treated cells (NT) and CNF1 toxin-treated cells (CNF1) are analyzed separately. **c** Network of association between the HACE1 protein interactome and the five sets of proteins belonging to common enriched UP_Keywords retrieved from the non-CNF1-treated cells (NT) and CNF1 toxin-treated cells (CNF1) analysis. Rhomboid boxes represent common UP_Keywords. White nodes in the network represent proteins detected in the dataset that are not significantly upregulated. Yellow nodes correspond to upregulated proteins in any of the pairwise NT comparisons (50 vs. 1, 4 vs. 1 or 50 vs. 4), a dark yellow edge of the node represents upregulation also for the CNF1 samples in any of the pairwise comparisons. Blue nodes correspond to the Hace1 interactome. Larger protein nodes are those that belong both to the protein-protein interaction network and to the UP_Keywords network. **d** Representative immunoblots anti-OPTN and anti-HACE1 showing endogenous protein levels in HUVECs plated 15 h on fibronectin-coated hydrogels of the indicated stiffness (kPa) in complete medium. Immunoblot anti-GAPDH shows control of protein loading. **e** Quantification of Optineurin (OPTN) protein level from western blot analysis by densitometry, cells treated as in **d**. The OPTN signal was normalized to that of GAPDH expressed as fold relative to the signal on 1 kPa ECM, set to 1. Bars represent means ± SD, $n = 5$ independent experiments (single dots). One-way ANOVA with Dunnett's correction test for multiple comparisons: *, 1 vs. 4 $p = 0.0137$; 1 vs. 8 $p = 0.0255$; 1 vs. 25 $p = 0.0255$; 1 vs. 50 $p = 0.0255$.

OPTN is through a transcriptional mechanism. The OPTN mRNA levels were constant in cells cultured on substrates of increasing stiffness (Supplementary Fig. 2E). These data imply that ECM stiffness controls OPTN abundance at the protein level.

## OPTN associates with the HACE1 E3 ubiquitin ligase and adapts its ligase activity toward Rac1-GTP to the level of ECM compliance

We monitored the interaction of HACE1 with OPTN by coimmuno-precipitation. Immunoprecipitation of N-terminal Flag-tagged OPTN followed by anti-HA immunoblotting established the association of HACE1 with OPTN, thereby confirming the association measured by Liu and collaborators in primary cells[59] (Fig. 3a).

We conducted ubiquitylation analysis of OPTN to determine whether OPTN is a target of HACE1 E3 ubiquitin ligase activity. Although OPTN presented a ubiquitylation profile, the latter was not induced by the presence of HACE1 or the Rac1Q61L active mutant (Supplementary Fig. 3A). This finding indicates that OPTN is not ubiquitylated by HACE1 in our experimental setting. We then tested whether OPTN regulates HACE1 E3 ubiquitin ligase activity. To this end, we measured the ubiquitylation of active Rac1Q61L in control or OPTN knockdown cells (Fig. 3b). We found that OPTN knockdown led to a reduction in Rac1Q61L ubiquitylation under conditions of endogenous or overexpressed HACE1 (Fig. 3b). Remarkably, this set of experiments establishes a novel regulatory function of OPTN in the control of Rac1 ubiquitylation. This finding prompted us to monitor whether OPTN is involved in the tuning of the E3 ubiquitin ligase activity of HACE1 as a function of ECM stiffness. We conducted ubiquitylation assays in cells expressing active Rac1Q61L that were cultured on fibronectin ECM of different compliances (Fig. 3c). Remarkably, this experiment revealed an increase in the ubiquitylation of Rac1Q61L as a function of ECM stiffening, suggesting OPTN-dependent control of Rac1 ubiquitylation driven by ECM compliance. We then challenged our hypothesis by overexpressing OPTN in cells plated on low stiffness ECM (1 kPa), where endogenous OPTN expression is low (Fig. 2d), and detected an OPTN-induced increase in Rac1 polyubiquitylation (Fig. 3d). Altogether, these data showed that Rac1 ubiquitylation by HACE1 is tuned by ECM stiffness via modulation of OPTN levels.

We next assessed the importance of OPTN in the ability of HACE1 to interact with its target Rac1. As shown in Fig. 3e, when OPTN was knocked down, HACE1 was impaired in its ability to bind to active Rac1Q61L. Together, these data showed that OPTN is required for the correct binding of HACE1 to its target Rac1 for ubiquitylation. We then analyzed the impact of OPTN on Rac1 activity in cells. We used a GST-PAK-RBD affinity column with lysates from the control or OPTN knockdown cells cultured on fibronectin-coated surfaces (Fig. 3f and Supplementary Fig. 3B). These experiments revealed an increase in endogenous Rac1 activity in the OPTN-depleted cells compared to the control cells. This result is in accordance with our ubiquitylation experiments showing less ubiquitylation of active Rac1 when OPTN

was depleted in cells. RhoA-ROCK activity is not significantly impacted in OPTN knockdown cells cultured on fibronectin-coated surfaces, as assessed by western blot measure of the phosphorylation of myosin light chain (Supplementary Fig. 3C). Integrin-mediated activation of Rac1 by fibronectin is associated with enhanced anchoring of the GTPase in detergent-resistant domains of the plasma membrane[63–65]. Therefore, we analyzed Rac1 distribution in isopycnic sucrose gradients. Rac1 partitioning into Transferrin receptor-positive high-density membrane fractions shows no difference between control and OPTN knockdown cells. By contrast in the OPTN knockdown cells, we found more Rac1 that segregated in flotillin-enriched fractions, i.e. low-density detergent-resistant membrane domains, compared to that of the control cells (Fig. 3g and Supplementary Fig. 3D). This finding confirms our measure of increased Rac-GTP levels on OPTN knockdown cells and indicates a critical role of OPTN in controlling Rac1 activity downstream of fibronectin-integrin signaling.

In conclusion, OPTN positively controls the ubiquitylation of Rac1 by HACE1 E3 ubiquitin ligase, thereby limiting Rac1 activity in cells.

## OPTN couples extracellular matrix stiffness sensing with cell proliferative behavior and cyclin D1 expression

Rac1 activity is crucial for translating ECM stiffness into mechanosensitive cell cycling (refs. 24, 48). Since OPTN adapts HACE1 E3 activity toward Rac1 to ECM stiffness, we analyzed the role of OPTN in cell proliferative behavior. Elastic properties of the ECM have profound effects on proliferation. Tumor and fibrotic tissue microenvironments are often stiffer, a physical parameter that contributes to aberrant hyperproliferation and pathological outcomes[25,44,45]. Conversely, cells cultured on low stiffness ECM have decreased cell cycle entry and cyclin D1 expression, and the softening of tissues inhibits tumor development[29,44]. We monitored cellular proliferation on fibronectin-coated low (1 kPa) and higher (8 kPa) stiffness hydrogels upon FACS analysis of DNA content. The proportion of the control cells that entered the S and G2/M phases of the cell cycle increased by 69% on high stiffness compared to low stiffness ECM (Fig. 4a), confirming that ECM stiffening stimulates cell proliferation in our experimental setup. Interestingly, the OPTN knockdown cells reached maximal proliferation when plated on low elastic modulus ECM. This finding suggests that the OPTN knockdown cells gained the capacity to proliferate independently of ECM stiffening (Fig. 4a). The analysis of cyclin D1 levels showed a stiffness-dependent increase in the control cells (Fig. 4b and Supplementary Fig. 4A). Notably, the OPTN knockdown cells presented higher cyclin D1 levels, and the amount in these cells cultured on low stiffness (1 kPa) was greater than that of the control cells cultured on higher stiffness (8 kPa) ECM (Fig. 4b and Supplementary Fig. 4A). The increase in cyclin D1 was lost upon inhibition of Rac1 by EHT1864 (Fig. 4b). Rac1 controls cyclin D1 transcription[66], and HACE1 knockdown cells display enhanced Rac1-dependent cyclin D1 transcription and proliferation[23]. We found that the increase in cyclin D1 levels in the OPTN knockdown cells was not

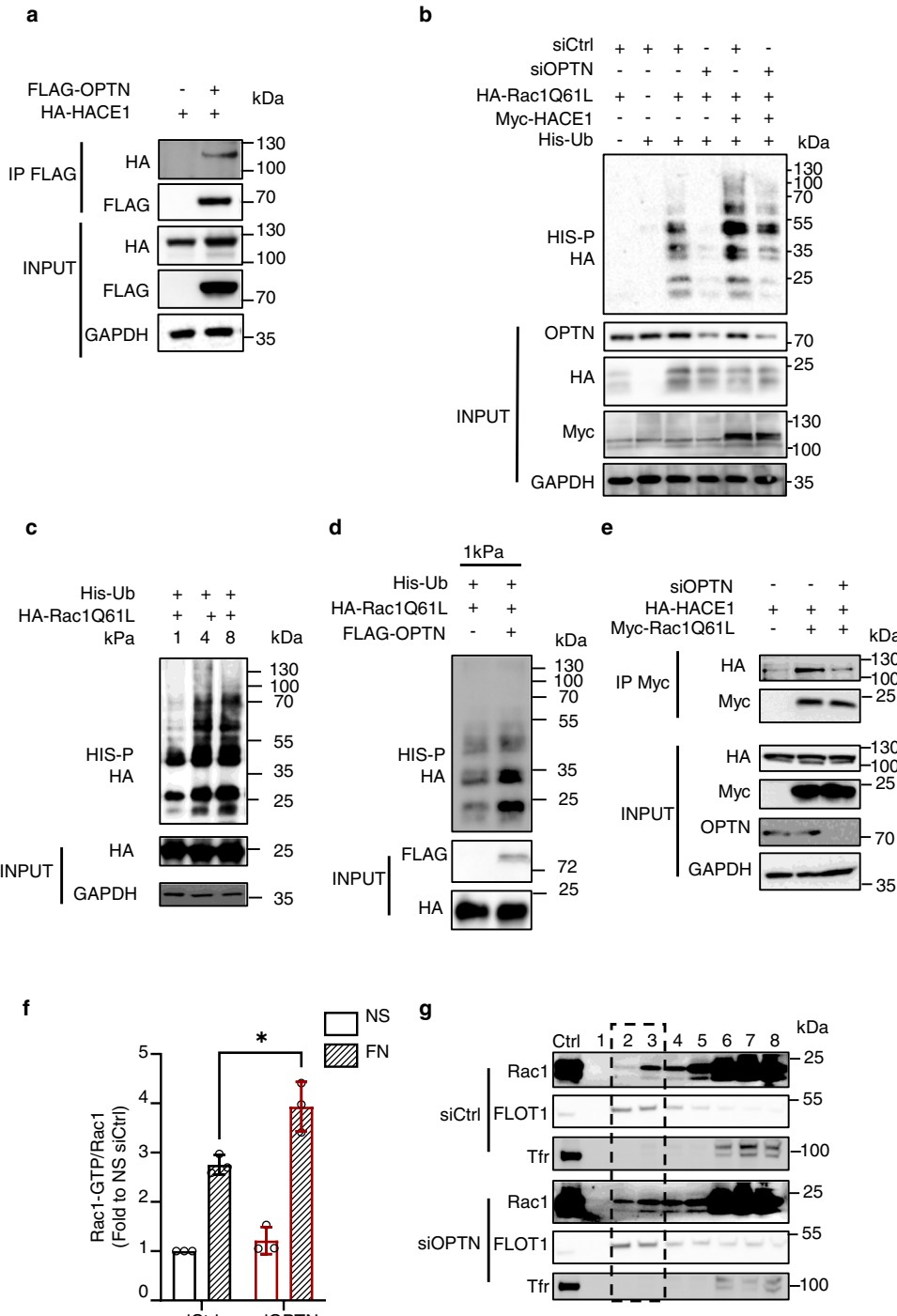

due to enhanced transcription of the gene or stabilization of the mRNA (Fig. 4c). Indeed, both the control and OPTN-silenced cells displayed a similar 4- to 5-fold induction of their mRNA levels in response to mitogen stimulation, while the HACE1 knockdown cells displayed a 9-fold induction of cyclin D1 mRNA levels in response to mitogens. Nevertheless, the OPTN knockdown cells displayed increased cyclin D1 protein levels compared to the controls (Fig. 4c). We therefore sought to analyze cyclin D1 protein stability. The half-life of cyclin D1 was measured by treating proliferating control or OPTN-silenced cells with cycloheximide (CHX) to inhibit the synthesis of proteins, and the time course for the disappearance of cyclin D1 was determined (Fig. 4d, e). Cyclin D1 is known to turnover rapidly. The half-life of cyclin D1 in the control cells was ~12 min, in concordance with published half-lives[67,68]. The knockdown of OPTN had no impact on cyclin D1 stability, as

indicated by the comparison of the decay curves (Fig. 4e). Nevertheless, a noticeable difference in the cyclin D1 level was observed at time 0, when CHX was added (Fig. 4d, e), suggesting a regulation of cyclin D1 protein synthesis. Increased cyclin D1 neosynthesis could be confirmed by metabolic labeling of proteins over a 1-h pulse with L-azidohomoalanine (AHA), followed by purification of labeled proteins from the control or OPTN knockdown cell lysates (Fig. 4f). The enhanced rate of cyclin D1 translation could be verified in an alternative approach, where CHX was thoroughly washed and cells were incubated in chase medium without CHX to reverse translational inhibition. Cyclin D1 resynthesis was markedly enhanced in the OPTN-silenced cells compared to the controls (Supplementary Fig. 4B, C). Altogether, these data demonstrate that silencing OPTN triggers an increase in the cyclin D1 translation rate without affecting cyclin D1

**Fig. 3 | OPTN controls the HACE1-dependent ubiquitylation of Rac1.**
**a** Coimmunoprecipitation of Flag-OPTN with HA-HACE1(IP FLAG) expressed in HUVEC cultured on plastic. Input: immunoblots on 0.5% total lysate to control protein levels. Representative experiment, $n$ = 3. **b** Immunoblot showing Rac1 ubiquitylation profile. HUVECs were transfected with siCtrl or siOPTN and expression vectors for Histidine-tagged ubiquitin, HA-Rac1Q61L and Myc-HACE1. His-Ub covalently bound to Rac1Q61L were purified(His-P) and the ubiquitylation profile of Rac1Q61L was revealed by anti-HA immunoblotting(HA). Input: immunoblots on 0.5% lysate to control protein expression or knockdown. Representative experiment, $n$ = 3. **c** Immunoblots showing Rac1 ubiquitylation profile. Cells were transfected with expression vectors for Histidine-tagged ubiquitin and HA-Rac1Q61L and plated overnight on fibronectin-coated hydrogels of indicated stiffness (kPa). Covalently bound His-Ubiquitinated proteins were purified(HIS-P) and resolved on 12% SDS-PAGE before anti-HA immunoblotting. Input: immunoblots on 1% lysate to control protein expression. Representative experiment, $n$ = 3. **d** Immunoblots showing Rac1 ubiquitylation profile. Cells were transfected with expression vectors for Histidine-tagged ubiquitin and HA-Rac1Q61L, with or without FLAG-OPTN and plated overnight on 1 kPa fibronectin-coated hydrogels. Covalently bound His-Ubiquitinated

proteins were purified(HIS-P) and resolved on 12% SDS-PAGE before anti-HA immunoblotting. Input: immunoblots on 1% lysate to control protein expression. Representative experiment $n$ = 3. **e** Coimmunoprecipitation of HA-HACE1 with Myc-Rac1Q61L (IP Myc) in siCtrl or siOPTN transfected HUVEC cells cultured on plastic. Input: immunoblots on 0.5% lysate, to control protein expression or knockdown. Representative experiment, $n$ = 3. **f** Quantification of Rac1-GTP levels in HUVEC transfected with siCtrl or siOPTN and stimulated by adhesion on fibronectin. Cells were G0 synchronized and active Rac1 level was assessed by GST-PAK pulldown on lysates from non-stimulated cells (NS) or cells plated on 15 μg/ml fibronectin plates for 1 h. Results represent fold change to non-stimulated siCtrl condition, set to 1. Means ± SD, $n$ = 3 independent experiments (single dots). Two-way ANOVA: *$p$ = 0.0121. **g** Accumulation of Rac1 in Flotilin (FLOT1) enriched fractions. HUVECs were transfected with siCtrl or siOPTN and G0 synchronized before plating on fibronectin 15 μg/ml plates for 1 h. Cell lysates were fractioned on a sucrose gradient (40, 30, 0%). The 8 fractions recovered were analyzed by SDS-PAGE and immunoblotted anti-Rac, anti-Flotilin 1(FLOT1) and anti-Transferrin Receptor(Tfr). Immunoblots anti-Tfr show detergent-soluble fractions (6–8) and anti-FLOT1 show detergent-resistant Fractions (2–3). Representative experiment, $n$ = 3.

---

degradation, leading to higher accumulation of the protein. Interestingly, the global rate of protein synthesis was similar in the dividing control and OPTN knockdown cells, suggesting the selectivity of enhanced protein synthesis from some mRNAs, including cyclin D1 mRNA (Fig. 4f).

In conclusion, our data indicate that OPTN controls the ECM-mediated stimulation of cyclin D1 levels by restricting Rac1 activation and thereby adapts the cyclin D1 translation rate and cell proliferation as a function of ECM stiffness.

## OPTN regulates cell-extracellular matrix adhesion

The formation of cell-ECM adhesive structures depends on Rac1 activation[36,69]. Because OPTN levels are controlled by ECM stiffness and OPTN regulates Rac1 activity in cells, we investigated whether OPTN influences integrin adhesion to the ECM and signaling. We monitored cell-ECM adhesion structures in the OPTN knockdown cells by immunolocalization of alpha-5 beta-1 integrin, the receptor for fibronectin in our cells[70], paxillin and zyxin. Interestingly, the OPTN knockdown cells displayed numerous centripetal FAs, as visualized by immunostaining for paxillin, alpha5 beta1 integrin (Fig. 5a and Supplementary Fig. 5A) and zyxin (Supplementary Fig. 5B). Quantitative multiparametric image analysis (QMPIA) of FA staining revealed a higher number of adhesion structures in the OPTN knockdown cells, as well as an increase in the FA mean area of 30% compared to that of the control cells (Fig. 5b and Supplementary Fig. 5C). Accordingly, we observed increased cell adhesion on fibronectin when OPTN was knocked down (Fig. 5c). Collectively, our data suggest that loss of OPTN allows increased formation and/or maturation of cell-ECM adhesions, without inducing the RhoA/ROCK pathway (Supplementary Fig. 3C). To estimate the contribution of Rac1-mediated protrusive forces to this phenotype, we measured cell elasticity by atomic force microscopy (AFM). This nanoindentation approach is based on following on a photodiode the displacement of a laser beam reflected on top of a cantilever holding on the other side a geometry-defined tip. The tip operates approach-contact-indentation-retraction cycles through piezo displacements over the cell area and force-distance curves are recorded. Fitting these curves with Hertz' model allows extracting the Young's modulus of the sample at a given location and indentation[71]. Here we analyzed the modulus corresponding to the plasma membrane and underlying cytoskeleton and cytoplasm (50 nm). In accordance with the contribution of unrestrained Rac1 activation to a gain in adhesive functions, we measured a significant increased elasticity of the cell cortex by AFM probing of the OPTN knockdown cells (9.7 ± 0.3 kPa, $n$ = 29 cells) compared to the control cells (8.4 ± 0.4 kPa, $n$ = 21 cells), which we could inhibit using the Arp2/3 pharmacological inhibitor CK-666 (5.9 ± 0.4 kPa, $n$ = 29 cells) (Fig. 5d

and Supplementary Fig. 5D). The OPTN KD cells also displayed elevated levels of F-actin compared with the control cells (Fig. 5e), in line with Rac1-driven enhanced actin polymerization. Moreover, the spreading of OPTN knockdown cells on FN-coated surfaces was markedly relying on Rac1 activity compared to control cells, with a decrease of 69% cell area in OPTN KD cells compared to 38% in control cells upon treatment with the Rac1 inhibitor EHT1864 (Supplementary Fig. 5E).

As a complementary approach, we expressed OPTN in cells and performed adhesion assays and cell-ECM adhesion structures monitoring by immunolocalization of alpha-5 beta-1 integrin and paxillin when cells adhered to fibronectin (Supplementary Fig. 5F, G). Mirroring our loss-of-function approach, OPTN expression was associated with a 50% loss of cell adhesion on fibronectin (Supplementary Fig. 5F) and decreased centripetal FAs at the expense of small peripheral dot-like adhesion structures (Supplementary Fig. 5G).

Since OPTN regulates integrin-mediated cell-ECM adhesion, we analyzed whether OPTN more generally impacts integrin-mediated mechanosignalling. For this, the control or OPTN-silenced cells were plated on a fibronectin matrix of increasing rigidity and we measured by western blot the phosphorylation levels of FAK-Y[397], Src-Y[416], Paxillin-Y[118] and p130-CAS-Y[410] in cell lysates. FAK phosphorylation was induced by ECM rigidity in the control and the OPTN-silenced cells with no significant differences (Fig. 5f). Interestingly, Src family kinase activity was induced by ECM rigidity in the control cells and was strongly induced in the OPTN knockdown cells with similar strength of activation in response to low and high stiffness ECM (Fig. 5f). The phosphorylation of paxillin on tyrosine 118, which depends on the FAK-Src kinase module in an ECM rigidity-dependent manner[55], was gradually induced in the control cells and displayed stronger levels in the OPTN-silenced cells (Fig. 5f). Similarly, p130cas phosphorylation on tyrosine 410, which is regulated by stretch-induced unfolding of the protein[56], was strongly stimulated in the OPTN-silenced cells (Fig. 5f). In conclusion, we observed that OPTN knockdown led to high and ECM stiffness-independent activation of Src family kinases, leading to enhanced adhesion-mediated signaling.

## OPTN regulates integrin-mediated cellular invasion by uropathogenic *E. coli*

We have established that integrin mechanoactivation is required for UPEC invasion induced by CNF1 (Fig. 1). We monitored the impact of OPTN levels on bacterial adhesion and invasion into HUVEC cells. OPTN knockdown was accompanied by a 41% reduction in Ec invasion (Fig. 6a) while it had no effect on bacterial adhesion to the cell surface

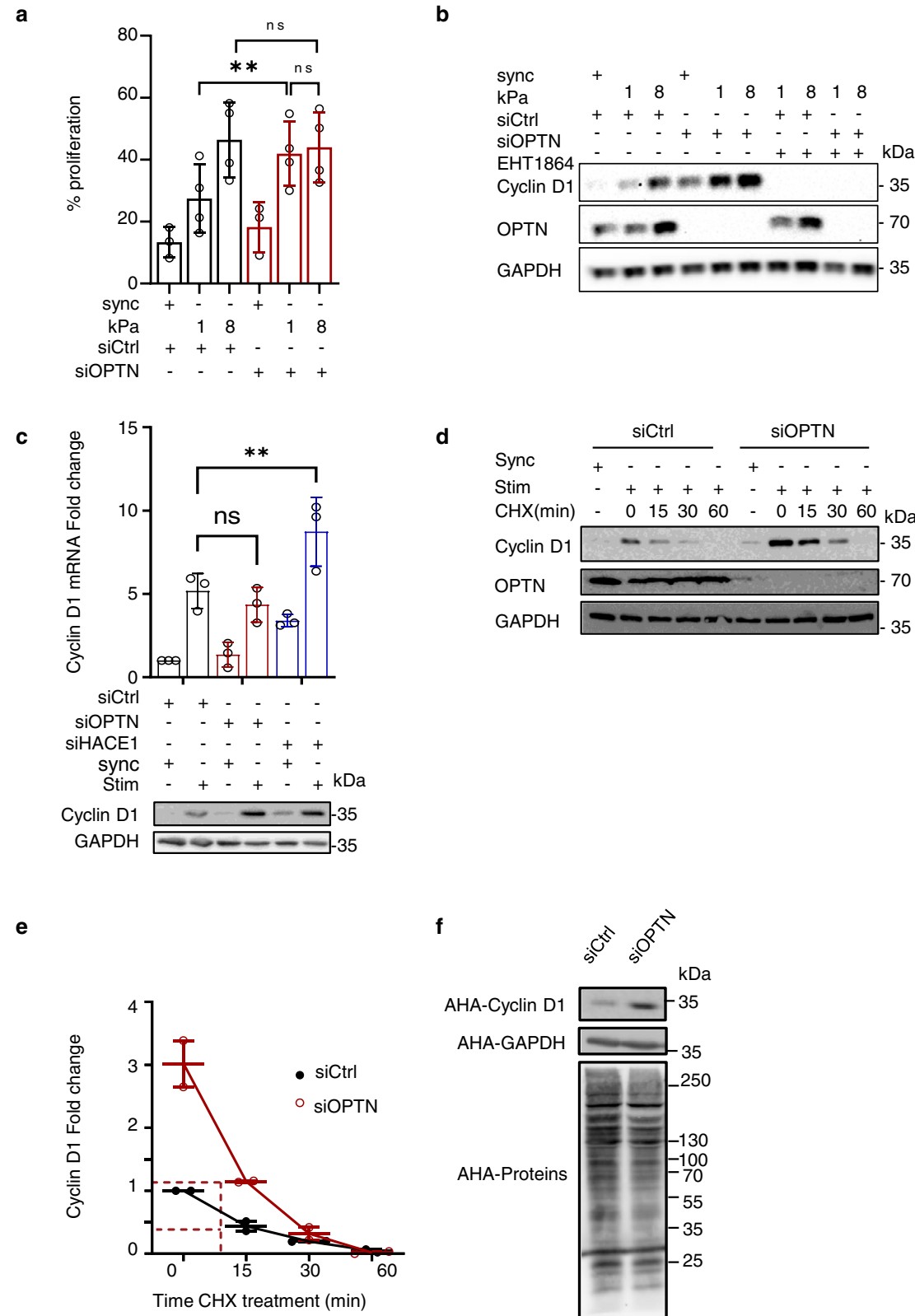

(Supplementary Fig. 6A). We could also establish that the role of OPTN for Ec invasion is conserved in the bladder epithelial cell line 5637 (Supplementary Fig. 6B).

Rac1 is directly activated by the CNF1 deamidase toxin from uropathogenic *E. coli*, leading to its sensitization to HACE1-dependent ubiquitin-mediated proteasomal degradation[16,22,72]. CNF1 treatment stimulated endogenous Rac1 activity by 5-fold in the control cells vs.

7.6-fold in the OPTN knockdown cells (Fig. 6b), confirming that OPTN also restricts the pool of CNF1-stimulated Rac1. RhoA-ROCK activity was not different in OPTN knockdown cells compared to control knockdown cells stimulated by the CNF1 toxin, as assessed on the phosphorylation of myosin light chain (Supplementary Fig. 6C).

In accordance with increased Rac1 stimulation, the measurement of cortical elasticity by AFM shows stiffening of the cell when

**Fig. 4 | OPTN regulates adhesion-mediated cell cycle progression and translation of cyclin D1. a** Graph representing the percentage of proliferating cells (S and G2/M phases) assessed by FACS analysis of DNA content. HUVEC transfected with siCtrl or siOPTN were G0 Synchronized(sync) and plated at low confluence on fibronectin-coated hydrogels of indicated stiffness (kPa) for 18 h before cell cycle analysis by FACS. Means ± SD, $n = 4$ independent experiments (single dots). One-way ANOVA with Tukey's multiple comparison test: ns, siCtrl8kPa vs. siCtrl 1 kPa $p = 0.1501$; siOPTN 8 kPa vs. siOPTN 1 kPa $p = 0.9997$ not significant; Two-way ANOVA with Sidak's multiple comparison test: siCtrl-siOPTN 1 kPa $p = 0.0030$. **b** Immunoblots showing cyclin D1 expression in cells treated as in **a**. Representative experiment, $n = 3$. **c** Level of cyclin D1 mRNA and protein expression in siCtrl or siOPTN transfected cells. HUVECs were G0 synchronized (sync) and then plated on 15 μg/ml fibronectin-coated plates and 20% serum for 18 h (Stim). Cyclin D1 mRNA levels were assessed by RT-qPCR and normalized to siCtrl synchronized cells condition, set to 1. Mean ± SD, $n = 3$ independent experiments (single dots). Two-way ANOVA with Tukey's correction for multiple comparisons: ns, $p = 0.6364$ not significant; **$p = 0.0047$. In parallel, cyclin D1 protein levels were determined by immunoblots, GAPDH was used as loading

control. **d** HUVEC transfected with siCtrl or siOPTN were G0 Synchronized (sync) and then plated on 15 μg/ml fibronectin-coated plates and 20% serum for 18 h (Stim), together with Cycloheximide (CHX) at 10 nM. Cells were harvested after 15, 30, 60 min treatment and processed for anti-cyclin D1 immunoblotting. Immunoblots anti-OPTN and anti-GAPDH were performed to control OPTN knockdown and protein loading, respectively. **e** Quantification of cyclin D1 half-life measured in **d**. Values are expressed as fold compared to level in untreated condition (time 0), set to 1. Mean ± SD, $n = 3$. Dashed lines show similar cyclin D1 half-life in siCtrl and siOPTN cells. **f** Immunoblots showing neo synthesized cyclin D1 upon metabolic labeling. HUVEC transfected with siCtrl or siOPTN were G0 Synchronized and then plated on 15 μg/ml fibronectin-coated plates in defined medium for 18 h. Cells were pulsed with AHA during the last hour. Cell lysates were subjected to Click-iT reaction with Biotin Alkyne followed by streptavidin pulldown. Cyclin D1 and GAPDH neosynthesis was then assessed upon immuno-blotting the resolved streptavidin-pulled-down reaction product. Total neo-synthesis of proteins was assessed upon immunoblotting the resolved total Click-iT reaction product with streptavidin-HRP (AHA-Proteins). Representative experiment, $n = 3$.

CNF1 stimulation is applied to the control cells ($13.3 ± 0.4$ vs. $10.2 ± 0.3$ kPa, Fig. 6c and Supplementary Fig. 6D middle panels). Knocking down OPTN led to further enhancement of the cell elastic modulus, which reached a 56% increase compared to that of the control non-treated cells ($15.9 ± 0.5$ kPa vs. $10.2 ± 0.3$ kPa, Fig. 6c and Supplementary Fig. 6D bottom panels). These measures indicate that although strong protruding forces are exerted against membrane downstream Rac1 activation, this is not enough to allow bacterial internalization. We then monitored integrin mechanosignalling fostered by CNF1 and bacteria at early infection steps. Indeed, we measured a defect in phosphorylation of FAK, Src kinases, Paxillin and P130CAS proteins at mechanoresponsive sites when OPTN was knocked down (Fig. 6d). This situation indicates that OPTN is involved in the mechanical coupling of integrins with the cytoskeleton in the context of bacterial invasion. To substantiate that such mechanical coupling is a limiting step for bacterial invasion into cells, we disrupted the integrin-actin linkage upon expressing the Talin head domain in cells. This construct leads to inside-out activation of beta1 integrins while uncoupling them from the cytoskeleton, thereby interfering with mechanical reinforcement of their engagement[73]. Upon infecting cells expressing the Talin head, bacteria encountered a strong restriction of their invasive capacities, with a 70% reduction in internalization (Fig. 6e and Supplementary Fig. 6E). This result indicates that Ec needs to trigger integrin mechanical activation and force transmission to efficiently invade cells, a situation that may be deficient in the OPTN knockdown cells.

## OPTN modulates the cell tensile forces exerted on the extracellular matrix

Adhesion structures are complex platforms of proteins with a nanoscale stratified architecture linking the actomyosin cytoskeleton to the ECM. Thus, adhesion structures transmit traction forces from actin stress fibers to the ECM and enable cells to sense the mechanical properties of the ECM[40,41,74,75]. Moreover, traction force generation on adhesion structures contributes to their growth and maturation by reinforcing protein clustering, allowing protein recruitment, integrin conformational activation and the stretching and unfolding of components to expose cryptic sites and enhance adhesion-mediated signaling[75–78]. Since OPTN is regulated by ECM compliance and modulates integrin-driven processes, we sought to determine whether OPTN could regulate cell traction forces. We first used the parallel microplates technique, which allows real-time measurement of the force generated by a single-cell on fibronectin-coated plates[79]. In this setup, a single-cell is caught between two parallel plates, one rigid and the other flexible with a calibrated stiffness $k$. The traction force $\mathbf{F}$ is measured through the deflection $d$ of the flexible plate: $\mathbf{F} = k\,d$ (Fig. 7a). Given that cell traction forces increase with the rigidity of the

substrate, we decided to test whether OPTN influences the maximum traction force the cells can generate, i.e., in infinite stiffness conditions[80]. Representative individual traction force curves (Fig. 7b) illustrate the difference in behavior between the control (siCtrl) or the OPTN knockdown cells (siOPTN). The rate of force buildup (slope of the force curve, $d\mathbf{F}/dt$) and the maximum traction force Fmax (plateau after ~20 min) were lower in the OPTN knockdown cells. These observations were statistically confirmed with Fmax = $120 ± 13$ nN and $d\mathbf{F}/dt = 0.17 ± 0.02$ nN/s in the control conditions (19 cells) vs. Fmax = $64 ± 11$ nN and $d\mathbf{F}/dt = 0.09 ± 0.01$ nN/s (17 cells) in the OPTN-depleted cells (Fig. 7c, d). Altogether, these data indicate that the OPTN knockdown cells build up half of the force that is exerted by the control cells when they adhere to fibronectin, a situation that strikingly contrasts with their ability to generate numerous and large FAs.

We then sought to determine whether the decrease in contractile activity of the OPTN knockdown cells could also be observed in a more usual geometry of 2D-culture substrates, i.e. using traction force microscopy, and using a different (lower) stiffness. To this end, control and OPTN knockdown cells were plated for 2 h on 5 kPa fibronectin-coated hydrogels with embedded fluorescent nanobeads. Traction forces exerted by spread cells cause deformation of the hydrogel, as shown by the displacement of the beads. Bead displacements were measured using a combination of particle imaging velocimetry and single particle tracking onto bead images of the same fields with and without cells on top[81]. Traction forces were then calculated using fast Fourier transform traction cytometry[82] from the measured displacement fields. Maps of traction force magnitude are shown in Fig. 7e for representative cells from the OPTN and control knockdown conditions. Traction stress was similarly distributed at peripheral zones of the cells in both the control and OPTN knockdown groups, without major differences in location. Interestingly, the OPTN knockdown cells displayed a reduced magnitude of traction compared to the control cells (Fig. 7e). We found that the OPTN knockdown cells generated 3.5-fold less traction than their control counterparts, with a mean force value of $72.46 ± 6.86$ nN for the OPTN-depleted cells ($n = 41$ cells) and $249.50 ± 24.32$ nN for the control cells ($n = 40$ cells) (Fig. 7f). We also calculated the total contractile energy generated by cells, which was obtained by integrating the scalar product of the force and displacement vectors across the entire cell. In accordance with total force measures, the contractile energy produced by the OPTN knockdown cells was lower than that produced by the control cells (Fig. 7g). This decrease in traction force measured for the OPTN knockdown cells plated on hydrogels could be replicated in an alternative setup using microfabricated pillars of 9 nN/μm stiffness (equivalent to 6 kPa gels) coated with fibronectin (Supplementary Fig. 7)[83]. The control and OPTN knockdown cells adhered similarly on pillars, and typical traction maps obtained after 3 h for each cell population showed that the

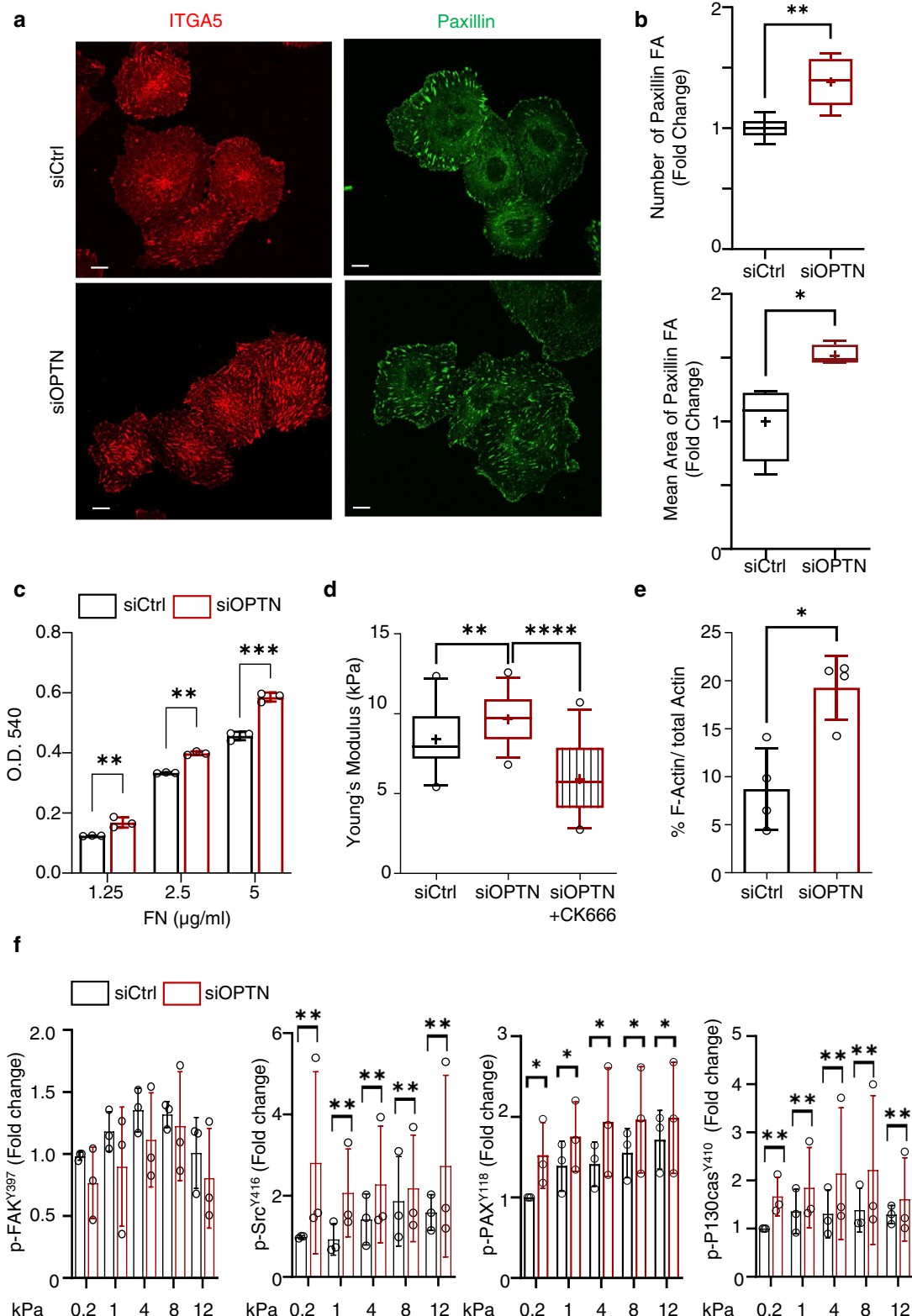

OPTN-depleted cells had a reduced capacity to pull on pillars, while the control cells induced substantial centripetal deflection of pillars located at the cell periphery (Supplementary Fig. 7A). The control cells reached a mean total force per cell of 305.50 ± 23.86 nN and expended 47.92 ± 5.93 fJ contractile energy ($n$ = 32 cells) (Supplementary Fig. 7B, C), while the OPTN knockdown cells exerted a mean of 180.40 ± 15.28 nN total force per cell with a 13.79 ± 2.06 fJ mean contractile energy per cell ($n$ = 33 cells) (Supplementary Fig. 7B, C).

Altogether, we found by three independent approaches that loss of OPTN expression is associated with a decrease in total force transmitted to the environment during cell adhesion. Thus, while the OPTN knockdown cells generated reduced traction forces, they nevertheless formed numerous FAs when adhered to fibronectin and displayed higher Rac1 activity and an increased capacity to adhere to fibronectin. However, the reduced capacity of the OPTN-depleted cells to build traction forces limits CNF1-producing UPEC invasion into cells, even if

**Fig. 5 | OPTN regulates cell adhesion to the extracellular matrix. a** Confocal sections showing enhanced formation of integrin alpha-5-positive (ITGA5) and paxillin-positive focal adhesions (FA) in siOPTN transfected HUVEC plated 2 h on fibronectin-coated coverslips in defined medium. Bar = 10 µm. Representative experiment, $n = 3$. **b** Number and mean area of FA structures positive for paxillin in siCtrl or siOPTN transfected HUVEC were quantified. Results are expressed as fold change to siCtrl condition, set to 1 (boxplots). $n = 30$ SiCtrl cells and 27 SiOPTN cells, three independent experiments. Wilcoxon rank sum test (two-tailed $p$ value): *$p = 0.0286$, **$p = 0.0043$. **c** Adhesion of siCtrl or siOPTN transfected HUVEC on 1.25, 2.5 and 5 µg/ml of fibronectin was assessed upon short-term plating of cells in triplicate in 96-well plates. Bars represent the extent of adhesion to fibronectin as function of absorbance upon elution of the crystal violet dye. Means ± SD, $n = 3$ independent experiments (single dots). Two-way ANOVA with Sidak's multiple comparison test $p$ value: siCtrl-siOPTN 1.25 *$p = 0.0080$; siCtrl-siOPTN 2.5 *$p = 0.0018$; siCtrl-siOPTN 5 ***$p = 0.0001$. **d** Cortical elastic modulus determined by AFM on cells plated for 1 h 30 on Fibronectin plates in defined medium, boxplots. siCtrl ($n = 21$ cells), siOPTN

($n = 29$ cells) and siOPTN cells treated with CK-666 (250 µM) to inhibit Arp2/3 activity ($n = 29$ cells). One-way ANOVA with Dunnett's correction test for multiple comparisons $p$ value: ****$p \leq 0.0001$. **e** Quantification of filamentous actin levels (F-Actin) to total actin (F + G-actin) in control and OPTN knocked down cells. Mean values ± SD, $n = 4$ independent experiments (single dots). Wilcoxon rank sum test (two-tailed): *$p = 0.0286$. **f** HUVEC transfected with siCtrl or siOPTN were G0 Synchronized and then plated at low confluence in defined medium on fibronectin-coated hydrogels of indicated stiffness (kPa) for 1 h. Immunoblots anti-phospho-FAK-tyr397, anti-phospho-Src-tyr416, anti-phospho-Paxilin-tyr118 and anti-phospho-p130CAS-tyr410, together with Anti-FAK, Src, Paxillin, p130CAS and GAPDH as control were performed. Graphs represent quantification of phospho-FAK(tyr-397), phospho-Src(tyr-416), phospho-Paxillin(tyr-118) and phospho-p130CAS(tyr-410) signals to total proteins and normalized to GAPDH. Densitometry was normalized to value on 0.2 kPa ECM, set to 1. Bars represent means ± SD, $n = 3$ independent experiments (single dots). Wilcoxon rank sum test (two-tailed $p$ value): p-FAK ns, $p = 0.1508$; p-Src **$p = 0.0079$; p-PAX *$p = 0.0317$; p-P130Cas **$p = 0.0079$.

strong Rac1 activation occurs. Collectively, our data indicate a role of OPTN in coupling cell-ECM adhesion structures with intracellular traction forces and with the tuning of Rac1 activity via HACE1-dependent proteasomal degradation of the small G active form.

## Discussion

Our work identifies a role of OPTN in integrin mechanotransduction via modulation of HACE1 E3 ligase-mediated ubiquitylation of Rac1 in response to ECM rigidity and via force transmission by integrins. OPTN levels are regulated by ECM stiffness to ensure proper tensional integrity and adaptation of the cyclin D1 translation rate with ECM mechanical properties for accurate cell cycle progression. During infection by CNF1-producing uropathogenic *E. coli*, OPTN is required for efficient bacterial invasion and associated mechanochemical reinforcement of integrins. Hence, an increase in ECM stiffness and mechanical coupling of integrins to the actin cytoskeleton are required for *E. coli* invasion. These findings demonstrate a role of tissue rigidity in host cell susceptibility to invasive *E. coli* infection.

How does mechanical coupling of integrins play a role in Ec invasion? Force transmission to integrins can stabilize or reinforce their binding to the bacterial FimH adhesive molecules, meaning that adhesion could switch to a "catch-bond" behavior of both the integrin and the FimH adhesin of the pilus. Force application on integrins can also modify their rate of endocytosis, but little is known about the influence of ECM rigidity on integrin endocytosis. Beta1 integrin studies show higher endocytosis on soft ECM, as exemplified by experiments conducted on mesenchymal stem cells cultured on collagen, showing enhanced caveolae and Raft-dependent endocytosis on soft ECM[84]. Similarly, the ILK-Pinch-Parvin complex reinforces integrin-ECM adhesion in *Drosophila* embryos in response to tension and decelerates integrin turnover at the plasma membrane[85]. Bacterial invasion being higher on high stiffness ECM, a direct parallel with a diversion of integrin endocytosis by bacteria for their uptake seems paradoxical in the actual state of knowledge. An interesting parallel can be drawn with professional phagocytic cells for which recent studies showed the requirement of force coupling for phagocytosis and the sensitivity of the system to the rigidity of the engulfed particles. In this situation, branched and linear actin networks work together to generate the forces triggering phagocytosis. A molecular clutch is engaged, resisting actin retrograde flow, comprising Talin, which connects β2 integrins to the cytoskeleton, and vinculin, whose recruitment is enhanced proportionately to the rigidity of the opsonized target, stabilizing the molecular scaffold and fortifying the linkage. Interestingly, while Arp2/3 and mDia are involved, the role of myosin II is not clear; instead, Syk kinases are functionally required to stabilize the clutch[86]. In the present situation, OPTN could be involved in the formation or stabilization of a clutch by a mechanism that

remains to be investigated, thereby allowing internalization of integrin-bound bacteria. The preemption of cellular biomechanics by bacterial pathogens has not yet been thoroughly investigated. Interestingly, one particularity of UPEC is its capacity to escape endocytic vesicles and proliferate in the cytosol to form intracellular bacterial communities in the bladder. A very recent publication showed that endosomal escape is facilitated when bladder epithelial cells are cultured on low stiffness ECM[87]. The other example is provided by *Listeria monocytogenes*, whose adhesion to cells is increased with greater ECM stiffness; however, its invasion efficiency is unaffected[88].

It has been firmly established that the regulation of Rac1 by ubiquitylation is a function of its strength of activation. This holds true for activation via the action of GEF regulatory proteins, point mutations and bacterial toxin-mediated Q61 deamidation of Rac1. Here, we report the first regulator of HACE1 E3 ubiquitin ligase activity, Optineurin, the expression of which is induced by the rigidity of the extracellular matrix. Moreover, OPTN expression in cells plated on a matrix of low compliance increased the level of Rac1 ubiquitylation. Thus, OPTN restrains the threshold of Rac1 activation in response to an increase in matrix stiffness by controlling the extent of degradation of the Rac1 active form. Interestingly, the filamin/FilGAP system has been shown to limit Rac1 activation in response to mechanical stretch during embryonic remodeling of the mitral valve[25]. Indeed, when the actin-binding protein filamin A is mechanically deformed, its rod2 region is spatially separated, thereby regulating FilGAP accessibility and Rac1 inactivation[89]. Therefore, the OPTN-HACE1 system that we describe represents a second example of negative regulation of Rac induced by mechanical strain of the actin cytoskeleton.

We showed that OPTN controls Rac1 activity and FA number and size. Nascent adhesions mature into focal complexes and then larger and elongated FAs through myosin II-mediated actomyosin contraction. FAs respond to tension by changing the composition and activation status of some proteins and triggering enhanced signaling[90–92]. Interestingly, tension-induced FA maturation involves the decrease of Rac1 activity through the dissociation of activating proteins, such as β−PIX, and effectors[90]. Our data showing that OPTN is regulated by ECM stiffness and is a negative regulator of Rac1 activity through induction of its degradation by HACE1 is in complete agreement with the notion that Rac1 has to be restrained for FA maturation, while FA turnover involves Rac1 activity. Interestingly, we found that the OPTN-silenced cells exhibit paradoxically impaired traction force buildup when adhered to fibronectin, although they display numerous and larger FAs than the control cells and have increased fibronectin attachment. Probing critical components downstream of integrin signaling indicated that hallmarks of contraction-dependent FA signaling are fulfilled in the OPTN-silenced cells. For example, high levels of FAK-Src-dependent phosphorylation of paxillin on Y118[55] and p130CAS

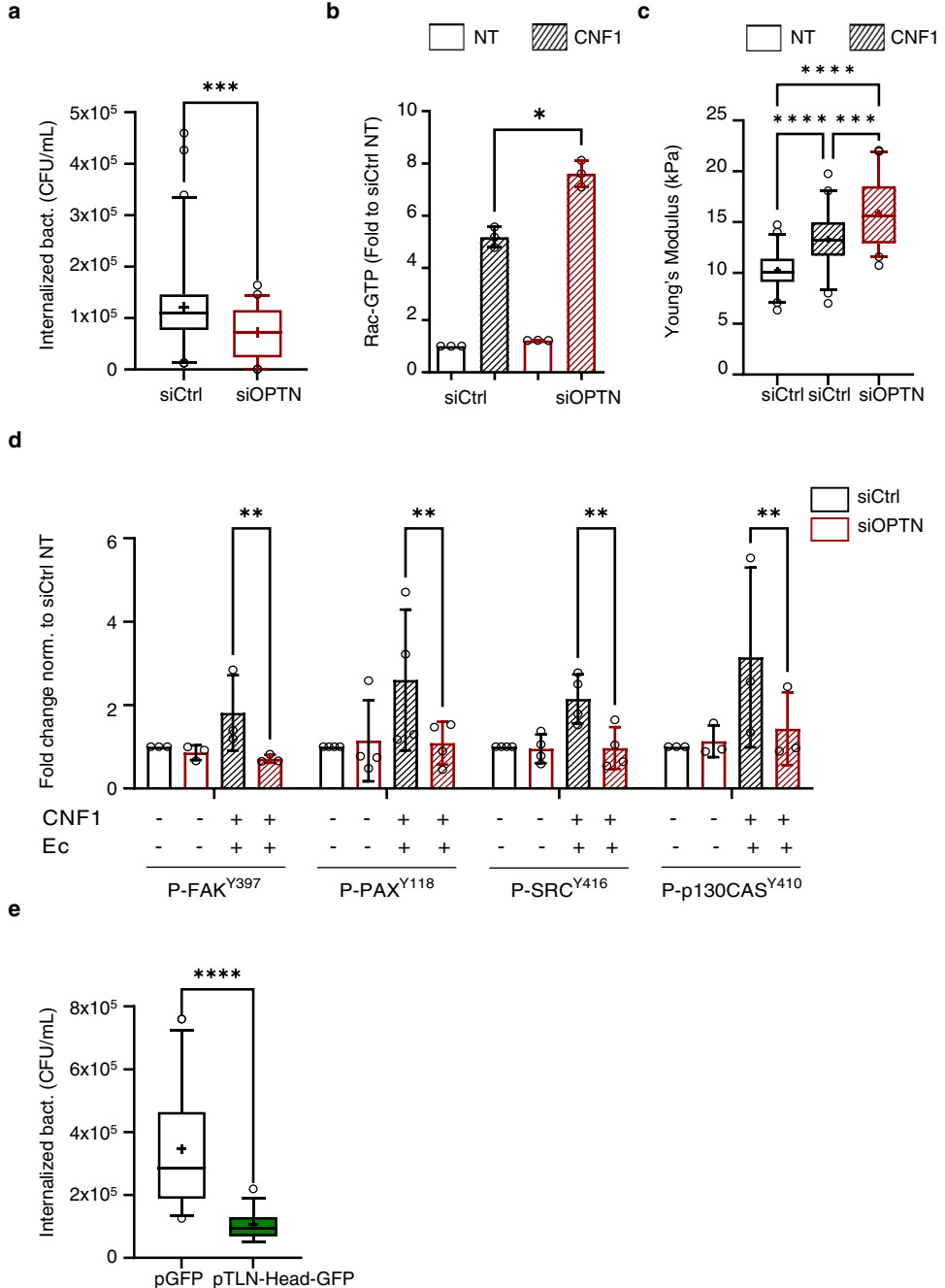

**Fig. 6 | OPTN controls cellular invasion by Ec and impairs CNF1 and FimH-mediated mechanochemical signaling of integrins. a** Quantification of internalized Ec 30 min after gentamicin treatment in siCTRL (control) or siOPTN (optineurin) HUVEC cells on plastic treated with 1 nM CNF1. Graph displays absolute CFU/ml. Boxplots of $n = 5$ independent experiments and three replicates per condition. Wilcoxon rank sum test (two-tailed): ***$p$ = 0.004. **b** Rac1 activation in HUVEC transfected with siCtrl or siOPTN on plastic and either untreated (UT) or treated 2 h with 1 nM CNF1 (CNF1), measured by GST-PAK affinity purification. Rac1-GTP levels were quantified and reported to total Rac1 and GAPDH. Fold change to untreated siCtrl condition, set to 1. Mean ± SD, $n = 3$ independent experiments (single dots). One-way ANOVA with Dunnett's test for multiple comparisons: *$p$ = 0.0460. **c** Cortical elastic modulus determined by AFM on cells cultured overnight on plastic in complete medium and either untreated or treated 2 h with 1 nM CNF1. SiCtrl ($n = 45$ cells), SiCtrl + CNF1 ($n = 44$ cells) and SiOPTN + CNF1 ($n = 41$ cells). Boxplots of $n = 5$ independent experiments and three replicates per condition. One-way ANOVA

with Dunnett's correction test for multiple comparisons $p$ value: ***$p$ = 0.0005; ****$p \leq 0.0001$. **d** Quantification of phospho-protein levels for phospho-FAK (tyr-397), phospho-Paxillin (tyr-118), phospho-Src (tyr-416) and phospho-p130CAS (tyr-410) in siCTRL (white bars) and siOPTN (hatched bars) cells on plastic, either non-treated (−) or after 30 min of Ec infection in presence of 1 nM CNF1. Bars display levels of phospho-proteins relative to GAPDH determined by immunoblotting and normalized to the siCTRL non-treated condition. Mean ± SD of $n = 3$ (FAK and p130CAS) or $n = 4$ (SRC and PAX) independent experiments (single dots). Two-way ANOVA with Tukey's correction test for multiple comparisons: **$p$ = 0.0021 $\leq$ 0.01. Total levels of the protein do not change in the experimental conditions. **e** Quantification of internalized Ec in HUVECs transfected with empty vector (pGFP) or Talin-Head expressing vector (pTLN-head-GFP), after 30 min of gentamicin treatment and under 1 nM CNF1 treatment, MOI:100. Boxplots display absolute number of CFU/ml of $n = 3$ biologically independent experiments and three replicates per condition. Wilcoxon rank sum test (two-tailed): ****$p \leq 0.0001$.

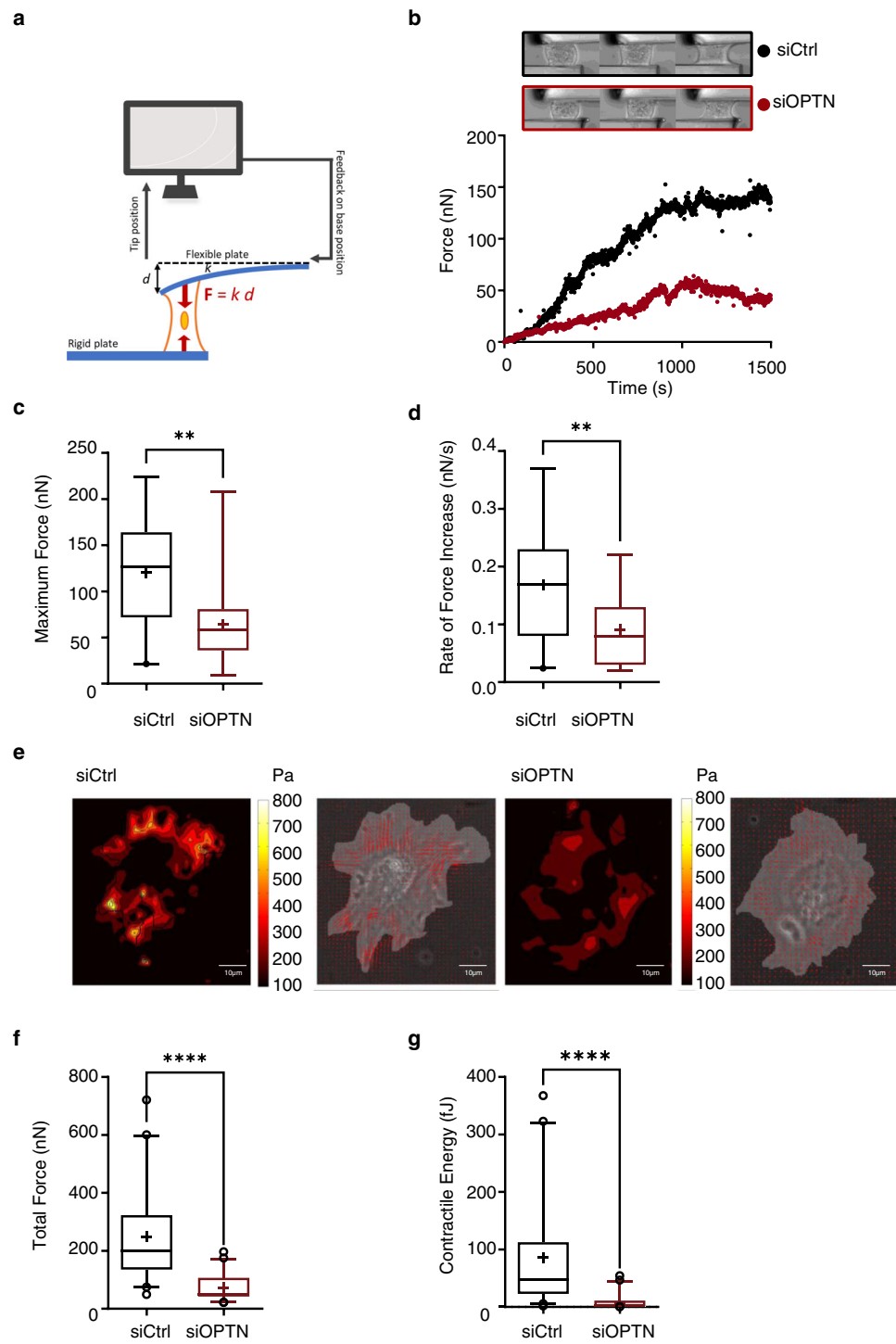

substrate domain on Y410[56], which are both exposed upon tension-induced stretching of the protein, were achieved upon adhesion of the OPTN knockdown cells. Furthermore, zyxin, a protein enriched in FAs in response to tension[91,93], was reinforced at FAs of the OPTN knockdown cells. The monitoring of Myosin light chain phosphorylation, indicative of the RhoA-Rock pathway stimulation, showed no increase in OPTN knockdown compared to control cells in response either to fibronectin adhesion or CNF1 toxin treatment. The phenotype we describe thus appears paradoxical in the current view of FA formation and maturation, since the OPTN-silenced cells seem to form mature FAs that are uncoupled from tension. This apparent paradox can be discussed in the context of the increased Rac1 activity measured in the OPTN-silenced cells. It has now been established that there is a

protrusive force originating from actin polymerization activity at early adhesion sites[94] that might be involved in the stretching of mechanoresponsive proteins. In line with this, p130cas substrate domain phosphorylation, induced by stretching of the cytoskeleton[56], was also shown to be stimulated upon N-WASP-Arp2/3-driven actin polymerization[26]. Accumulation of active Rac1 in the OPTN-silenced cells may have intensified this response. Moreover, recent reports have demonstrated a role of stress fiber assembly at the adhesion site as a structural template that facilitates maturation, demonstrating that 20% myosin-dependent cellular tension is enough for compositional maturation of FAs[37]. In our cells, knocking down OPTN led to a decrease of 40% to 70% in traction forces, depending on the experimental setup, which indicates that we are in a range of tensions

**Fig. 7 | Loss of OPTN is associated with a decrease in adhesion-mediated cell forces.** Measurement of cellular forces using single-cell traction force (**a**–**d**) and traction force microscopy (**e**–**g**) methods. **a** Schematic illustration of the single-cell traction force experiment. One isolated cell is spreading between parallel glass microplates, one rigid, the other flexible and used as a spring of calibrated stiffness to allow one to measure cell traction force, $F = k\,d$, with **F** the traction force, $d$ the flexible plate deflection and $k$ its stiffness. A feedback loop acts on the flexible plate to maintain a cell constant length, thus corresponding to infinite stiffness conditions. Details on the principle of the method in ref. 80. **b** Individual traction force curves for control (siCtrl, black) and OPTN knocked-down (siOPTN, red) single HUVEC cells. The inset shows typical shape changes both for siCTRL and siOPTN at 15, 60 and 3000 s. **c** Maximum traction force values for siCtrl and siOPTN cells are represented in boxplots. (Three independent experiments, $n = 19$ siCtrl and 17 siOPTN cells, Wilcoxon rank sum test (two-tailed): ** $p = 0.0017$). **d** The rate-of-force generation dF/dt is significantly lower in SiOPTN cells compared to control

cells, Boxplots of $n = 19$ siCtrl and 17 SiOPTN cells, three independent experiments. Wilcoxon rank sum test (two-tailed): ** $p = 0.0067$. **e** Traction force microscopy was applied to siCtrl and siOPTN cells on fibronectin-coated 5 kPa polyacrylamide gel. Displacements of beads embedded in the gel are used to reconstruct the stress map (magnitude of the traction forces color coded: brighter signals indicate higher force, see scale on the right), which can also be represented as a vector field (right image: superimposition of bright field image with the defined region used for computing and displacement vectors). Scale bar = 10 μm. **f** Total traction force values for siCtrl and siOPTN cells. Boxplots for $n = 40$ siCtrl and 41 siOPTN cells, two independent experiments. Forces are significantly lower for OPTN knocked-down cells. Wilcoxon rank sum test (two-tailed): **** $p \le 0.0001$. **g** The contractile energy measures the scalar product of the traction by the displacement integrated over the cell surface to give the total energy transferred by the cellular adhesive surface to the elastic substratum (boxplot). Contractile energy is significantly decreased upon loss of OPTN expression. Wilcoxon rank sum test (two-tailed): **** $p \le 0.0001$.

compatible with a contribution of an actin template-driven maturation mechanism in OPTN knockdown cells. Pertinent studies also showed that deformation of the cell cortex, a situation experienced by cells upon Rac-driven branched actin polymerization, induces a rapid increase in cytoplasmic G-actin, which induces mDIA1 processing activity and a subsequent increase in F-actin amounts. This mechanosensitivity of mDIA1 and FRL1 formins requires actin turnover, but RhoA activity is dispensable[95]. Thus, in accordance with these studies, an actin template-mediated mechanism that takes over an actin contraction-mediated process may contribute to FA growth in the OPTN-depleted cells. Alternative mechanisms like a reduced turnover of FAs in OPTN knockdown cells could also contribute to their increase in size. Further researches are needed to investigate such hypothesis and the underlying molecular pathways. Interestingly, some integrin- and/or Rac-dependent cellular functions were not fully enhanced in the OPTN knockdown cells, such as full stimulation of the cell cycle on stiff ECM or enhanced cyclin D1 transcription. Such events may strictly require force transmission to the nucleus[96].

Altogether, our transdisciplinary study establishes the critical requirement of OPTN for the coupling between integrins and cell internal tension, to adapt FA signaling to ECM stiffness and to control proper mechanosensitive cell cycle progression via regulation of HACE1 E3 ubiquitin ligase activity on Rac1. Our study also indicates that bacteria take advantage of OPTN for their integrin-mediated invasion. Silencing OPTN led to the acquisition of enhanced integrin-mediated adhesion and ECM stiffness-independent signaling and proliferation, attributes that may contribute to HACE1- and OPTN-induced tumor suppressor functions[23,59,97,98]. Nevertheless, OPTN knockdown cells displayed lower integrin-actin traction force and lower integrin-mediated bacterial uptake. More generally, this work suggests that tissue mechanical properties can influence the bacterial infection outcome and defines unexplored proteins and mechanisms.

## Methods
### Cell culture and transfection
Unless for experiments shown in Supplementary Fig. 6b, where 5637 cells (ATCC HTB-9) were used, all experiments were performed on human vein endothelial cells (HUVECs) before passage 5, that are primary cells, an important feature to keep normal ubiquitin-mediated Rac regulation. HUVEC also represent a relevant system to study host-pathogen interaction with UPEC, these pathogens frequently causing systemic infections upon dissemination from the initial infection focus through blood vasculature. HUVEC were cultured on gelatin-coated cultureware in human endothelial SFM medium (Thermo 11111044) containing 20% fetal bovine serum, 20 ng/ml FGF-2, 10 ng/ml EGF, and 1 μg/ml heparin referred to as "complete SFM". Experiments on fibronectin (BD Biosciences) were carried out with G0-synchronized cells that were plated at the density of 55,000 cells/cm² in serum-free human endothelial SFM medium containing 20 ng/ml FGF-2

(Peprotech RB4475), 10 ng/ml EGF (Peprotech 100-18C), 1 g/ml heparin (Sigma H3393) and ITS + 1 supplement (Sigma I2521), referred to as "defined medium", unless otherwise specified. G0 synchronization was achieved after 15 h of serum and growth factor starvation of confluent cultures. Cells were electroporated at 300 V and 450 μF with 10 μg of the plasmids HA-HACE1, Myc-Rac1Q61L, Myc-Rac1N17T, HA-Rac1Q61L, Myc-HACE1, Flag-OPTN, pTalin head-GFP (32856 Addgene), and pEGFP-C1 (Clontech).

For RNAi experiments, the ON-TARGETplus siRNA J-016269-05 (Dharmacon) targeting human OPTN, smart pool siRNA (Dharmacon) L-012949-00 targeting Talin1, L-009288-00-0005 targeting Vinculin, L-011927-00-0041 targeting Icap-1, L-004506-00-020 targeting integrin beta1 and SR-CL000 (Eurogentec) as a control were used at 100 nM with magnetofection technology (OZ Biosciences PN31000) following the manufacturer's instructions on cells between passage 2 and 3. Cells were used 48 h post-magnetofection. Hydrogels of different stiffnesses were purchased from Matrigen. Cells cultured on the fibronectin-coated hydrogels of varying stiffness displayed similar densities between 1 and 50 kPa and no significant differences were detected in the number of attached cells. The CNF1 toxin was prepared as described previously[99] and used at 1 nM. The pharmacological inhibitors used were EHT1864 (10 μM, Merck-Millipore E1657), blebbistatin (20 μM, Sigma-Aldrich B0560), CK-666 (250 μM, MedChemExpress HY-16926), and cycloheximide (10 nM, Sigma-Aldrich 01810) with a 2 h pre-treatment of cells when infections were performed. For pulse chase experiments, cells were treated for 90 min with 10 nM cycloheximide (CHX, Sigma 01810) before being washed twice and pulsed with complete medium for 15 min and 30 min. Cells were lysed with RIPA buffer (Biotech, RB4475) at each time point before analysis by western blots.

### Infection techniques
Because the UPEC strain UTI89 encodes the pore-forming toxin HlyA, which is highly cytolytic to many cell types in vitro[100], we worked with a genetically engineered *hlyA* chromosomal deletion. We further deleted the *cnf1* toxin-encoding gene (see *E. coli*[CNF1-] in ref. 50, here referred to as "Ec") to address its contribution to cell invasion. Complementation of the strain was realized by exogenous addition of 1 nM recombinant CNF1 prepared as described previously[99], to avoid internal variability due to unequal secretion. This concentration leads to 100% cytotoxic activity[99] and is one order of magnitude above the amount produced by bacteria cultured in the condition of our infection protocol (see below). When indicated, bacteria had further deletion of the FimH adhesin-encoding gene (EcΔFimH) using the lambda recombination technique[101] with the following oligos for mutagenesis:

Forward 5'-TGGTGGAGCCACTCAGGGAACCATTCAGGCAGTGA TTAGCATCACCTATACCTACAGCTGAACCCAAAGAGATGATTGTAGTG TAGGCTGGAGCTGCTTCG-3'

Reverse 5'-GCCTACAAAGGGCTAACGTGCAGGTTTTAGCTTCAGG
TAATATTGCGTACCTGCATTAGCAATGCCCTGTGATTTCTGGTCCATA
TGAATATCCTCCTTA-3'.

Bacterial internalization was assessed by gentamicin protection[102]. Briefly, HUVECs were seeded on 12-well plates at a density of 250,000 cells/well 24 h before the assay (triplicate wells). Exponentially growing Ec ($OD_{600}$ = 0.6 in LB) was added to cells at an MOI of 100, followed by 10 min centrifugation at $1000 \times g$. Cell infection was performed for 20 min at 37 °C and 5% $CO_2$. Infected cells were washed three times with PBS and either lysed for cell-bound bacteria measurements and western blotting or incubated for another 30 min in the presence of 50 µg/ml gentamicin before lysis for internalized bacteria measurements. Cells were lysed in RIPA buffer, and bacteria were serially diluted and plated on LB agar supplemented with 200 µg/ml streptomycin for colony-forming unit (CFU) counting. When indicated, the 5637 bladder epithelial cell line (ATCC HTB-9) was used for the gentamicin protection assay at the density of 250,000 cells/well, seeded in 12-well plates 24 h before the assay (triplicate wells).

A titration of CNF1 toxin amounts encoded by Ec during infection at MOI of 100, here corresponding to $2.5 \times 10^7$ bacteria in 1 ml medium, was assessed using the activity normalization method[99]. Exponentially growing Ec ($OD_{600}$ = 0.6 in LB) were collected by centrifugation and resuspended in 1 ml HUVEC cell culture medium prior to mechanical lysis using a French Press. Supernatants after pelleting bacteria debris were filtered on 0.2 µm, serially diluted and used for cytotoxicity assay on Hep2 cells in parallel to purified toxin for reference. In total, 96% lysis of bacteria was verified by plating the lysate before filtration for CFU determination. After 48 h, cells were washed in PBS and stained with Giemsa (GS500, Sigma-Aldrich). Multinucleated cells in response to purified toxin were enumerated and the results were expressed as percentage of the cell population to draw a standard curve. The CNF1 activity in the bacterial lysate was calculated from the % multinucleation obtained at different dilutions and interpolation from the standard curve. This indicates that Ec used for infection at MOI 100 produce 0.11 nM ± 0.01 CNF1 in average (Supplementary Fig. 1k).

For assessment of pharmacological inhibitors toxicity on Ec, bacteria were diluted from a fresh overnight culture to obtain an initial $OD_{600}$ = 0.01 in LB supplemented with the chemical inhibitors at the indicated concentrations. Growth was performed at 37 °C, 120 rpm orbital shaking in a in 96-well microplate reader, with a volume of 100 µl per well and in quadruplicates. Absorbance at 600 nm was measured every 10 min to generate growth curves.

**Quantitative proteomics**

LC-MS/MS sample preparation: for label-free quantitative proteomic analysis of stiffness-related proteins, HUVECs ($0.5 \times 10^6$) were seeded on 35 mm Matrigen hydrogels of different stiffnesses after fibronectin coating for 2.5 h at 37 °C. Four independent biological replicates were prepared and analyzed for each condition. After overnight seeding, complete medium was changed, and CNF1 at 1 nM was added for 2 h where indicated. Cells were washed with PBS and lysed in urea lysis buffer (8 M urea, 20 mM HEPES, pH 8, filtered) supplemented with a cocktail of protease (A32961 Pierce) and phosphatase inhibitors (PhosStop Roche). LC-MS/MS and data analysis: purified peptides for shotgun analysis were re-dissolved in 20 µl solvent A (0.1% TFA in water/ACN (98:2, v/v) and peptide concentration was determined on a Lunatic spectrophotometer (Unchained Labs). In total, 2 µg of each sample was injected for LC-MS/MS analysis on an Ultimate 3000 RSLCnano system in-line connected to an Orbitrap Fusion Lumos mass spectrometer (Thermo) equipped with a pneu-Nimbus dual ion source (Phoenix S&T). Trapping was performed at 10 µl/min for 4 min in solvent A on a 20 mm trapping column (made in-house, 100 µm internal diameter (I.D.), 5 µm beads, C18 Reprosil-HD, Dr. Maisch, Germany) and the sample was loaded on a 200 cm long micropillar array column (PharmaFluidics) with C18-endcapped functionality mounted in the

Ultimate 3000's column oven at 50 °C. For proper ionization, a fused silica PicoTip emitter (10 µm inner diameter) (New Objective) was connected to the µPAC™ outlet union and a grounded connection was provided to this union. Peptides were eluted by a non-linear increase from 1 to 55% MS solvent B (0.1% FA in water/ACN (2:8, v/v)) over 175 min, first at a flow rate of 750 ml/min, then at 300 ml/min, followed by a 10-min wash reaching 99% MS solvent B and re-equilibration with MS solvent A (0.1% FA in water). The mass spectrometer was operated in data-dependent mode, automatically switching between MS and MS/MS acquisition. Full-scan MS spectra (300–1500 $m/z$) were acquired in 3 s acquisition cycles at a resolution of 120,000 in the Orbitrap analyzer after accumulation to a target AGC value of 200,000 with a maximum injection time of 250 ms. The precursor ions were filtered for charge states (2–7 required), dynamic range (60 s; ±10 ppm window) and intensity (minimal intensity of 3E4). The precursor ions were selected in the multipole with an isolation window of 1.2 Da and accumulated to an AGC target of 1.2E4 or a maximum injection time of 40 ms and activated using HCD fragmentation (34% NCE). The fragments were analyzed in the Ion Trap Analyzer at normal scan rate. MaxQuant search: LC-MS/MS runs of all 24 samples were searched together using MaxQuant[103] (version 1.6.11.0) with mainly default search settings, including a false discovery rate (FDR) set at 1% on peptide-to-spectrum matches, peptide and protein level. Spectra were searched against the human protein sequences in the Swiss-Prot database (database release version of 2019_06), containing 20,960 sequences (www.uniprot.org). The mass tolerance for precursor and fragment ions was set to 4.5 and 20 ppm, respectively, during the main search. Enzyme specificity was set as C-terminal to arginine and lysine, also allowing cleavage at proline bonds with a maximum of two missed cleavages. Variable modifications were set to oxidation of methionine residues and acetylation of protein N-termini, while carbamidomethylation of cysteine residues was set as fixed modification. Matching between runs was enabled with a matching time window of 0.7 min and an alignment time window of 20 min. Only proteins with at least one unique or razor peptide were retained leading to the identification of 3515 proteins. Proteins were quantified by the MaxLFQ algorithm integrated in the MaxQuant software. A minimum ratio count of two unique or razor peptides was required for quantification. Further data analysis of the shotgun results was performed with an in-house R script using the proteinGroups output table from MaxQuant. Reverse database hits were removed, LFQ intensities were log2 transformed, and replicate samples were grouped. Proteins with less than three valid values in at least one group were removed, and missing values were imputed from a normal distribution centered around the detection limit (package DEP[104]), leading to a list of 1804 quantified proteins in the experiment that were used for further data analysis. For comparison of protein abundance between pairs of sample groups (4 kPa vs. 1 kPa, 50 kPa vs. 1 kPa and 50 kPa vs. 4 kPa within the NT and CNF1 sample groups), statistical testing for differences between two group means was performed using the limma package[105]. Statistical significance for differential regulation was set to a FDR of <0.05 and fold change of >2- or <0.5-fold (|log2FC| = 1). The results are presented in Supplementary Data 1 and 2. Z-scored LFQ intensities from significantly regulated proteins were plotted in a heatmap after non-supervised hierarchical clustering (results listed in Supplementary Data 3). Functional enrichment analysis and network: functional enrichment analysis was performed using DAVID[106] on a list of proteins with significantly upregulated expression under increased stiffness conditions, and a background list of all the proteins quantified in the dataset was used in the analysis. A 0.1 EASE score cut-off allowed us to identify a group of enriched UP_KEYWORDS and KEGG pathways. The results from non-CNF1 toxin-treated cells (NT) and CNF1-treated cells (CNF1) were analyzed separately. A network of associations between the HACE1 protein interactome (retrieved from the IntAct database[107]) and the five sets of proteins belonging to common

enriched UP_KEYWORDS in the NT and the CNF1 analysis was created with Cytoscape (version 3.9.1)[106].

## Cell biology techniques

Immunofluorescence was performed on paraformaldehyde-fixed cells permeabilized in 0.5% Triton X-100. FITC-phalloidin at 1 µg/ml (Sigma, P5282) was used to stain actin, and DAPI (Life Technologies, D1306) was used to stain nuclei. The adhesive structures were stained with antibodies against paxillin (349, BD-Transduction Laboratories, 1/200), zyxin (ZZ001, Invitrogen, 1/100), and integrin alpha 5 (AB1928, Chemicon, 1/400). Images were acquired on a Nikon A1R confocal laser microscope using a ×60 oil objective and NIS Elements software (version 4.50.00). QMPIA of immunofluorescence images was performed in two sequential rounds of calculations by the automated image analysis software Motion Tracking (version 8.80.23 ×64), described previously[108]. Briefly, in the first round, nuclei were found by maximum entropy-based local thresholding and cells by a region-growing algorithm based on the watershed transform. Additionally, aiming at the identification of fluorescent structures, image intensity was fitted by a sum of powered Lorenzian functions. The coefficients of those functions were then used to describe the features of individual objects (e.g., intensity and area). In the second round, a set of statistics was extracted from the distributions of the fluorescent structure parameters measured in the first round. From this automated pipeline of analysis, the numbers and areas of each fluorescent structure were extracted per cell, and the mean values per image were obtained. The mean of the control cells for each parameter was normalized to 1, allowing a fold representation of the variations of knockdown cells vs. control cells. Graphs represent the normalized mean ± standard error to the mean.

DNA content was determined upon staining ethanol-fixed cells with 40 µg/ml propidium iodide (Sigma, 81845) in 0.1% NP-40 (Thermo Scientific, 11596671) and analysis on a BD FACSAria flow cytometer.

For adhesion assays, cells were detached and replated in triplicate in 96-well microtiter plates coated with fibronectin for periods of 7–15 min. Nonadherent cells were removed by inverting plates, and attached cells were fixed and stained with crystal violet. The extent of adhesion to the substrate was determined as a function of absorbance upon elution of the dye and reading of the optical density (540 nm). The blank value corresponding to BSA-coated wells was systematically subtracted. To measure cell area, pictures were taken from the three replicate wells with an inverted microscope at ×5 magnification and analyzed with the ImageJ software (version 2.3.1) using the particle analysis tool on FIJI. A threshold has been set and a mask created with a minimum size of 0.02 µm² and circularity 0.00–1.00 criteria. Cells from the edges were excluded from the analysis.

## Biochemical techniques

For OPTN-HACE1 coimmunoprecipitations, HUVECs ($10^7$ cells) were transfected using either 40 µg of pCMV2-Flag-OPTN WT or empty vector and 40 µg of pKH3-HA$_3$-Hace1. Cells were lysed in 1 ml of RIPA buffer (25 mM Tris-HCl, pH 7.6, 150 mM NaCl, 1% NP-40, 1% sodium deoxycholate, 0.1% SDS). Cleared lysates were incubated for 2 h at 4 °C with 30 µl of precleared slurry EZ view Red Anti-FLAG M2 Affinity Gel beads (Sigma, A2220). Beads were washed three times with RIPA buffer and resuspended in 50 µl of Laemmli buffer. Proteins were resolved on 4–12% SDS-PAGE. In total, 0.05% of the total lysate was mixed with Laemmli buffer and loaded (INPUT) to check expression levels. For Rac1-Hace1 associations, HUVEC cells were transfected with control or OPTN SiRNA by magnetofection and further transfected after 24 h by 30 µg of pRK5-myc-Rac1Q61L or empty vector and 40 µg of pKH3-HA$_3$-Hace1. Lysis and immunoprecipitation were done as above using EZ view Red Anti-c-Myc Affinity Gel (Sigma, E6654). For metabolic labeling, cells were pulsed with L-azidohomoalanine (AHA, 50 µM, 1 h). Cell lysates were subjected to a Click-iT reaction (Molecular Probes) with

biotin alkyne followed by streptavidin pulldown following the manufacturer's instructions, and newly synthetized proteins were detected by western blots.

For western blots, the primary antibodies used were anti-HA clone 16B12 (Covance, 1/2000), anti-c-myc clone 9E10 (Roche, 1/1000) anti-Flag clone M2 (Sigma, 1/1000), anti-Rac clone 102/Rac (BD Transduction Laboratories, 1/1000), anti-Flotilin clone 18/Flotillin-1 (BD Transduction Laboratories, 1/5000), anti-Transferrin Receptor clone H68.4 (Zymed, 1/1000), anti-Src clone L4A1 (Cell Signaling Technologies, 1/1000), anti-Flotilin clone 18/Flotillin-1 (BD Transduction Laboratories, 1/5000), anti-FAK clone 77/FAK (BD Transduction Laboratories, 1/1000), anti-OPTN (Abcam 23666; Santa Cruz clone C-1, 1/500), anti-cyclin D1 clone M-20 (Santa Cruz 1/500), anti-p-FAK-Y397 clone 18/FAK(pY397) (BD Transduction Laboratories 1/500), anti-p-Src-Y416 clone D49G4 (Cell Signaling Technologies, 1/500), anti-Paxillin (BD Transduction Laboratories, 1/1000), anti-p-Paxillin-Y118 (Invitrogen, 1/500), anti-p-P130CAS-Y410 (Cell Signaling Technologies, 1/500), anti-p130CAS clone 35B.1A4 (Santa Cruz, 1/1000), anti-GAPDH clone 0411 (Santa Cruz, 1/5000), anti-p-MLC Y18/S19 (Cell Signaling Technologies, 1/500), Beta1 integrin clone 18/CD29 (BD Transduction Laboratories, 1/1000), ICAP-1 (ITGBP1) (Atlas Antibodies, 1/500), Vinculin clone hVIN1 (Sigma, 1/500), Talin clone TA205 (Millipore, 1/500), HACE-1 (Custom-made, clone 9D3, 1/500), GFP clones 7.1 and 13.1 (Roche, 1/1000).

The secondary antibodies used were HRP-conjugated anti-mouse or anti-rabbit (Dako, 1/5000). Signals were imaged using the Fujifilm LAS-3000 system and quantified with ImageJ software (version 2.3.1). For immunoaffinity precipitations: EZ view Red Anti-FLAG M2 Affinity Gel (Sigma, F2426) and EZ view Red Anti-c-Myc Affinity Gel (Sigma, E6654). For immunofluorescence: paxillin clone 349 (BD-Transduction Laboratories, 1/200), zyxin clone ZZ001 (Invitrogen 1/100), and integrin alpha 5 (Millipore, 1/400).

An original uncropped version of each western blot is available in the source Data file.

For in vivo ubiquitylation measurements, HUVECs ($5 × 10^6$) were transfected with 5 µg of His-tagged ubiquitin expression vector together with 5 µg of HA-tagged Rac1 Q61L and Myc-tagged HACE1 WT constructs. Ubiquitylated proteins were purified by His-tag affinity purification on cobalt resin under denaturing conditions, as described previously[99].

Isopycnic density-gradient centrifugation was performed upon lysis for 30 min at 4 °C of $7 × 10^6$ cells in 150 µl of lysis buffer containing 1% Triton X-100, 150 mM NaCl, and 25 mM Tris, pH 7.4. For each gradient, we used 120 µl of cell lysate mixed with 240 µl of 60% OptiPrep™ (Sigma, D1556) prepared in 25 mM Tris pH 7.4 and 150 mM NaCl to yield 40% sucrose. Then, 300 µl of the prepared mixture was sequentially overlaid with 600 µl of 30% sucrose in 25 mM Tris and 150 mM pH 7.4 NaCl followed by 300 µl of 25 mM Tris buffer and 150 mM pH 7.4 NaCl. The tubes were centrifuged at 100,000 × g for 2 h at 4 °C (S55S rotor, Sorvall-RC-M120GX). Eight fractions of 150 µl were collected from the top of the tube and resolved by SDS-PAGE. All steps were performed at 4 °C.

Rac-GTP affinity pulldown was performed as described previously[99].

An in vivo F/G actin biochemical assay was performed using the Cytoskeleton kit BK037 following the manufacturer's instructions. Briefly, an equal number of cells were lysed in LAS2 buffer, and F-actin and G-actin fractions were extracted after ultracentrifugation at 100,000 × g. Samples were resolved by SDS-PAGE, and the levels of F- and G-actin were quantified by western blotting using the anti-actin antibody provided in the kit. The ratio of F-actin to F + G-actin was determined by densitometry.

For qRT–PCR, RNA was extracted from cultured HUVECs using an RNeasy kit (Qiagen, 74104). Reverse transcription was performed on 1 µg of total RNA using a high-capacity c-DNA Reverse transcription kit (Applied Biosystems, 10400745) according to the manufacturer's

instructions. The following primers were used for amplification on a StepOne Real Time PCR system (Applied Biosystems) (OPTN: forward CAAGCCATGAAAGGGAGATTTGA, reverse GCCATTAGACGCTCTTTT GCTTC; cyclin D1: forward TGTTTGCAAGCAGGACTTTG, reverse TCATCCTGGCAATGTGAGAA; 36B4: forward TGCATCAGTACCCC ATTCTATCAT, reverse AAGGTGTAATCCGTCTCCACAGA).

## Biophysical techniques

**Traction force measurements using the parallel microplate setup.** Traction force measurements with the parallel plate setup are similar in their principle to AFM force measurements, which are based on the defection of a cantilever of known stiffness. The difference here is that the mechanical setup is coupled with bright-field monitoring of cell shape evolution during force generation[109]. In brief, a single-cell is caught between two parallel plates, one rigid and the other flexible, with a calibrated stiffness $k$. The traction force **F** is measured through the deflection $d$ of the flexible plate: $\mathbf{F} = k\,d$[79]. Moreover, we developed an original real-time stiffness-control feedback loop, allowing us to set the apparent stiffness felt by the cell from 0 to an infinite value[80]. All experiments presented here were performed under these infinite stiffness conditions to ensure that we truly measured the maximum traction force that a single-cell could generate in the parallel plate geometry.

Before experiments, glass plates were pulled using a micropipette puller (PB-7; Narishige) and shaped with a microforge (MF-900; Narishige). Then, the stiffness k of the flexible plates is calibrated[109]. During the experiments, the flexible glass plate is illuminated through bright light, and its deflection $d$ is optically detected in real time by a position-sensitive detector (Hamamatsu) used in inverted-contrast mode[109]. Technically, glass microplates were cleaned for 10 min in a "piranha" mixture (67% sulfuric acid + 33% hydrogen peroxide), rinsed in water, dipped in a (90% ethanol + 8% water + 2% 3-aminopropyltriethoxysilane) bath for 2 h, and then rinsed in ethanol and water. Finally, microplates were coated with 5 µg/ml fibronectin (Sigma). Cells were trypsinized and suspended in defined medium. First, rigid and flexible microplates were placed near the bottom of the manipulation chamber. Then, the chamber was filled with suspended cells, and we waited until deposition of cells on the chamber's bottom. The microplates were then lowered toward the chamber's bottom and placed in contact with a cell. After a few seconds, the two microplates were simultaneously and smoothly lifted to 60 µm from the chamber's bottom to obtain the desired configuration of one cell adherent between two parallel plates. Cells spreading between the microplates were visualized under bright light illumination with a Plan Fluotar L ×63/0.70 objective and a Micromax digital CCD camera (Princeton Instruments, Roper Scientific). The setup, enclosed in a Plexiglas box, was maintained at 37 ± 0.2 °C by an Air-Therm heater controller (World Precision Instruments). Vibration isolation was achieved by a TS-150 active anti-vibration table (HWL Scientific Instruments Gmbh).

## Traction force measurements on polyacrylamide hydrogels

Carboxylate-modified polystyrene fluorescent beads (dark red 200 nm, Invitrogen F-8807) were sonicated for 3 min and then mixed with a 5 kPa polyacrylamide solution prior to adding ammonium persulfate and N,N,N0,N0-tetramethylethylenediamine (TEMED). The gel was poured and allowed to polymerize in 25 mm glass coverslips; one was previously silanized, and the other was coated with fibronectin (Sigma) and Alexa546-conjugated fibrinogen (Invitrogen), as described previously[110]. During polymerization, the hydrogel adheres to the silanized coverslip, and fibronectin proteins are trapped on top of the acrylamide mash. Coverslips were gently separated in deionized water, and a 5 kPa fibronectin-coated gel was retrieved with a silanized coverslip and used after two washes in PBS. Cells were seeded at low density in defined medium and imaged after 2 h of adhesion. Beads

and cells were then imaged using a ×63 oil objective lens (numerical aperture 1.4) combined with a 1.5 optical multiplier on a Nikon Ti-E microscope with a CCD camera (Coolsnap, Roper Scientific) and driven with µManager. While imaging, cells were kept at 37 °C. Cells were then gently trypsinized, and a second set of bead images was performed at the same coordinates. Force calculations were performed as described previously[81]: bead displacement was measured through a 2-step cross-correlation between bead images of substrate with and without adherent cells on top. Pictures corrected for experimental drift were divided into 6.72 µm square windows to yield the average displacement and then processed with a regular grid of 0.84 µm spacing using linear interpolation to achieve high spatial resolution. Force reconstruction was conducted with the assumption that the substrate is a linear elastic half space using Fourier transform traction cytometry with zero-order regularization[82]. All calculations and image processing were performed using MATLAB software (version R2019a).

## Traction force measurements on micropillar substrates

Micropillar substrates were prepared using previously published methods[83]. Briefly, the elastomer polydimethyldiloxane (PDMS, Sylgard®184 Silicone Elastomer Kit, Dow Corning) was cast using patterned silicon wafers as molds. Uncured PDMS (with a base-polymer-to-cross-linker ratio of 10:1) was cast by pouring it on silicon wafers and then baking them at 80 °C for 2 h to cure the PDMS. The cured PDMS had a Young's modulus of ~2 MPa. The micropillars had a diameter of 2 µm and a height of 9 µm, resulting in a stiffness of 9 nN/µm. The stiffness was calculated using the finite element method. After decasting the PDMS micropillar substrates from the silicon wafers, microcontact printing was used to selectively deposit fibronectin (conjugated with the fluorescent far-red dye ATTO647N) on micropillar tops. The substrates were immersed in Pluronic®F-127 solution for 1 h to prevent cell adhesion and migration between micropillars. The substrates were then rinsed with PBS and used for cell culture. Cells were allowed to adhere for 3–4 h before imaging. Force ($F$) on a micropillar was calculated as $F = k\,x$, where $k$ is the micropillar stiffness and $x$ is the micropillar displacement. The strain energy ($E$) stored in a micropillar was calculated as $E = 0.5\,k\,x^2$.

## Atomic force microscopy

Cells transfected with siRNA were either treated with CNF1 toxin for 2 h or detached and plated for 1 h 30 on 35 mm WilcoWell glass bottom dishes coated with fibronectin. AFM experiments were performed on a JPK NanoWizard III mounted on a Zeiss Observer Z1 optical microscope with temperature control. Before each experiment, cantilevers (precalibrated PFQNM-LC-A-CAL, Bruker) were calibrated according to the SNAP procedure[111] to eliminate any error in the deflection sensitivity determination. Each experiment was carried out on the same day, with the same cell culture and the same cantilever to account for slight variation in the tip shape manufacture, and was repeated at least three times. Cells were scanned in the perinuclear region in force distance mode (QI) with the following parameters: 300 pN force threshold and 50 µm/s tip velocity on a 40✕40 (Fig. 5) or 20✕20 (Fig. 6), 6 µm² surface. Cortical elasticity (Young's modulus) was computed using in-house Python software (version 3.6.9) by fitting the first 50 nm of cell indentation with the Hertz model for a sphere of 65 nm radius.

## Data visualization and comparison of distributions

Distributions were visualized using boxplots with the features indicated below. The middle line indicate the median, the cross represents the mean. Boxes show the 25th and 75th percentile, while whiskers indicate the 5th and 95th percentile. Outliers are represented by single dots. Bars displaying mean, SD and individual values (single dots) were used for western blot quantification.

## Statistical analysis

Statistical tests were performed using GraphPad Software Prism 9.0. All experiments were repeated to ensure reproducibility. $p$ values were calculated using Wilcoxon rank sum test, or one-way or two-way ANOVA with corrections for multiple comparisons as indicated in the figure legends. Significance levels are indicated as follows: ns, not significant $p > 0.05$, $*p \le 0.05$, $**p \le 0.01$, $***p \le 0.001$, $****p \le 0.0001$. Exact $p$ values, when possible, are provided in the figure legends.

## Reporting summary

Further information on research design is available in the Nature Research Reporting Summary linked to this article.

## Data availability

All data generated or analyzed during this study are included in this published article and its Supplementary Information files. The mass spectrometry proteomics data have been deposited to the ProteomeXchange Consortium via the PRIDE partner repository with the dataset identifier PXD035396. Source data used to generate the graphs are provided in the Supplementary Information/Source Data file. GMO can be obtained from the corresponding authors. Source data are provided with this paper.

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

## Acknowledgements

This study was supported by institutional INSERM, CNRS and Institut Pasteur fundings, "Investments for the Future" LabEx SIGNALIFE ANR-11-LABX-0028-01 and grants from the Ligue Nationale contre le Cancer (LNCC labellisation and RS20/75-63), Fondation ARC pour la Recherche contre le Cancer (ARC PJA20191209650), the French National Research Agency (ANR-21-CE15-0006), EPIC-XS, project number 823839, funded by the Horizon 2020 program of the European Union (PRC-5074) and FRM-Piraud prize to E.L., Bio-SPC "Contrat Doctoral" PhD fellowship to S.P., LNCC PhD fellowship to D.H. Fundings from the Vancouver Prostate Centre to M.D., and Team Finn and The Ride to Concur Cancer to P.H.S., fundings from ANR to A.A. ("ImmunoMeca" ANR-12-BSV5-0007-01, "Initiatives d'excellence" Idex ANR-11-IDEX-0005-02, and "Labex Who Am I?" ANR-11-LABX-0071) and to F.L. (ANR-10-EQPX-04-01 and FEDER 12001407). B.L. acknowledges financial supports from the Mechanobiology Institute, the European Research Council under the European Union's 7th Framework Program (FP7/2007-2013)/ERC no. 617233 and NUS-USPC collaborative program. The authors would like to thank Yannis Kalaidzidis from the laboratory of Marino Zerial (MPI-CBG, Dresden, Germany) for the free access to Motion tracking software; P. Tafelmeyer (Hybrigenics, France), O. Visvikis, P. Munro (INSERM U1065) and A. Marabelle (INSERM U1015) for helpful advices; P. A. Roldan-Quiros (ITCR, Costa Rica), A. Loubat at the iBV-Cell sorting Facility and D. Van Haver from the VIBS Proteomics Core (UGent Dept of Biomolecular Medicine, Gent, Belgium) for their technical support. We acknowledge the GIS-IBISA multi-sites platform "Microscopy Imagerie Côte d'Azur" (MICA), and particularly the imaging site of C3M (INSERM U1065) supported by "Conseil Régional", "Conseil Départemental", and IBISA.

## Author contributions

A.M. and E.L. Conceived the project, designed and supervised research, provided main fundings and wrote the original draft; S.P., A.D., A.M. and D.H. principally conducted experiments and analyzed the data in cell biology, biochemistry and infection with the help of L.G.P. and J.G.; A.A., F.F., E.V., S.J. and M.G. principally conducted experiments and A.A., M.B., F.L. and B.L. supervised experiments and analyzed the data in biophysics approaches; T.M.M. and F.I. performed and analyzed the quantitative proteomics experiments; M.D and P.H.S. provided guidance and reagents; S.P., A.D., D.H. and A.M. performed data presentation/visualization. All authors reviewed and approved the manuscript.

## Competing interests

The authors declare no competing interests.

## Additional information

[1]Institut Pasteur, Université Paris Cité, CNRS UMR6047, INSERM U1306, Unité des Toxines Bactériennes, F-75015 Paris, France. [2]Université Côte d'Azur, INSERM, C3M, Team Microbial Toxins in Host-Pathogen Interactions, Nice, France. [3]Equipe Labellisée Ligue Contre le Cancer, Nice, France. [4]Université Paris Cité, CNRS, Laboratoire Matière et Systèmes Complexes, UMR7057, F-75013 Paris, France. [5]Université Grenoble Alpes, CNRS, LiPhy, F-38000 Grenoble, France. [6]Université de Lille, CNRS, INSERM, CHU Lille, Institut Pasteur de Lille, U1019-UMR9017, CIIL—Center for Infection and Immunity of Lille, F-59000 Lille, France. [7]Université Paris Cité, CNRS, Institut Jacques Monod, F-75013 Paris, France. [8]Université Côte d'Azur, INSERM, C3M, Team Cellular and Molecular Pathophysiology of Obesity and Diabetes, Nice, France. [9]VIB-UGent Center for Medical Biotechnology, VIB, Ghent, Belgium. [10]Department of Biomolecular Medicine, Ghent University, Ghent, Belgium. [11]VIB Proteomics Core, VIB, Ghent, Belgium. [12]Université Côte d'Azur, CNRS, INSERM, Institut de Biologie Valrose (iBV), 06108 Nice, France. [13]Vancouver Prostate Centre, Vancouver, BC V6H 3Z6, Canada. [14]Department of Urologic Sciences, University of British Columbia, Vancouver, BC, Canada. [15]Department of Molecular Oncology, BC Cancer Research Center, University of British Columbia, Vancouver, BC V5Z1L3, Canada. [16]These authors contributed equally: Serena Petracchini, Daniel Hamaoui. [17]These authors jointly supervised this work: Emmanuel Lemichez, Amel Mettouchi. ✉e-mail: emmanuel.lemichez@pasteur.fr; amel.mettouchi@pasteur.fr

