## [Peer review file · Nature Communications]

REVIEWER COMMENTS

Reviewer #1 (Remarks to the Author):

Manuscript Summary:

In this manuscript the authors identify a substrate stiffness-modulated role of OPTN in regulating the ligase activity of HACE1 on Rac1. They show, that the invasion of *E. coli* within HUVEC host cells is dependent on this mechanism. Overall, I think this work is interesting, but I have major concerns about data presentation and analysis, few overstatements and missing technical information.

Major/General Comments:

(1) Nowhere is it stated how many HUVEC cells were seeded. Confluence is not discussed anywhere even though it is stated that proliferation rate changes due cyclin D1 dependency. Confluence can vary significantly depending on matrix stiffness especially given that many cell types (including HUVEC) are not very keen on attaching on very soft matrices. How do authors control for that? Confluence can have a dramatic effect on cell behavior and needs to be at least documented. If anything, the authors appear to seed cells subconfluently so that single cells or cell clusters are analyzed (in the images provided). How relevant is that considering that in blood vessels endothelial cells form a continuous single monolayer? Why did they choose to seed cell sparsely? And why were HUVEC (primary umbilical vein endothelial cells) chosen to conduct this type experiments?

(2) In Figure 5F how is it possible to make hydrogels of 0.2 kPa (this should be almost acting as fluid). It is very hard to believe that HUVEC could attach on such a soft and highly porous material. Did you confirm that the stiffnesses of the gels are really what you claim through AFM? Given the centrality of ECM stiffness on this paper and given that authors performed AFM on their cells, I consider very important to at least validate once that assumed ECM stiffness is correct. Also, polyacrylamide hydrogels are notorious to show some deviation from gel to gel and batch to batch on their expected stiffness. That also is important to know through AFM measurements on the actual gels. Also, in different figures (experiments) different combinations of matrix stiffnesses are used. Why is that?

(3) In the last section of the paper authors state that "OPTN controls cell contractility". This is a very strong statement that authors could more precisely address by comparing how siOPTN cells act on high and low stiffness matrices as compared to OPTN expressing cells. If OPTN indeed controls cell

contractility, then by doing TFM on varying stiffness hydrogels one should see a gradual increase in forces for WT HUVEC cells. Moreover, the changes should vanish (if they are matrix stiffness dependent and OPTN indeed controls them) in siOPTN cells. I believe this is a very important experiment missing.

(4) The generation of primary HUVEC cells is a very important piece of information that is also missing. What generations were used? Often generation >P8 result in HUVEC with a pancake morphology which is reminiscent of some images seen e.g., at Figure 7 (as opposed to spindled like morphology seen in younger generations P2-P8).

(5) Why were AFM measurements performed with a tiny tip and not both a larger spherical tip that would allow to get a better idea of bulk stiffness properties. Perhaps also the large deviation of the AFM data from the mean is due to the usage of a small tip.

(6) There is missing information in the figure legends. What are the data presented? How many independent experiments were performed? What is the very basic experimental structure? Also, formatting is inconsistent in text and figures.

Specific comments:

(1) In figure 1A: At what density were HUVEC (by cell I assume HUVEC cell) seeded? What generation?

(2) In figure 1B-C: Why is standard error of the mean shown and not standard deviation? Why are not individual values shown on top of boxplots? Why are controls without error bar?

(3) In Figure 2D: How many independent experiments were performed?

(4) In Figure 3A: Stiffness of the matrix not indicated in text/figure legend after claiming OPTN levels are very stiffness dependent. Also why is here no loading control but in other figures with 'input' there is?

(5) Figure 3B: Authors state 'We found that OPTN knockdown led to a reduction in Rac1Q61L ubiquitylation under conditions of endogenous or overexpressed HACE1'. Authors probably refer to siOPTN + conditions with either Myc-HACE1 + or - . But there is also no big difference between siOPTN + / myc-HACE1 + and siOPTN - / myc-HACE1 -. So, if siOPTN + / myc-HACE1 + is considered a reduction in

ubiquitylation then in cells without a siOPTN knockdown but endogenous levels of HACE1 the same reduction takes place? So, it is not OPTN dependent in that case?

(6) Figure 3C: Why did authors chose different stiffnesses than the ones tested for OPTN expression?

(7) Figure 3D: Why is here no loading control in the input? It would be nice to see the results on a stiff matrix (e.g., 25kPa) to directly see if overexpression of OPTN could abolish stiffness dependent effects.

(8) Figure 3G: What is in fractions 1,4,5?

(9) Figure 4A: Here sync is used as annotation in the figure, but only explained in the figure legend referring to 4C. Also, why do authors again change the matrix stiffnesses tested (as compared to the previous figure)? In the previous (Figure 3C) they used 1,4 and 12 kPa.

Authors claim that “the fraction of proliferating cells doubles between control cells on high and low stiffness matrix”. This is clearly not true, for 1 kPa is at around 27%, 8 kPa is at around 47%. Also, authors state: ‘ Interestingly, the OPTN knockdown cells reached maximal proliferation when plated on low elastic modulus ECM.’ This is also not quite true. Their max is still at 8kPa even though the fraction of proliferating cells greatly increases compared to the control group (also if there is no significant difference for 1 and 8 kPa for the knockdown that is worth being denoted in the graph).

(10) In Figure 4A and B and elsewhere what is “-“ kPa?

(11) In Figure 4C: What matrix stiffness was used here? It would have been nice to see soft/stiff conditions compared again, as that was the point in the up and down regulation of cyclin D1 dependency of OPTN.

(12) In Figure 5: In the text the font size changes all the time which is not nice to read.

(13) In Figure 5D as compared to Figure 6C the stiffness of the cotrol cells is different. Why is that? It looks like this difference is similar to the different between siCntr and siOPTN which is significant. Why SEM and not SD? What are the authors thoughts on the high variance? Does it have to do with where the cells were poked (nucleus versus cell edge). Why are the authors not providing the stiffness maps of cells since they probed stiffness by scanning? Also, what matrix stiffness was used?

(14) Figure 5F: Here there is yet a new set of tested matrix stiffnesses. 0.2 kPa has not been tested before (only in one supplemental figure panel as a control) and now 4 kPa was added. Also 0.2 kPa is in essence liquid. I don't think that cells could attach there. It would be a nice proof to include images (e.g., phase contrast) of how cells look at different matrix stiffnesses reported here and also of course to validate that the stiffnesses are the ones that authors claim.

(15) Figure 6: Is only one stiffness use here and which? Also in panel C why is the scanning size different than in the previous figure (40x40 versus 20x20)? In panel D why are there no error bars for the negative controls?

(15) Figure 7A: What does it mean traction force curves? Is that the maximum force? Or is it the mean absolute force along the cell area? How was segmentation done to define cell borders or was that not done? Why just a single cell is shown when the average of many cells (and SD) could be shown? Circles in the legends are almost invisible.

(16) Figure 7B-C: Where is the standard deviation. Also, how many independent experiments were done?

(17) Figure 7D: It would have been highly relevant to compare how siOPTN cells act on high and low stiffness as compared to OPTN expressing cells. If OPTN indeed controls cell contractility, then by doing TFM on varying stiffness hydrogels one should see a gradual increase in forces. Also, the changes should vanish (if they are matrix stiffness dependent and OPTN indeed controls them) in siOPTN cells. Also, on the left panels authors show magnitude of traction stresses. On the right is this a phase contrast image? Why/how is the phase contrast differently adjusted for cell versus its surrounding environment?

(18) Figure 7E-F: Authors do not state which statistical methods they used to compare forces, energy etc. That is a general comment. Also, when force vectors are added up to compare the total forces, did they take the total size of the cell into consideration to account for morphological changes in area or not?

Reviewer #2 (Remarks to the Author):

This work builds on previous studies indicating that activation of Rac1 by the UPEC-associated toxin CNF1 can facilitate bacterial entry into host cells. Host cells can counter aberrant Rac1 activation by CNF1 through HACE1-mediated ubiquitination and subsequent proteosomal degradation of activated

Rac1. Petracchini et al. dig deeper into this system, showing that CNF1 activities augment FimH-mediated invasion of host cells by UPEC via integrin receptors while also providing new insight into the regulatory control of Rac1 and cytoskeletal tension/dynamics. More specifically, they show that OPTN can regulate HACE1 activity, thereby impacting the levels of active Rac1, in response to external ECM stiffness. The paper is dense with information and utilizes a great set of diverse experimental approaches with well-written text that should help make the work accessible to a wide range of individuals with varied backgrounds. Many pathogens, in addition to UPEC, hijack integrin receptors. This, and clear links to cell cycle progression and a number of other key cellular pathways, will make this study of great interest to a broad swath of scientists. There are a few mostly minor issues that should be clarified, but overall enthusiasm for this well-thought-out paper is high.

Bigger Issues:

- This study uses the HUVEC cell line. Some explanation for this choice should be provided, as most studies that examine UPEC invasion employ bladder epithelial cells. The specific cell line used should be made clear at the start of the Results section. It would be useful to the UTI field to know if some of the key findings were also true in bladder cells. For example, would silencing OPTN expression also inhibit UPEC entry into bladder epithelial cells?
- Do any of the inhibitors used in the paper (e.g. CK-666 or blebbistatin) affect growth or viability of UTI89 at the concentrations used?
- CNF1 is used at a concentration of 1 nM. Can the authors comment on the physiological relevance of this concentration? Is this level seen during actual infections?
- The study uses MOIs of 100, which is substantially higher than is typically used in the field. Why were so many bacteria used? Are there dosage effects with the HUVECs? This is not necessarily a major issue, but it would be good to understand why the high MOI was chosen.

Minor issues:

- The Summary could do a better job of highlighting the key major findings of the paper more directly. Lines 57-59 seem somewhat redundant with the final sentence of the Summary.

- Line 75 – it is not entirely clear how 'critical' FimH is for UPEC persistence within the gut, as at least one paper noted that FimH is dispensable in some situations (doi: 10.1128/IAI.00746-17)
- Line 102 and line 387 should have references.
- Line 212 – use of 4 kPa as an “intermediate” stiffness value seems strange in considering the lower (1 kPa) and higher “50 kPa) values used in the paper. Why not use 25 kPa, or was 4 kPa used in anticipation of non-linear effects?
- The font size used for Fig 2C makes the text very hard to read, even with zooming. The CNF1 nodes (yellow with orange edges) could be made more visible to help readers.

Reviewer #3 (Remarks to the Author):

The manuscript by Petracchini and colleagues explores the role of ECM stiffness during bacterial invasion. The authors identify a protein, OPTN, which is upregulated by stiffness. OPTN has been previously shown to interact with the Ubiquitin ligase HACE1, which regulates the ubiquitylation and degradation of active Rac1. Silencing OPTN promotes Rac1 activation and cells that show more and larger focal adhesions, but surprisingly a decreased traction force.

The manuscript tries to connect the bacterial toxin CNF1 with Rac1 activation, enhanced stiffness with increased invasion, and OPTN mediated Rac1 inactivation to mechanotransduction. The individual observations presented in this manuscript are interesting and suggest a concerted mechanism connecting CNF1 to mechanotransduction to regulation of Rac1 activity. However, in my opinion, these connections are not properly validated.

The individual claims are well supported by the results (e.g. OPTN expression is regulated by cellular stiffness). However, the connections between the independent results are not clearly supported. For example, a causal link between Rac1 activation and the phenotypes observed is not demonstrated in this article. It is not even tested that some of the phenotypes observed, such as increased FA number, or decreased traction force, are mediated by the increase in Rac1 activation observed in the absence of OPTN.

Most of the phenotypes observed suggest and involvement of other Rho proteins, especially RhoA. Changes in RhoA activity have not been tested in this article, nor RhoA was ever mentioned in the discussion, which I believe are important omissions, especially when CNF1 can also target RhoA.

In my opinion, the key element missing here is to demonstrate that the phenotype changes observed when OPTN is depleted are Rac1 mediated.

This lack of conclusive characterization of the causative roles of each of the key players characterized here make the conclusion read as overinterpreted, missing the caveats or alternative explanations possible.

Major Comments:

-With some exceptions noted below the experiments are sound and well designed.

-The focal adhesions phenotype suggests involvement of RhoA rather than Rac1. This is also true for some of the mechanical response to stiffness observed. Have the authors rule out the involvement of RhoA? This is important, especially since RhoA can also be activated by CNF1.

Specific Comments

-In p6 line 161-162, the activation of Rac1 by CNF1 in different stiffness levels should be shown, at least in the Supp Figures. Similarly, the OTN mRNA in different stiffnesses (Lines 265-266) should also be shown

-In figure 1B-C the increase in the levels of p-PAX, pSrc and p-FAK 9to a lesser extent), correlate with the increase in the levels of the corresponding total proteins. It is hard to explain the substantial increase in total levels of these proteins after only 30-60 min of treatment, especially since GAPDH levels do not change the same way. In my opinion, and based on these observations, normalizing to the levels of GAPDH is not the best option, and the results should be shown as a ratio of phospho/total levels. I wonder, based on the shape of the WB lanes (the pattern is identical), if some of these phospho and total blots were reblotted (compare total PAX to p-PAX or total Src vs p-Src).

-Related to the previous point, the statement in lines 179-180 that the signal reaches max levels in 30 min may also be affected by the total levels, which are drastically lower at 60 min (same argument as above applies)

-Representative WB for S1D should be shown (again related to my comments above)

-In p11 line 310, the authors state that "OPTN positively controls HACE1 activity towards Rac1". This has actually not been directly measured and the increase of ubiquitylated Rac1 may just reflect an increase in binding as shown, but not an increase in activity.

-The focal adhesion phenotype observed in Figure 5 is more representative of active RhoA than Rac1 (which when hyperactivated stimulates the formation of focal contacts rather than mature adhesions), and despite correlation between siOPTN and Rac1 activity, these results do not actually connect the

activation of Rac1 with the phenotype observed. Since CNF1 can also activate RhoA, it is important to rule out that this effect is not mediated by RhoA.

-The decrease in traction force observed in siOPTN is intriguing. Analyzing stress fibers and pMLC could shed some light on the mechanism by which this is occurring. Also, looking at focal adhesions turnover would be informative too, as turnover defects have been shown to promote phenotypes similar to the one observed in this manuscript.

Minor Comments:

-Fig 1D and 1E show the same type of measurement. However, the scales are very different in magnitude. Are the results in 1D normalized?

-Low quality on several blots (S1E, GAPDH for example among others)

-In S3A it looks like expressing HACE1 and RacQ61 may have a negative effect on OPTN ubiquitylation. I disagree with the text conclusion that there is no change.

REVIEWER COMMENTS & POINT-BY-POINT RESPONSE

Reviewer #1 (Remarks to the Author):

Manuscript Summary:

In this manuscript the authors identify a substrate stiffness-modulated role of OPTN in regulating the ligase activity of HACE1 on Rac1. They show, that the invasion of E. coli within HUVEC host cells is dependent on this mechanism. Overall, I think this work is interesting, but I have major concerns about data presentation and analysis, few overstatements and missing technical information.

Major/General Comments:

(1) Nowhere is it stated how many HUVEC cells were seeded. Confluence is not discussed anywhere even though it is stated that proliferation rate changes due cyclin D1 dependency. Confluence can vary significantly depending on matrix stiffness especially given that many cell types (including HUVEC) are not very keen on attaching on very soft matrices. How do authors control for that? Confluence can have a dramatic effect on cell behavior and needs to be at least documented. If anything, the authors appear to seed cells subconfluently so that single cells or cell clusters are analyzed (in the images provided). How relevant is that considering that in blood vessels endothelial cells form a continuous single monolayer? Why did they choose to seed cell sparsely? And why were HUVEC (primary umbilical vein endothelial cells) chosen to conduct this type experiments?

Density seeding for the infection experiment was indicated in the methods section, but the information was omitted regarding the adhesion/proliferation assays. It is now indicated “cells were plated at the density of 55,000 cells/cm²”. The cell density is thus similar in all types of experiments performed.

We have been using in this study a setting of sub-confluent cells for infection experiments to draw a parallel with the study of Rac1 signaling on cell cycle progression, thereby avoiding contact-inhibition. This allows us to draw transposable conclusions on mechanistic analyses whenever possible. Focal adhesion phenotypes and traction force measures at integrin-mediated adhesive structures are also classically analyzed in sub-confluent cells to avoid confounding effects of interdependencies of integrin-mediated and cadherin-mediated forces. Confluence was controlled by keeping identical seeding number on the different ECM stiffnesses, and identical timings post-seeding to perform experiments. Phase-contrast images are now included in Sup. Fig. 2A, they show that HUVEC confluence is similar on a different range of fibronectin stiffnesses : 1 kPa, 4 kPa and 50 kPa, 16 hours post-seeding (overnight). Images in Fig5A and Sup. Fig 5A show cells at 2 hours post-seeding on glass coverslips (70 GPa).

Primary human cells were chosen for our work to ensure at best that Rho GTPases ubiquitylation regulation was normally conserved, since this pathway is often deregulated in cell lines as studied in ref : doi/10.1091/mbc.E05-09-0876. Source for primary human cells amenable to plasmid and SiRNA transfection and biochemistry-compatible high culture yields are scarce, HUVEC offer these advantages. Moreover, uropathogenic E. coli frequently gain access to the bloodstream, and some isolates can cause neonatal meningitidis, making endothelial cells a pathophysiological relevant cell type. To answer Reviewer 2 request, we have confirmed the role of OPTN for bacterial invasion in bladder epithelial cells, supporting the transposable nature of the findings we did in HUVEC cells (Supl. Fig6B).

(2) In Figure 5F how is it possible to make hydrogels of 0.2 kPa (this should be almost acting as fluid). It is very hard to believe that HUVEC could attach on such a soft and highly porous material. Did you confirm that the stiffnesses of the gels are really what you claim through AFM? Given the centrality of ECM stiffness on this paper and given that authors performed AFM on their cells, I consider very important to at least validate once that assumed ECM stiffness is correct. Also, polyacrylamide hydrogels are notorious to show some deviation from gel to gel and batch to batch on their expected stiffness. That also is important to know through AFM measurements on the actual gels. Also, in different figures (experiments) different combinations of matrix stiffnesses are used. Why is that?

The reviewer is correct in that 0.2 kPa gels are quite malleable, but they are definitely solid. We have obtained the hydrogel culture systems from the Matrigen company. After discussing this point with the

manufacturer, Matrigen is confident in the accuracy in their measurements of 0.2kPa (or lower) gels because they make sure they are performing the indentations in a range in which the Hertz theory can be applied. They do not elect to use AFM/nanoindentation due to potential artifacts (such as gel adhesion to the AFM probe) that make estimating the stiffness of soft substrates difficult with these methods. With the macroindentation method performed by Matrigen, there is no ambiguity regarding the position of the gel surface, and therefore, the tip-sample contact point.

Nevertheless, a recent publication presents validation data for 1 kPa and 50 kPa hydrogels from Matrigen using AFM measures. They measure a Young's modulus mean value of 0.95 kPa and 43.39 kPa respectively (DOI: 10.1126/scisignal.abd4077; Supl Fig 1).

The 0.2 kPa gels conditions were used to set the base level of integrin signaling allowing to calculate fold inductions of FAK, Src, P130CAS and Paxillin phosphorylation levels in response to ECM stiffness increase (eg instead of using cells in suspension). As seen below, cells adhere but do not spread much on 0.2 kPa:

(3) In the last section of the paper authors state that “OPTN controls cell contractility”. This is a very strong statement that authors could more precisely address by comparing how siOPTN cells act on high and low stiffness matrices as compared to OPTN expressing cells. If OPTN indeed controls cell contractility, then by doing TFM on varying stiffness hydrogels one should see a gradual increase in forces for WT HUVEC cells. Moreover, the changes should vanish (if they are matrix stiffness dependent and OPTN indeed controls them) in siOPTN cells. I believe this is a very important experiment missing.

This sentence was indeed misleading, thank you.

We have changed the title of figure 7 for : “Loss of OPTN is associated with a decrease in adhesion-mediated cell forces”

and the sentence in line 518 for : “loss of OPTN expression is associated with a decrease in total force transmitted to the environment during cell adhesion”.

Here we show, at least at two different substrate stiffness (experiments on hydrogels or pillar arrays of ~5 kPa modulus, and parallel plates assays at infinite stiffness) that lack of OPTN leads to lower forces applied by cells on their substrates, while these siOPTN cells show adhesion structures of increased size. Indeed, classically in integrin-based rigidity-sensing, the size of adhesion complexes is known to increase with traction forces, and thus with increased substrate stiffness. Our observation suggests a kind of decoupling between force generation and growth/extension of adhesion complexes in the absence of OPTN. Further experimentation that are beyond the scope of our current paper will be needed to comprehensively address the role of OPTN in this phenotype.

(4) The generation of primary HUVEC cells is a very important piece of information that is also missing. What generations were used? Often generation >P8 result in HUVEC with a pancake morphology which is reminiscent of some images seen e.g., at Figure 7 (as opposed to spindled like morphology seen in younger generations P2-P8).

HUVEC are used before P5, and SiRNA transfection was done between P2 and P3. This has now been specified in the methods section (page 23 lines 646 and 665).

(5) Why were AFM measurements performed with a tiny tip and not both a larger spherical tip that would allow to get a better idea of bulk stiffness properties. Perhaps also the large deviation of the AFM data from the mean is due to the usage of a small tip.

Zemła et al. have shown that any kind of tip geometry can measure differences in elasticity either in soft gels or in living cells (Zemła, J., Bobrowska, J., Kubiak, A., Zieliński, T., Pabijan, J., Pogoda, K., ... & Lekka, M. (2020). Indenting soft samples (hydrogels and cells) with cantilevers possessing various shapes of probing tip. *European Biophysics Journal*, 49(6), 485-495).

There were several reasons why we decided to use these AFM probes for this study:

- PFQNM-LC-A-CAL are one of the few pre-calibrated (LDV) commercial probes, meaning they provide the best calibration of the instrument to date, eliminating any bias coming from the use of several cantilevers and the variation in calibration (Schillers, H., Rianna, C., Schäpe, J., Luque, T., Doschke, H., Wälte, M., ... & Radmacher, M. (2017). Standardized nanomechanical atomic force microscopy procedure (SNAP) for measuring soft and biological samples. *Scientific reports*, 7(1), 1-9).
- The use of large colloidal tip has some advantages (better bulk properties evaluation with large indentations as suggested by the reviewer), but also one serious drawback in this case: the point of contact is hard to find precisely, which leads to large errors in the extraction of E for small indentations. This is especially critical for the evaluation of the cortical elasticity of cells. Furthermore, the bottom effect is more pronounced with large indenters on these rather flat and thin cells (Garcia, P. D., & Garcia, R. (2018). Determination of the elastic moduli of a single cell cultured on a rigid support by force microscopy. *Biophysical journal*, 114(12), 2923-2932).
- These probes are neither sharp, nor large spherical shape. The shape is paraboloid with a spherical diameter of 130 nm in diameter. It's a medium size with a well-defined geometry, that could also have deciphered spatial variations in elasticity.

Finally, concerning statistics, each measure used for the graph corresponds the median of several hundreds of elasticity measurements for one cell. As a consequence, the variation seen on the plot is not due the heterogeneity as probed by the AFM tip on a scan, but is a description of the cell population variation.

(6) There is missing information in the figure legends. What are the data presented? How many independent experiments were performed? What is the very basic experimental structure? Also, formatting is inconsistent in text and figures.

We have modified the legends accordingly to include all these information, and we standardized formatting

Specific comments:

(1) In figure 1A: At what density were HUVEC (by cell I assume HUVEC cell) seeded? What generation? HUVECs infections were performed at density of 250,000 cells/well on 12-well plates and before passage P5. This is indicated in the methods section

(2) In figure 1B-C: Why is standard error of the mean shown and not standard deviation? Why are not individual values shown on top of boxplots? Why are controls without error bar?

We have now shown standard deviation; individual values are represented as open dots. Untreated controls are set to 1 for internal normalization of western-blot of each of the three independent experiments, and correspond to the densitometric value of the phospho-signal band relative to the densitometric value of the loading control band (GAPDH). The goal here is to avoid variations in antibody binding efficiency, membrane blocking and washing efficiencies, decay rate of the chemiluminescent substrate used for revelation, that would prevent from cross-comparison of experiments. Therefore, the non-treated sample on each gel is used to normalize every other bands on that same western-blot to obtain fold inductions. Therefore, there is no error bar on controls, and error bars on the rest of measures correspond to variability in the fold inductions from the different independent experiments.

(3) In Figure 2D: How many independent experiments were performed?

We have now included as panel 2E a quantification of 5 independent experiments.

We have cropped the membranes shown in panel 2D to start at 1kPa for coherence with ECM stiffnesses mostly used in this study. This does not change information.

(4) In Figure 3A: Stiffness of the matrix not indicated in text/figure legend after claiming OPTN levels are very stiffness dependent. Also why is here no loading control but in other figures with 'input' there is? Figure 3A investigates OPTN and HACE1 interaction in HUVECs cultured on plastic cultureware (a condition where endogenous OPTN is highly expressed) upon expression of tagged OPTN and HACE1 proteins. This is now indicated in the legend. GAPDH was not systematically blotted in the transfection experiments since we usually refer to the level of expression of each tagged protein. We have included the endogenous GAPDH western blot for this panel.

(5) Figure3B: Authors state 'We found that OPTN knockdown led to a reduction in Rac1Q61L ubiquitylation under conditions of endogenous or overexpressed HACE1'. Authors probably refer to siOPTN + conditions with either Myc-HACE1 + or - .

Yes, endogenous HACE1 expression refers to Myc-HACE1 (-) while overexpressed HACE1 refers to Myc-HACE1 (+) lanes.

But there is also no big difference between siOPTN + / myc-HACE1 + and siOPTN - / myc-HACE1 -. So, if siOPTN + / myc-HACE1 + is considered a reduction in ubiquitylation then in cells without a siOPTN knockdown but endogenous levels of HACE1 the same reduction takes place? So, it is not OPTN dependent in that case?

We are not sure of the point raised by the reviewer. Our aim is to address the regulatory role of OPTN on the function of HACE1, an E3 ligase targeting active Rac1. We show here that knocking down OPTN reduces Rac1 ubiquitylation by HACE1. This holds true in conditions where cells do not overexpress HACE1 (lanes 3 vs 2) and overexpress the HACE1 E3 ligase (lanes 6 vs 5). The fact that Rac-Ub levels in lane 6 are similar to those in lane 3 can either reflect that the balance between HACE1 overexpression and OPTN knockdown is in favor of HACE1 (there is still some OPTN expressed, it is not a genetic KO). An alternative situation would be that HACE1 overexpression bypasses partially the regulation by OPTN, for example because it would be active at a particular location thanks to OPTN and exogenous expression leads to saturation of the cellular locations.

(6) Figure3C: Why did authors chose different stiffnesses than the ones tested for OPTN expression? We apologize for this coarse mistake, the assay was indeed performed on 1-4-8 kPa. This was corrected in the figure.

(7) Figure3D: Why is here no loading control in the input? It would be nice to see the results on a stiff matrix (e.g., 25kPa) to directly see if overexpression of OPTN could abolish stiffness dependent effects.

GAPDH was not systematically blotted in the transfection experiments, since we usually refer to the level of expression of each protein and the important control here is the equal expression of HA_Rac1Q61L in the two lanes.

(8) Figure 3G: What is in fractions 1,4,5?

We now give more details about the sucrose gradient fractionation method in the text (page 11, line 306). It is difficult to answer this question directly. This assay is aimed at separating detergent-resistant membranes (and associated proteins like flotillins) from detergent soluble membranes (and associated proteins like Transferrin Receptor). Other fractions could contain membranes in single uniform phases but with properties intermediate (partial detergent insolubility) between flotillin-enriched detergent resistant and detergent soluble membranes. Characterization of these fractions, if possible, would need further work beyond the scope of our research.

(9) Figure 4A: Here sync is used as annotation in the figure, but only explained in the figure legend referring to 4C. This has now been added in the legend of figure 4A.

Also, why do authors again change the matrix stiffnesses tested (as compared to the previous figure)? In the previous (Figure 3C) they used 1,4 and 12 kPa. We apologize for the mistake in 3C, where 1-4 and 8 kPa were used. Stiffnesses used in 3A and 4C are coherent.

Authors claim that “the fraction of proliferating cells doubles between control cells on high and low stiffness matrix”. This is clearly not true, for 1 kPa is at around 27%, 8 kPa is at around 47%. The exact calculation has now been indicated in the text. “ The proportion of the control cells that entered the S and G2/M phases of the cell cycle increased by 69% on high stiffness compared to low stiffness ECM” [calculation : (46.35-27.4)/ 46.35]

Also, authors state: ' Interestingly, the OPTN knockdown cells reached maximal proliferation when plated on low elastic modulus ECM.' This is also not quite true. Their max is still at 8kPa even though the fraction of proliferating cells greatly increases compared to the control group (also if there is no significant difference for 1 and 8 kPa for the knockdown that is worth being denoted in the graph).

We meant by this sentence that proliferation of SiOPTN cells at 1 and 8 kPa is the same, and not significantly different from the SiCTRL cells plated on 8 kPa. We have now added on the graph that there is no significant difference at 1 versus 8 kPa for the OPTN knockdown cells.

(10) In Figure 4A and B and elsewhere what is “-“ kPa?

This corresponds to the synchronized conditions. For more simplicity/clarity we have just left empty space.

(11) In Figure 4C: What matrix stiffness was used here? It would have been nice to see soft/stiff conditions compared again, as that was the point in the up and down regulation of cyclin D1 dependency of OPTN.

Soft/stiff regulation of cyclin D1 dependency of OPTN is indeed shown in figure 4B. In Figure 4C we used a “full mitogen” stimulation by stiff ECM-here fibronectin coated plastic- and 20% serum in order to show that while all possible proliferative signals are provided, OPTN knockdown cells display induced cyclin D1 protein levels without impacting the rate of its transcription.

(12) In Figure 5: In the text the font size changes all the time which is not nice to read.

This has been corrected in the new figure

(13) In Figure 5D as compared to Figure 6C the stiffness of the control cells is different. Why is that? It looks like this difference is similar to the different between siCntr and siOPTN which is significant. Why SEM and not SD? What are the authors thoughts on the high variance? Does it have to do with where the cells were probed (nucleus versus cell edge). Why are the authors not providing the stiffness maps of cells since they probed stiffness by scanning? Also, what matrix stiffness was used?

Cells in 5D and 6C are in different culture conditions : 5D are cells plated for 1h30 on fibronectin-coated plates in defined medium, while 6C are cells cultured overnight in complete medium. This may be why cortical elastic moduli of control cells differ, and this makes the comparison difficult. The variance has been discussed in general comment N°5. Measurements were always performed in the similar perinuclear region of cells to avoid 2 common issues, bottom effect on the edge of the cell, and nucleus effect (nucleus is an order of magnitude stiffer than surrounding cytoplasm). This is specified in the methods section. Representative stiffness maps are now provided for these two sets of experiments in Sup. Fig.5D and Sup Fig.6D.

(14) Figure 5F: Here there is yet a new set of tested matrix stiffnesses. 0.2 kPa has not been tested before (only in one supplemental figure panel as a control) and now 4 kPa was added. Also 0.2 kPa is in essence liquid. I don't think that cells could attach there. It would be a nice proof to include images (e.g., phase contrast) of how cells look at different matrix stiffnesses reported here and also of course to validate that the stiffnesses are the ones that authors claim.

This comment has been addressed in our response to specific comment N°2. Cells do attach on 0.2 kPa fibronectin-coated hydrogels but they don't spread much (see above image). Phase contrast images of cells used for the proteomics analysis were included in Sup. Fig. 2A

(15) Figure 6: Is only one stiffness use here and which? Also in panel C why is the scanning size different than in the previous figure (40x40 versus 20x20)? In panel D why are there no error bars for the negative controls?

Cells are on plastic, this has now been indicated in the legends.

The AFM scanning size is not different between figures, it is $6 \mu\text{m}^2$ in the perinuclear region of cells in both experiments. What differs is the number of pixels in each map. This has no influence on the measurement, it only governs the map resolution (now provided as a supplemental figure) and the time of scanning. All of critical parameters were kept constant between experiments: tip shape, cantilever spring constant, tip velocity, and indentation.

Panel D : Concerning no error bars on controls, see our response to specific comment N° 2

(15) Figure 7A: What does it mean traction force curves? Is that the maximum force? Or is it the mean absolute force along the cell area? How was segmentation done to define cell borders or was that not done? Why just a single cell is shown when the average of many cells (and SD) could be shown? Circles in the legends are almost invisible.

The traction force curve represents a measure of the uniaxial force (F) exerted by a cell overtime, as measured with the parallel plate setup. We have added a scheme of the experimental setup in Fig. 7A, together with an inset representing a brightfield image of a SiCTRL and a SiOPTN cell caught between the cantilevers (Fig. 7B) to help comprehension of the technique by the reader. There is no segmentation of cells, F is measured through the deflection d of the flexible plate of known stiffness k : $F = k d$. Traction force measurements with the parallel plate setup are similar in their principle to AFM force measurements, which are based on the deflection of a cantilever of known stiffness. The representation of all curves superimposed would be unintelligible and an average curve would obliterate the characteristic shape of each traction curve due to variability. This is why we plot in panels 7B (new 7C) and 7C (new 7D) the maximal force reached for each cell and the rate-of-force increase (slope of each curve) respectively, allowing mean/median value calculations and statistical evaluations.

(16) Figure 7B-C: Where is the standard deviation. Also, how many independent experiments were done? The new plots, according to Nature Communications guidelines for this amount of measures, are box and whiskers (“For sample sizes larger than 10, please consider box-and-whisker or violin plots as alternatives. Measures of centrality, dispersion and/or error bars should be plotted and described in the figure legend”) 3 independent experiments were performed, it is now indicated in the legend.

(17) Figure 7D: It would have been highly relevant to compare how siOPTN cells act on high and low stiffness as compared to OPTN expressing cells. If OPTN indeed controls cell contractility, then by doing TFM on varying stiffness hydrogels one should see a gradual increase in forces. Also, the changes should vanish (if they are matrix stiffness dependent and OPTN indeed controls them) in siOPTN cells. Also, on the left panels authors show magnitude of traction stresses. On the right is this a phase contrast image? Why/how is the phase contrast differently adjusted for cell versus its surrounding environment?

This comment has been addressed above, see General comment N°3

On the left we plotted the magnitude of traction stresses, on the right it is a phase contrast image where we super-imposed the force vectors and the region that we used to crop the force field in order to compute the associated stresses. This is now specified in the figure legend.

(18) Figure 7E-F: Authors do not state which statistical methods they used to compare forces, energy etc. That is a general comment. Also, when force vectors are added up to compare the total forces, did they take the total size of the cell into consideration to account for morphological changes in area or not?

The statistical test used is Student’s t test, it is now specified in the legend.

The total force that is plotted is not taking into account the surface of the cell (now panel 7F). The contractile energy (now panel 7G) is the force multiplied by each displacement integrated over the whole cell surface, so it is integrating the cell surface in its definition.

Reviewer #2 (Remarks to the Author):

This work builds on previous studies indicating that activation of Rac1 by the UPEC-associated toxin CNF1 can facilitate bacterial entry into host cells. Host cells can counter aberrant Rac1 activation by CNF1 through HACE1-mediated ubiquitination and subsequent proteosomal degradation of activated Rac1.

Petracchini et al. dig deeper into this system, showing that CNF1 activities augment FimH-mediated invasion of host cells by UPEC via integrin receptors while also providing new insight into the regulatory control of Rac1 and cytoskeletal tension/dynamics. More specifically, they show that OPTN can regulate HACE1 activity, thereby impacting the levels of active Rac1, in response to external ECM stiffness. The paper is dense with information and utilizes a great set of diverse experimental approaches with well-written text that should help make the work accessible to a wide range of individuals with varied backgrounds. Many pathogens, in addition to UPEC, hijack integrin receptors. This, and clear links to cell cycle progression and a number of other key cellular pathways, will make this study of great interest to a broad swath of scientists. There are a few mostly minor issues that should be clarified, but overall enthusiasm for this well-thought-out paper is high.

Bigger Issues:

- This study uses the HUVEC cell line. Some explanation for this choice should be provided, as most studies that examine UPEC invasion employ bladder epithelial cells. The specific cell line used should be made clear at the start of the Results section. It would be useful to the UTI field to know if some of the key findings were also true in bladder cells. For example, would silencing OPTN expression also inhibit UPEC entry into bladder epithelial cells?

HUVEC are primary cells (human) and were chosen for our work to ensure at best that Rho GTPases ubiquitylation regulation was normally conserved, since this pathway is often deregulated in cell lines as studied in ref : doi/10.1091/mbc.E05-09-0876. Source for primary human cells amenable to plasmid and SiRNA transfection and biochemistry-compatible high culture yields are scarce, HUVEC offer these advantages. Moreover, uropathogenic E. coli frequently gain access to the bloodstream, and some isolates can cause neonatal meningitis, making endothelial cells a pathophysiological relevant cell type. To address reviewer-2 concern, we have performed OPTN silencing in HTB-9 bladder epithelial cells and we could confirm that the function of OPTN in bacterial invasion is conserved. Also, similarly to our findings in HUVECs, OPTN silencing does not impact bacteria attachment to cells. This is now included in Sup. Fig.6B

- Do any of the inhibitors used in the paper (e.g. CK-666 or blebbistatin) affect growth or viability of UTI89 at the concentrations used?

The pharmacological inhibitors used had no effect on the bacterial growth. This is now showed in Sup. Fig.11

- CNF1 is used at a concentration of 1 nM. Can the authors comment on the physiological relevance of this concentration? Is this level seen during actual infections?

This concentration corresponds to the maximal activity of the toxin, as measured in the cytotoxicity assay on Hep-2 cells (ref 99, DOI : 10.1016/S0076-6879(06)06033-2) : 100% multinucleated cells in 48h. It is one order of magnitude above the amount produced by UTI89 in our conditions of infection (MOI:100). We have now included a thorough description of why we use this concentration in the methods section, and SupFig1J.

- The study uses MOIs of 100, which is substantially higher than is typically used in the field. Why were so many bacteria used? Are there dosage effects with the HUVECs? This is not necessarily a major issue, but it would be good to understand why the high MOI was chosen.

A MOI of 100 is typical of *in vitro* studies addressing very short experimental windows, as it is in our case with 30 min adhesion of bacteria and 30 min antibiotic treatment to recover only intracellular population. As an example, in a recent paper addressing host restriction pathways during infection of HTB-9 bladder epithelial cells with the UPEC strain CFT073, authors use MOI of 100 in 30 min infections or MOI of 50 in 2 hours infections (doi : 10.1038/s41467-021-22726-8).

Moreover, inoculation of bladder to induce UTI in mice models is classically done using $1-2 \times 10^8$ CFU/mouse (doi:10.1038/nprot.2009.116) and the amount of urothelial cells in a mouse bladder has been estimated to $1-5 \times 10^6$ cells (doi:10.1007/BF02890136). Therefore, we think that MOI 100 should not be necessarily regarded as overwhelming.

Minor issues:

- The Summary could do a better job of highlighting the key major findings of the paper more directly. Lines 57-59 seem somewhat redundant with the final sentence of the Summary.

We have modified the summary

- Line 75 – it is not entirely clear how 'critical' FimH is for UPEC persistence within the gut, as at least one paper noted that FimH is dispensable in some situations (doi: 10.1128/IAI.00746-17)

We have now replaced “essential” by “important” in our sentence to tone down slightly the requirement of FimH:

Expression of chaperone-usher pathway (CUP) type I pili tipped with the adhesin FimH is important for colonization, invasion and persistence of UPEC in the mouse bladder and to form persistent reservoirs in the gastrointestinal tract

- Line 102 and line 387 should have references.

References were added in the manuscript

- Line 212 – use of 4 kPa as an “intermediate” stiffness value seems strange in considering the lower (1 kPa) and higher “50 kPa) values used in the paper. Why not use 25 kPa, or was 4 kPa used in anticipation of non-linear effects?

By “intermediate” we did not mean mathematical half but rather an ECM elastic modulus that is not characterized as “soft” but above, and would have a biological relevance in epithelial cells proliferation control, as reported in the literature. The 1 to 4 kPa difference is the order of difference seen in cancer-associated ECM stiffening, as characterized in breast cancer in the reference : Paszek, M. J. *et al.* Tensional homeostasis and the malignant phenotype. *Cancer Cell* **8**, 241–254 (2005).

We have eliminated the adjective “intermediate” to avoid confusion.

- The font size used for Fig 2C makes the text very hard to read, even with zooming. The CNF1 nodes (yellow with orange edges) could be made more visible to help readers.

The visualization of this panel has been improved.

Reviewer #3 (Remarks to the Author):

The manuscript by Petracchini and colleagues explores the role of ECM stiffness during bacterial invasion. The authors identify a protein, OPTN, which is upregulated by stiffness. OPTN has been previously shown to interact with the Ubiquitin ligase HACE1, which regulates the ubiquitylation and degradation of active Rac1. Silencing OPTN promotes Rac1 activation and cells that show more and larger focal adhesions, but surprisingly a decreased traction force.

The manuscript tries to connect the bacterial toxin CNF1 with Rac1 activation, enhanced stiffness with increased invasion, and OPTN mediated Rac1 inactivation to mechanotransduction. The individual observations presented in this manuscript are interesting and suggest a concerted mechanism connecting CNF1 to mechanotransduction to regulation of Rac1 activity. However, in my opinion, these connections are not properly validated.

The individual claims are well supported by the results (e.g. OPTN expression is regulated by cellular stiffness). However, the connections between the independent results are not clearly supported. For example, a causal link between Rac1 activation and the phenotypes observed is not demonstrated in this article. It is not even tested that some of the phenotypes observed, such as increased FA number, or decreased traction force, are mediated by the increase in Rac1 activation observed in the absence of OPTN.

Most of the phenotypes observed suggest an involvement of other Rho proteins, especially RhoA. Changes in RhoA activity have not been tested in this article, nor RhoA was ever mentioned in the discussion, which I believe are important omissions, especially when CNF1 can also target RhoA. In my opinion, the key element missing here is to demonstrate that the phenotype changes observed when OPTN is depleted are Rac1 mediated.

This lack of conclusive characterization of the causative roles of each of the key players characterized here make the conclusion read as overinterpreted, missing the caveats or alternative explanations possible.

Major Comments:

-With some exceptions noted below the experiments are sound and well designed.
-The focal adhesions phenotype suggests involvement of RhoA rather than Rac1. This is also true for some of the mechanical response to stiffness observed. Have the authors rule out the involvement of RhoA? This is important, especially since RhoA can also be activated by CNF1.

Specific Comments

-In p6 line 161-162, the activation of Rac1 by CNF1 in different stiffness levels should be shown, at least in the Supp Figures.

We have now added this measure in Supplemental Fig. 1B

Similarly, the OTN mRNA in different stiffnesses (Lines 265-266) should also be shown

We have now added this experiment in Supplemental Fig. 2E

-In figure 1B-C the increase in the levels of p-PAX, pSrc and p-FAK (to a lesser extent), correlate with the increase in the levels of the corresponding total proteins. It is hard to explain the substantial increase in total levels of these proteins after only 30-60 min of treatment, especially since GAPDH levels do not change the same way. In my opinion, and based on these observations, normalizing to the levels of GAPDH is not the best option, and the results should be shown as a ratio of phospho/total levels. I wonder, based on the shape of the WB lanes (the pattern is identical), if some of these phospho and total blots were reblotted (compare total PAX to p-PAX or total Src vs p-Src).

-Related to the previous point, the statement in lines 179-180 that the signal reaches max levels in 30 min may also be affected by the total levels, which are drastically lower at 60 min (same argument as above applies)

The blots we were presenting in the original figure were indeed blotted first against the phospho-proteins (Top of membrane) and the loading control GAPDH (Bottom of membrane). Top membranes were then stripped and re-probed against each of the total proteins. We agree with reviewer 3 that this generated artefactual variations in the level of total proteins, because stripping did not erase totally the previous antibody. This indeed motivated our choice to quantify phospho-signals relative to GAPDH. We have re-run blots of the experiments corresponding to the ones shown/used in Figure 1B and 1C to generate new panel Supplemental 1C and verify our quantifications. Our quantifications are correct by virtue of total protein levels being invariable, as shown in the graph below representing total FAK, Src, p130CAS and PAX expression levels relative to GAPDH densitometric measures on each of the migrations.

We added the sentence in the figure legend:

“Expression of FAK, PAX, Src and p130CAS are not modified in our experimental conditions.”

-Representative WB for S1D should be shown (again related to my comments above)

Representative western blots are now provided in the Supplemental 1F panel together with the quantifications.

-In p11 line 310, the authors state that “OPTN positively controls HACE1 activity towards Rac1”. This has actually not been directly measured and the increase of ubiquitylated Rac1 may just reflect an increase in binding as shown, but not an increase in activity.

Reviewer 3 is absolutely correct. The sentence has now been changed to:

“In conclusion, OPTN positively controls the ubiquitylation of Rac1 by HACE1 E3 ubiquitin ligase, thereby limiting Rac1 activity in cells.”

-The focal adhesion phenotype observed in Figure 5 is more representative of active RhoA than Rac1 (which when hyperactivated stimulates the formation of focal contacts rather than mature adhesions), and despite correlation between siOPTN and Rac1 activity, these results do not actually connect the activation of Rac1 with the phenotype observed. Since CNF1 can also activate RhoA, it is important to rule out that this effect is not mediated by RhoA.

To address this major criticism, we have included measures of RhoA-ROCK activity, assessed by p-MLC monitoring, in response to fibronectin attachment, and to CNF1 stimulation. We could see that cells knocked down for OPTN do not present an over-activation of the RhoA pathway (Supplemental Fig. 3C and Supplemental Fig. 6C), which could have explained the focal adhesions phenotype. If any difference, a tendency was towards less p-MLC in OPTN-KD cells, although non-significant. We have also tested the effect of Rac inhibition on cell spreading, and could measure that the OPTN-KD cells were relying on the Rac pathway to a higher extent than control cells (Supplemental Fig. 5E, 69% inhibition of spreading vs 38% respectively).

-The decrease in traction force observed in siOPTN is intriguing. Analyzing stress fibers and pMLC could shed some light on the mechanism by which this is occurring. Also, looking at focal adhesions turnover would be informative too, as turnover defects have been shown to promote phenotypes similar to the one observed in this manuscript.

We agree with Rev3 on the compelling nature of this phenotype. A deep and comprehensive understanding of OPTN role in the connection between adhesion structures and force buildup would need the setup of specialized techniques and further extensive researches that are beyond the scope of this study. They would per se uphold another publication.

We have added the following sentence in the discussion section : “Alternative mechanisms like a reduced turnover of FAs in OPTN knockdown cells could also contribute to their increase in size. Further researches are needed to investigate such hypothesis and the underlying molecular pathways”

Minor Comments:

-Fig 1D and 1E show the same type of measurement. However, the scales are very different in magnitude. Are the results in 1D normalized?

Yes, results in 1D represent CFU/mL for each SiRNA transfection normalized to CFU/mL obtained in the SiCtrl condition. To avoid confusion, the y axis legend was changed from “ Internalized bact (Norm. CFU/mL) ” in the initial graph to “Internalized bact (fold change to SiCtrl; Norm. CFU/mL)

-Low quality on several blots (S1E, GAPDH for example among others)

We provided better western blots for S1E (now S1G) and 1B total proteins (new S1C); and included several new western blots that we wish to be of good quality for figure panels S1F, Fig 3A GAPDH, S3C, Fig 4B, S6A, S6B, S6C.

-In S3A it looks like expressing HACE1 and RacQ61 may have a negative effect on OPTN ubiquitylation. I disagree with the text conclusion that there is no change.

Reviewer 3 is correct. We intended in this approach to determine if OPTN was a target of the HACE1 E3 ligase activity. We have replaced “not modified” by “not induced”; the sentence is as follows:

“Although OPTN presented a ubiquitylation profile, the latter was not induced by the presence of HACE1 or the Rac1Q61L active mutant (Sup Fig. 3A).”

REVIEWER COMMENTS

Reviewer #1 (Remarks to the Author):

Manuscript Summary:

In this manuscript the authors identify an extracellular matrix stiffness-modulated role of OPTN in regulating the ligase activity of HACE1 on Rac1. Authors show that the invasion of E. coli within HUVEC host cells is dependent on this mechanism. Overall, I think this work is very interesting, and authors have addressed quite of my prior concerns. Below please find some further comments I have that I think are important to be addressed.

Comments:

1) In Suppl Figure 2A where images of cells on varying stiffness matrices are shown the cells are quite confluent. That is not the case in the figure shown for cells at 0.2 kpa (response to point number #2 of my prior revisions). Moreover, the contrast in this image is not good, making inspection hard. Moreover, there is no scale bar in the image. In other experiments conducted cells are sparsely seeded. In general cell confluence is a very important parameter to account for and the effect of cell confluence can be much more significant than say the effect of matrix stiffness. Both biochemical and biomechanical measurements can change based on the seeding density of cells. See for example previous works: Califano and Reinhart-King, 2010; Heng et al., 2011; Hur et al., 2012. So one big concern I have has to do with the fact that in some experiments cells are examined while subconfluent whereas in other cells are confluent. This is not correct to do. Also, for example on 0.2 kPa stiffens matrices cells are unable to spread. So are results that authors demonstrate due to the stiffness or due to the decreased spreading area? That should be at least discussed. Also, proper images of cells on all stiffness matrices tested should be shown at least in the supplement.

2) Regarding Figure 7, in the figure legend “schematic” needs to be capital. Also, authors mention that “F” stands for the traction force but nowhere in the schematic is that shown (and this is the most important variable measured). For someone who is not familiar with h this technique it is important that the sketch at least is able to explain what forces are acting and where.

3) Again, in Figure 7 since authors also used classical TFM (shown in panel E) it is pertinent in the figure legends to explain which boxplots were calculated with which of the force measuring techniques. That is

currently not clear. Also, in the images where forces are shown as vectors there should be somewhere a scale bar (i.e. a vector of a certain length and how much that is in nN). That is correctly done in Suppl. Fig. 7A but not in main Figure 7.

4) For assessing whether the mean of distributions is significantly different in condition “A” versus “B” authors used everywhere a Student’s t-test. This test assumes that the data are normally distributed, which I doubt is the case (i.e. it is a parametric test). Unless authors show that this is the case, non-parametric tests (e.g. Wilcoxon ranksum) should be used instead because assessment of significance is based on an assumption (normality of data distribution) that is actually not tested.

References

Califano, J.P., and Reinhart-King, C.A. (2010). Substrate Stiffness and Cell Area Predict Cellular Traction Stresses in Single Cells and Cells in Contact. *Cell Mol Bioeng* 3, 68-75. 10.1007/s12195-010-0102-6.

Heng, B.C., Bezerra, P.P., Preiser, P.R., Law, S.K., Xia, Y., Boey, F., and Venkatraman, S.S. (2011). Effect of cell-seeding density on the proliferation and gene expression profile of human umbilical vein endothelial cells within ex vivo culture. *Cytotherapy* 13, 606-617. 10.3109/14653249.2010.542455.

Hur, S.S., del Alamo, J.C., Park, J.S., Li, Y.S., Nguyen, H.A., Teng, D., Wang, K.C., Flores, L., Alonso-Latorre, B., Lasheras, J.C., and Chien, S. (2012). Roles of cell confluency and fluid shear in 3-dimensional intracellular forces in endothelial cells. *Proc Natl Acad Sci U S A* 109, 11110-11115. 1207326109 [pii] 10.1073/pnas.1207326109 [doi].

Reviewer #2 (Remarks to the Author):

The authors nicely address the concerns previously raised by this reviewer, but there are still a few issues regarding the use of blebbistatin and CK666.

- The data presented in Supp Fig 1I indicate that blebbistatin and CK666 do not affect UT189 growth rate. However, since dead bacteria can contribute to OD readings, it is a good idea to plate the bacteria (at least at the endpoint) to better rule out possible effects of the drugs on bacterial viability. Alternatively,

data showing that these drugs do not alter UTI89 levels of adherence to host cells would also support the idea that the drugs do not affect pathogen viability in the cell culture assays. This information should be included anyway to support the invasion data shown in Fig. 1E. The Methods should note how long the drug treatments were prior to infection and the gentamicin treatments.

- Line 645, should read "Except for the experiments shown in Sup Fig 6B, where 5637 cells (ATCC HTB-9) were used...". ATCC notes that this cell line should be referred to as 5637 (ATCC HTB-9). The cells are 5637 cells (not HTB-9 cells).

Reviewer #3 (Remarks to the Author):

In this revised version, the authors have addressed all my concerns, either in the text, or by providing additional experiments. I believe the authors have worked very hard on addressing the concerns of the other two reviewers, so I recommend this article for publications.

REVIEWER COMMENTS & POINT-BY-POINT RESPONSE – Second Revision

Reviewer #1 (Remarks to the Author):

Manuscript Summary:

In this manuscript the authors identify an extracellular matrix stiffness-modulated role of OPTN in regulating the ligase activity of HACE1 on Rac1. Authors show that the invasion of *E. coli* within HUVEC host cells is dependent on this mechanism. Overall, I think this work is very interesting, and authors have addressed quite of my prior concerns. Below please find some further comments I have that I think are important to be addressed.

Comments:

1) In Suppl Figure 2A where images of cells on varying stiffness matrices are shown the cells are quite confluent. That is not the case in the figure shown for cells at 0.2 kPa (response to point number #2 of my prior revisions). Moreover, the contrast in this image is not good, making inspection hard. Moreover, there is no scale bar in the image. In other experiments conducted cells are sparsely seeded. In general cell confluence is a very important parameter to account for and the effect of cell confluence can be much more significant than say the effect of matrix stiffness. Both biochemical and biomechanical measurements can change based on the seeding density of cells. See for example previous works: Califano and Reinhart-King, 2010; Heng et al., 2011; Hur et al., 2012. So one big concern I have has to do with the fact that in some experiments cells are examined while subconfluent whereas in other cells are confluent. This is not correct to do. Also, for example on 0.2 kPa stiffens matrices cells are unable to spread. So are results that authors demonstrate due to the stiffness or due to the decreased spreading area? That should be at least discussed. Also, proper images of cells on all stiffness matrices tested should be shown at least in the supplement.

The cells on 0.2 kPa were shown in our previous response as a request (rev1 point2) to prove that a hydrogel of this stiffness is not a fluid and that cells do attach. The 0.2 kPa condition was only used to set a reference for integrin signaling *ie* to set the base level of integrin signaling allowing to calculate fold inductions of FAK, Src, P130CAS and Paxillin phosphorylation levels in response to ECM stiffness increase (Figure 5F). It corresponds to short time point post-seeding. A scale bar was added on the figure for rev (point 2, first revisions).

The seeding density is the same in our experiments, but the timing for observation differs which may confuse reviewer1. When focal adhesion or adhesion-mediated integrin signaling are studied, which are adhesion-proximal events, experiments are performed at 2h and 1 h post-seeding respectively as indicated in legends. Other experiments assessing more distal events are done on cells after overnight seeding. We do in our study what laboratories classically do when assessing cellular effects in response to varying ECM stiffness, namely seeding cells at a fixed density on varying ECM stiffnesses, and measuring their relative behavior by monitoring their responses, *eg* cell division, signaling, infection.

We have added the following sentence in text (line 668):

“Cells cultured on the fibronectin-coated hydrogels of varying stiffness displayed similar densities between 1 and 50 kPa and no significant differences were detected in the number of attached cells.”

Counting cells on the different ECM stiffnesses indicate a similar adhesion upon overnight incubation :

2) Regarding Figure 7, in the figure legend “schematic” needs to be capital. Also, authors mention that “F” stands for the traction force but nowhere in the schematic is that shown (and this is the most important variable measured). For someone who is not familiar with this technique it is important that the sketch at least is able to explain what forces are acting and where.

We have modified the scheme to picture the uniaxial traction force and to make apparent at first glance the calculation to obtain the traction force F value from the measured variables. We have also added the reference for the original publication of this method in the figure legend.

3) Again, in Figure 7 since authors also used classical TFM (shown in panel E) it is pertinent in the figure legends to explain which boxplots were calculated with which of the force measuring techniques. That is currently not clear. Also, in the images where forces are shown as vectors there should be somewhere a scale bar (i.e. a vector of a certain length and how much that is in nN). That is correctly done in Suppl. Fig. 7A but not in main Figure 7.

We have now specified in the legend for Figure 7: Measurement of cellular forces using single-cell traction force (A-D) and traction force microscopy (E-G) methods.

In Figure 7E, we represent a map of the magnitude of the traction forces (heatmap). We have now specified in the legend: “magnitude of the traction forces is color coded, brighter signals indicate higher force, see scale on the right”.

4) For assessing whether the mean of distributions is significantly different in condition “A” versus “B” authors used everywhere a Student’s t-test. This test assumes that the data are normally distributed, which I doubt is the case (i.e. it is a parametric test). Unless authors show that this is the case, non-parametric tests (e.g. Wilcoxon ranksum) should be used instead because assessment of significance is based on an assumption (normality of data distribution) that is actually not tested.

We have redone the statistical tests in order to replace all Student’s t-test by a Wilcoxon rank sum test. This does not change the statistical validations, except one analysis displaying a lower p value (figure 5E with now $p^* < 0.05$ instead of $p^{**} \leq 0.01$). All our conclusions remain unchanged. Figure legends and methods sections were modified accordingly.

References

- Califano, J.P., and Reinhart-King, C.A. (2010). Substrate Stiffness and Cell Area Predict Cellular Traction Stresses in Single Cells and Cells in Contact. *Cell Mol Bioeng* 3, 68-75. 10.1007/s12195-010-0102-6.
- Heng, B.C., Bezerra, P.P., Preiser, P.R., Law, S.K., Xia, Y., Boey, F., and Venkatraman, S.S. (2011). Effect of cell-seeding density on the proliferation and gene expression profile of human umbilical vein endothelial cells within ex vivo culture. *Cytotherapy* 13, 606-617. 10.3109/14653249.2010.542455.
- Hur, S.S., del Alamo, J.C., Park, J.S., Li, Y.S., Nguyen, H.A., Teng, D., Wang, K.C., Flores, L., Alonso-Latorre, B., Lasheras, J.C., and Chien, S. (2012). Roles of cell confluency and fluid shear in 3-dimensional intracellular forces in endothelial cells. *Proc Natl Acad Sci U S A* 109, 11110-11115. 1207326109 [pii] 10.1073/pnas.1207326109 [doi].

Reviewer #2 (Remarks to the Author):

The authors nicely address the concerns previously raised by this reviewer, but there are still a few issues regarding the use of blebbistatin and CK666.

- The data presented in Supp Fig II indicate that blebbistatin and CK666 do not affect UTI89 growth rate. However, since dead bacteria can contribute to OD readings, it is a good idea to plate the bacteria (at least at the endpoint) to better rule out possible effects of the drugs on bacterial viability. Alternatively, data showing that these drugs do not alter UTI89 levels of adherence to host cells would also support the idea that the drugs do not affect pathogen viability in the cell culture assays. This information should be

included anyway to support the invasion data shown in Fig. 1E. The Methods should note how long the drug treatments were prior to infection and the gentamicin treatments.

We have replaced the initial growth curves by two graphs: one graph representing OD values in the exponential and stationary phases of bacterial growth and one graph representing assessment of CFU values at each of these phases (at same time points the culture OD is measured and serial dilutions are performed and plated). The graphs are now shown in SupFig1J.

We have added data showing no effect of the drugs on bacterial attachment to cells in Sup Fig1I

The drugs are added 2h prior to infection. This is now specified in the methods, line 673:

“The pharmacological inhibitors used were with a two hour pre-treatment of cells when infections were performed.”

- Line 645, should read “Except for the experiments shown in Sup Fig 6B, where 5637 cells (ATCC HTB-9) were used...”. ATCC notes that this cell line should be referred to as 5637 (ATCC HTB-9). The cells are 5637 cells (not HTB-9 cells).

We thank Reviewer 2 for his remark on ATCC nomenclature. The name of this cell line was changed into 5637 cells throughout the manuscript

Reviewer #3 (Remarks to the Author):

In this revised version, the authors have addressed all my concerns, either in the text, or by providing additional experiments. I believe the authors have worked very hard on addressing the concerns of the other two reviewers, so I recommend this article for publications.

We are grateful to reviewer 3 for the quality and interest of his/her comments during the revision and for acknowledging our efforts.

REVIEWER COMMENTS & POINT-BY-POINT RESPONSE – First Revision-

Reviewer #1 (Remarks to the Author):

Manuscript Summary:

In this manuscript the authors identify a substrate stiffness-modulated role of OPTN in regulating the ligase activity of HACE1 on Rac1. They show, that the invasion of E. coli within HUVEC host cells is dependent on this mechanism. Overall, I think this work is interesting, but I have major concerns about data presentation and analysis, few overstatements and missing technical information.

Major/General Comments:

(1) Nowhere is it stated how many HUVEC cells were seeded. Confluence is not discussed anywhere even though it is stated that proliferation rate changes due cyclin D1 dependency. Confluence can vary significantly depending on matrix stiffness especially given that many cell types (including HUVEC) are not very keen on attaching on very soft matrices. How do authors control for that? Confluence can have a dramatic effect on cell behavior and needs to be at least documented. If anything, the authors appear to seed cells subconfluently so that single cells or cell clusters are analyzed (in the images provided). How relevant is that considering that in blood vessels endothelial cells form a continuous single monolayer? Why did they choose to seed cell sparsely? And why were HUVEC (primary umbilical vein endothelial cells)

chosen to conduct this type experiments?

Density seeding for the infection experiment was indicated in the methods section, but the information was omitted regarding the adhesion/proliferation assays. It is now indicated “cells were plated at the density of 55,000 cells/cm²”. The cell density is thus similar in all types of experiments performed.

We have been using in this study a setting of sub-confluent cells for infection experiments to draw a parallel with the study of Rac1 signaling on cell cycle progression, thereby avoiding contact-inhibition. This allows us to draw transposable conclusions on mechanistic analyses whenever possible. Focal adhesion phenotypes and traction force measures at integrin-mediated adhesive structures are also classically analyzed in sub-confluent cells to avoid confounding effects of interdependencies of integrin-mediated and cadherin-mediated forces. Confluence was controlled by keeping identical seeding number on the different ECM stiffnesses, and identical timings post-seeding to perform experiments. Phase-contrast images are now included in Sup. Fig. 2A, they show that HUVEC confluence is similar on a different range of fibronectin stiffnesses : 1 kPa, 4 kPa and 50 kPa, 16 hours post-seeding (overnight). Images in Fig5A and Sup. Fig 5B show cells at 2 hours post-seeding on glass coverslips (70 GPa).

Primary human cells were chosen for our work to ensure at best that Rho GTPases ubiquitylation regulation was normally conserved, since this pathway is often deregulated in cell lines as studied in ref : doi/10.1091/mbc.E05-09-0876. Source for primary human cells amenable to plasmid and SiRNA transfection and biochemistry-compatible high culture yields are scarce, HUVEC offer these advantages. Moreover, uropathogenic E. coli frequently gain access to the bloodstream, and some isolates can cause neonatal meningitidis, making endothelial cells a pathophysiological relevant cell type. To answer Reviewer 2 request, we have confirmed the role of OPTN for bacterial invasion in bladder epithelial cells, supporting the transposable nature of the findings we did in HUVEC cells (Supl. Fig6B).

(2) In Figure 5F how is it possible to make hydrogels of 0.2 kPa (this should be almost acting as fluid). It is very hard to believe that HUVEC could attach on such a soft and highly porous material. Did you confirm that the stiffnesses of the gels are really what you claim through AFM? Given the centrality of ECM stiffness on this paper and given that authors performed AFM on their cells, I consider very important to at least validate once that assumed ECM stiffness is correct. Also, polyacrylamide hydrogels are notorious to show some deviation from gel to gel and batch to batch on their expected stiffness. That also is important to know through AFM measurements on the actual gels. Also, in different figures (experiments) different combinations of matrix stiffnesses are used. Why is that?

The reviewer is correct in that 0.2 kPa gels are quite malleable, but they are definitely solid. We have obtained the hydrogel culture systems from the Matrigen company. After discussing this point with the manufacturer, Matrigen is confident in the accuracy in their measurements of 0.2kPa (or lower) gels because they make sure they are performing the indentations in a range in which the Hertz theory can be applied. They do not elect to use AFM/nanoindentation due to potential artifacts (such as gel adhesion to the AFM probe) that make estimating the stiffness of soft substrates difficult with these methods. With the macroindentation method performed by Matrigen, there is no ambiguity regarding the position of the gel surface, and therefore, the tip-sample contact point.

Nevertheless, a recent publication presents validation data for 1 kPa and 50 kPa hydrogels from Matrigen using AFM measures. They measure a Young's modulus mean value of 0.95 kPa and 43.39 kPa respectively (DOI: 10.1126/scisignal.abd4077; Supl Fig 1).

The 0.2 kPa gels conditions were used to set the base level of integrin signaling allowing to calculate fold inductions of FAK, Src, P130CAS and Paxillin phosphorylation levels in response to ECM stiffness increase (eg instead of using cells in suspension). As seen below, cells adhere but do not spread much on 0.2 kPa (scale bar= 100μm):

(3) In the last section of the paper authors state that “OPTN controls cell contractility”. This is a very strong statement that authors could more precisely address by comparing how siOPTN cells act on high and low stiffness matrices as compared to OPTN expressing cells. If OPTN indeed controls cell contractility, then by doing TFM on varying stiffness hydrogels one should see a gradual increase in forces for WT HUVEC cells. Moreover, the changes should vanish (if they are matrix stiffness dependent and OPTN indeed controls them) in siOPTN cells. I believe this is a very important experiment missing.

This sentence was indeed misleading, thank you.

We have changed the title of figure 7 for : “Loss of OPTN is associated with a decrease in adhesion-mediated cell forces”

and the sentence in line 518 for : “loss of OPTN expression is associated with a decrease in total force transmitted to the environment during cell adhesion”.

Here we show, at least at two different substrate stiffness (experiments on hydrogels or pillar arrays of ~5 kPa modulus, and parallel plates assays at infinite stiffness) that lack of OPTN leads to lower forces applied by cells on their substrates, while these siOPTN cells show adhesion structures of increased size. Indeed, classically in integrin-based rigidity-sensing, the size of adhesion complexes is known to increase with traction forces, and thus with increased substrate stiffness. Our observation suggests a kind of decoupling between force generation and growth/extension of adhesion complexes in the absence of OPTN. Further experimentation that are beyond the scope of our current paper will be needed to comprehensively address the role of OPTN in this phenotype.

(4) The generation of primary HUVEC cells is a very important piece of information that is also missing. What generations were used? Often generation >P8 result in HUVEC with a pancake morphology which is reminiscent of some images seen e.g., at Figure 7 (as opposed to spindle like morphology seen in younger generations P2-P8).

HUVEC are used before P5, and SiRNA transfection was done between P2 and P3. This has now been specified in the methods section (page 23 lines 646 and 665).

(5) Why were AFM measurements performed with a tiny tip and not both a larger spherical tip that would allow to get a better idea of bulk stiffness properties. Perhaps also the large deviation of the AFM data from the mean is due to the usage of a small tip.

Zemła et al. have shown that any kind of tip geometry can measure differences in elasticity either in soft gels or in living cells (Zemła, J., Bobrowska, J., Kubiak, A., Zieliński, T., Pabijan, J., Pogoda, K., ... & Lekka, M. (2020). Indenting soft samples (hydrogels and cells) with cantilevers possessing various shapes of probing tip. *European Biophysics Journal*, 49(6), 485-495).

There were several reasons why we decided to use these AFM probes for this study:

- PFQNM-LC-A-CAL are one of the few pre-calibrated (LDV) commercial probes, meaning they provide the best calibration of the instrument to date, eliminating any bias coming from the use of several cantilevers and the variation in calibration (Schillers, H., Rianna, C., Schäpe, J., Luque, T., Dorschke, H., Wälte, M., ... & Radmacher, M. (2017). Standardized nanomechanical atomic force

microscopy procedure (SNAP) for measuring soft and biological samples. *Scientific reports*, 7(1), 1-9).

- The use of large colloidal tip has some advantages (better bulk properties evaluation with large indentations as suggested by the reviewer), but also one serious drawback in this case: the point of contact is hard to find precisely, which leads to large errors in the extraction of E for small indentations. This is especially critical for the evaluation of the cortical elasticity of cells. Furthermore, the bottom effect is more pronounced with large indenters on these rather flat and thin cells (Garcia, P. D., & Garcia, R. (2018). Determination of the elastic moduli of a single cell cultured on a rigid support by force microscopy. *Biophysical journal*, 114(12), 2923-2932).
- These probes are neither sharp, nor large spherical shape. The shape is paraboloid with a spherical diameter of 130 nm in diameter. It's a medium size with a well-defined geometry, that could also have deciphered spatial variations in elasticity.

Finally, concerning statistics, each measure used for the graph corresponds the median of several hundreds of elasticity measurements for one cell. As a consequence, the variation seen on the plot is not due the heterogeneity as probed by the AFM tip on a scan, but is a description of the cell population variation.

(6) There is missing information in the figure legends. What are the data presented? How many independent experiments were performed? What is the very basic experimental structure? Also, formatting is inconsistent in text and figures.

We have modified the legends accordingly to include all these information, and we standardized formatting

Specific comments:

(1) In figure 1A: At what density were HUVEC (by cell I assume HUVEC cell) seeded? What generation? HUVECs infections were performed at density of 250,000 cells/well on 12-well plates and before passage P5. This is indicated in the methods section

(2) In figure 1B-C: Why is standard error of the mean shown and not standard deviation? Why are not individual values shown on top of boxplots? Why are controls without error bar?

We have now shown standard deviation; individual values are represented as open dots. Untreated controls are set to 1 for internal normalization of western-blot of each of the three independent experiments, and correspond to the densitometric value of the phospho-signal band relative to the densitometric value of the loading control band (GAPDH). The goal here is to avoid variations in antibody binding efficiency, membrane blocking and washing efficiencies, decay rate of the chemiluminescent substrate used for revelation, that would prevent from cross-comparison of experiments. Therefore, the non-treated sample on each gel is used to normalize every other bands on that same western-blot to obtain fold inductions. Therefore, there is no error bar on controls, and error bars on the rest of measures correspond to variability in the fold inductions from the different independent experiments.

(3) In Figure 2D: How many independent experiments were performed?

We have now included as panel 2E a quantification of 5 independent experiments.

We have cropped the membranes shown in panel 2D to start at 1kPa for coherence with ECM stiffnesses mostly used in this study. This does not change information.

(4) In Figure 3A: Stiffness of the matrix not indicated in text/figure legend after claiming OPTN levels are very stiffness dependent. Also why is here no loading control but in other figures with 'input' there is?

Figure 3A investigates OPTN and HACE1 interaction in HUVECs cultured on plastic cultureware (a condition where endogenous OPTN is highly expressed) upon expression of tagged OPTN and HACE1 proteins. This is now indicated in the legend. GAPDH was not systematically blotted in the transfection experiments since we usually refer to the level of expression of each tagged protein. We have included the endogenous GAPDH western blot for this panel.

(5) Figure3B: Authors state 'We found that OPTN knockdown led to a reduction in Rac1Q61L

ubiquitylation under conditions of endogenous or overexpressed HACE1'. Authors probably refer to siOPTN + conditions with either Myc-HACE1 + or - .

Yes, endogenous HACE1 expression refers to Myc-HACE1 (-) while overexpressed HACE1 refers to Myc-HACE1 (+) lanes.

But there is also no big difference between siOPTN + / myc-HACE1 + and siOPTN - / myc-HACE1 - . So, if siOPTN + / myc-HACE1 + is considered a reduction in ubiquitylation then in cells without a siOPTN knockdown but endogenous levels of HACE1 the same reduction takes place? So, it is not OPTN dependent in that case?

We are not sure of the point raised by the reviewer. Our aim is to address the regulatory role of OPTN on the function of HACE1, an E3 ligase targeting active Rac1. We show here that knocking down OPTN reduces Rac1 ubiquitylation by HACE1. This holds true in conditions where cells do not overexpress HACE1 (lanes 3 vs 2) and overexpress the HACE1 E3 ligase (lanes 6 vs 5). The fact that Rac-Ub levels in lane 6 are similar to those in lane 3 can either reflect that the balance between HACE1 overexpression and OPTN knockdown is in favor of HACE1 (there is still some OPTN expressed, it is not a genetic KO). An alternative situation would be that HACE1 overexpression bypasses partially the regulation by OPTN, for example because it would be active at a particular location thanks to OPTN and exogenous expression leads to saturation of the cellular locations.

(6) Figure3C: Why did authors chose different stiffnesses than the ones tested for OPTN expression?

We apologize for this coarse mistake, the assay was indeed performed on 1-4-8 kPa. This was corrected in the figure.

(7) Figure3D: Why is here no loading control in the input? It would be nice to see the results on a stiff matrix (e.g., 25kPa) to directly see if overexpression of OPTN could abolish stiffness dependent effects.

GAPDH was not systematically blotted in the transfection experiments, since we usually refer to the level of expression of each protein and the important control here is the equal expression of HA_Rac1Q61L in the two lanes.

(8) Figure 3G: What is in fractions 1,4,5?

We now give more details about the sucrose gradient fractionation method in the text (page 11, line 306). It is difficult to answer this question directly. This assay is aimed at separating detergent-resistant membranes (and associated proteins like flotillins) from detergent soluble membranes (and associated proteins like Transferrin Receptor). Other fractions could contain membranes in single uniform phases but with properties intermediate (partial detergent insolubility) between flotillin-enriched detergent resistant and detergent soluble membranes. Characterization of these fractions, if possible, would need further work beyond the scope of our research.

(9) Figure 4A: Here sync is used as annotation in the figure, but only explained in the figure legend referring to 4C. This has now been added in the legend of figure 4A.

Also, why do authors again change the matrix stiffnesses tested (as compared to the previous figure)? In the previous (Figure 3C) they used 1,4 and 12 kPa. We apologize for the mistake in 3C, where 1-4 and 8 kPa were used. Stiffnesses used in 3A and 4C are coherent.

Authors claim that “the fraction of proliferating cells doubles between control cells on high and low stiffness matrix”. This is clearly not true, for 1 kPa is at around 27%, 8 kPa is at around 47%. The exact calculation has now been indicated in the text. “ The proportion of the control cells that entered the S and G2/M phases of the cell cycle increased by 69% on high stiffness compared to low stiffness ECM” [calculation : (46.35-27.4)/ 46.35]

Also, authors state: ' Interestingly, the OPTN knockdown cells reached maximal proliferation when plated on low elastic modulus ECM.' This is also not quite true. Their max is still at 8kPa even though the fraction of proliferating cells greatly increases compared to the control group (also if there is no significant difference for 1 and 8 kPa for the knockdown that is worth being denoted in the graph).

We meant by this sentence that proliferation of SiOPTN cells at 1 and 8 kPa is the same, and not significantly different from the SiCTRL cells plated on 8 kPa. We have now added on the graph that there is no significant difference at 1 versus 8 kPa for the OPTN knockdown cells.

(10) In Figure 4A and B and elsewhere what is “-“ kPa?

This corresponds to the synchronized conditions. For more simplicity/clarity we have just left empty space.

(11) In Figure 4C: What matrix stiffness was used here? It would have been nice to see soft/stiff conditions compared again, as that was the point in the up and down regulation of cyclin D1 dependency of OPTN.

Soft/stiff regulation of cyclin D1 dependency of OPTN is indeed shown in figure 4B. In Figure 4C we used a “full mitogen” stimulation by stiff ECM-here fibronectin coated plastic- and 20% serum in order to show that while all possible proliferative signals are provided, OPTN knockdown cells display induced cyclin D1 protein levels without impacting the rate of its transcription.

(12) In Figure 5: In the text the font size changes all the time which is not nice to read.

This has been corrected in the new figure

(13) In Figure 5D as compared to Figure 6C the stiffness of the control cells is different. Why is that? It looks like this difference is similar to the different between siCntr and siOPTN which is significant. Why SEM and not SD? What are the authors thoughts on the high variance? Does it have to do with where the cells were probed (nucleus versus cell edge). Why are the authors not providing the stiffness maps of cells since they probed stiffness by scanning? Also, what matrix stiffness was used?

Cells in 5D and 6C are in different culture conditions : 5D are cells plated for 1h30 on fibronectin-coated plates in defined medium, while 6C are cells cultured overnight in complete medium. This may be why cortical elastic moduli of control cells differ, and this makes the comparison difficult. The variance has been discussed in general comment N°5. Measurements were always performed in the similar perinuclear region of cells to avoid 2 common issues, bottom effect on the edge of the cell, and nucleus effect (nucleus is an order of magnitude stiffer than surrounding cytoplasm). This is specified in the methods section. Representative stiffness maps are now provided for these two sets of experiments in Sup. Fig.5D and Sup Fig.6D.

(14) Figure 5F: Here there is yet a new set of tested matrix stiffnesses. 0.2 kPa has not been tested before (only in one supplemental figure panel as a control) and now 4 kPa was added. Also 0.2 kPa is in essence liquid. I don't think that cells could attach there. It would be a nice proof to include images (e.g., phase contrast) of how cells look at different matrix stiffnesses reported here and also of course to validate that the stiffnesses are the ones that authors claim.

This comment has been addressed in our response to specific comment N°2. Cells do attach on 0.2 kPa fibronectin-coated hydrogels but they don't spread much (see above image). Phase contrast images of cells used for the proteomics analysis were included in Sup. Fig. 2A

(15) Figure 6: Is only one stiffness use here and which? Also in panel C why is the scanning size different than in the previous figure (40x40 versus 20x20)? In panel D why are there no error bars for the negative controls?

Cells are on plastic, this has now been indicated in the legends.

The AFM scanning size is not different between figures, it is $6 \mu\text{m}^2$ in the perinuclear region of cells in both experiments. What differs is the number of pixels in each map. This has no influence on the measurement, it only governs the map resolution (now provided as a supplemental figure) and the time of scanning. All of critical parameters were kept constant between experiments: tip shape, cantilever spring constant, tip velocity, and indentation.

Panel D : Concerning no error bars on controls, see our response to specific comment N° 2

(15) Figure 7A: What does it mean traction force curves? Is that the maximum force? Or is it the mean absolute force along the cell area? How was segmentation done to define cell borders or was that not done? Why just a single cell is shown when the average of many cells (and SD) could be shown? Circles in the legends are almost invisible.

The traction force curve represents a measure of the uniaxial force (F) exerted by a cell overtime, as measured with the parallel plate setup. We have added a scheme of the experimental setup in Fig. 7A, together with an inset representing a brightfield image of a SiCTRL and a SiOPTN cell caught between the cantilevers (Fig. 7B) to help comprehension of the technique by the reader. There is no segmentation of cells, F is measured through the deflection d of the flexible plate of known stiffness k : $F = k d$. Traction force measurements with the parallel plate setup are similar in their principle to AFM force measurements, which are based on the deflection of a cantilever of known stiffness. The representation of all curves superimposed would be unintelligible and an average curve would obliterate the characteristic shape of each traction curve due to variability. This is why we plot in panels 7B (new 7C) and 7C (new 7D) the maximal force reached for each cell and the rate-of-force increase (slope of each curve) respectively, allowing mean/median value calculations and statistical evaluations.

(16) Figure 7B-C: Where is the standard deviation. Also, how many independent experiments were done? The new plots, according to Nature Communications guidelines for this amount of measures, are box and whiskers (“For sample sizes larger than 10, please consider box-and-whisker or violin plots as alternatives. Measures of centrality, dispersion and/or error bars should be plotted and described in the figure legend”) 3 independent experiments were performed, it is now indicated in the legend.

(17) Figure 7D: It would have been highly relevant to compare how siOPTN cells act on high and low stiffness as compared to OPTN expressing cells. If OPTN indeed controls cell contractility, then by doing TFM on varying stiffness hydrogels one should see a gradual increase in forces. Also, the changes should vanish (if they are matrix stiffness dependent and OPTN indeed controls them) in siOPTN cells. Also, on the left panels authors show magnitude of traction stresses. On the right is this a phase contrast image? Why/how is the phase contrast differently adjusted for cell versus its surrounding environment?

This comment has been addressed above, see General comment N°3

On the left we plotted the magnitude of traction stresses, on the right it is a phase contrast image where we super-imposed the force vectors and the region that we used to crop the force field in order to compute the associated stresses. This is now specified in the figure legend.

(18) Figure 7E-F: Authors do not state which statistical methods they used to compare forces, energy etc. That is a general comment. Also, when force vectors are added up to compare the total forces, did they take the total size of the cell into consideration to account for morphological changes in area or not?

The statistical test used is Student’s t test, it is now specified in the legend.

The total force that is plotted is not taking into account the surface of the cell (now panel 7F). The contractile energy (now panel 7G) is the force multiplied by each displacement integrated over the whole cell surface, so it is integrating the cell surface in its definition.

Reviewer #2 (Remarks to the Author):

This work builds on previous studies indicating that activation of Rac1 by the UPEC-associated toxin CNF1 can facilitate bacterial entry into host cells. Host cells can counter aberrant Rac1 activation by CNF1 through HACE1-mediated ubiquitination and subsequent proteosomal degradation of activated Rac1. Petracchini et al. dig deeper into this system, showing that CNF1 activities augment FimH-mediated invasion of host cells by UPEC via integrin receptors while also providing new insight into the regulatory control of Rac1 and cytoskeletal tension/dynamics. More specifically, they show that OPTN can regulate HACE1 activity, thereby impacting the levels of active Rac1, in response to external ECM stiffness. The paper is dense with information and utilizes a great set of diverse experimental approaches with well-written text that should help make the work accessible to a wide range of individuals with varied backgrounds. Many pathogens, in addition to UPEC, hijack integrin receptors.

This, and clear links to cell cycle progression and a number of other key cellular pathways, will make this study of great interest to a broad swath of scientists. There are a few mostly minor issues that should be clarified, but overall enthusiasm for this well-thought-out paper is high.

Bigger Issues:

- This study uses the HUVEC cell line. Some explanation for this choice should be provided, as most studies that examine UPEC invasion employ bladder epithelial cells. The specific cell line used should be made clear at the start of the Results section. It would be useful to the UTI field to know if some of the key findings were also true in bladder cells. For example, would silencing OPTN expression also inhibit UPEC entry into bladder epithelial cells?

HUVEC are primary cells (human) and were chosen for our work to ensure at best that Rho GTPases ubiquitylation regulation was normally conserved, since this pathway is often deregulated in cell lines as studied in ref : doi/10.1091/mbc.E05-09-0876. Source for primary human cells amenable to plasmid and SiRNA transfection and biochemistry-compatible high culture yields are scarce, HUVEC offer these advantages. Moreover, uropathogenic E. coli frequently gain access to the bloodstream, and some isolates can cause neonatal meningitis, making endothelial cells a pathophysiological relevant cell type.

To address reviewer-2 concern, we have performed OPTN silencing in HTB-9 bladder epithelial cells and we could confirm that the function of OPTN in bacterial invasion is conserved. Also, similarly to our findings in HUVECs, OPTN silencing does not impact bacteria attachment to cells. This is now included in Sup. Fig.6B

- Do any of the inhibitors used in the paper (e.g. CK-666 or blebbistatin) affect growth or viability of UTI89 at the concentrations used?

The pharmacological inhibitors used had no effect on the bacterial growth. This is now showed in Sup. Fig.11

- CNF1 is used at a concentration of 1 nM. Can the authors comment on the physiological relevance of this concentration? Is this level seen during actual infections?

This concentration corresponds to the maximal activity of the toxin, as measured in the cytotoxicity assay on Hep-2 cells (ref 99, DOI : 10.1016/S0076-6879(06)06033-2) : 100% multinucleated cells in 48h. It is one order of magnitude above the amount produced by UTI89 in our conditions of infection (MOI:100).

We have now included a thorough description of why we use this concentration in the methods section, and SupFig1J.

- The study uses MOIs of 100, which is substantially higher than is typically used in the field. Why were so many bacteria used? Are there dosage effects with the HUVECs? This is not necessarily a major issue, but it would be good to understand why the high MOI was chosen.

A MOI of 100 is typical of *in vitro* studies addressing very short experimental windows, as it is in our case with 30 min adhesion of bacteria and 30 min antibiotic treatment to recover only intracellular population.

As an example, in a recent paper addressing host restriction pathways during infection of HTB-9 bladder epithelial cells with the UPEC strain CFT073, authors use MOI of 100 in 30 min infections or MOI of 50 in 2 hours infections (doi : 10.1038/s41467-021-22726-8).

Moreover, inoculation of bladder to induce UTI in mice models is classically done using 1-2 x 10⁸ CFU/mouse (doi:10.1038/nprot.2009.116) and the amount of urothelial cells in a mouse bladder has been estimated to 1-5 x 10⁸ cells (doi:10.1007/BF02890136). Therefore, we think that MOI 100 should not be necessarily regarded as overwhelming.

Minor issues:

- The Summary could do a better job of highlighting the key major findings of the paper more directly. Lines 57-59 seem somewhat redundant with the final sentence of the Summary.

We have modified the summary

- Line 75 – it is not entirely clear how 'critical' FimH is for UPEC persistence within the gut, as at least one paper noted that FimH is dispensable in some situations (doi: 10.1128/IAI.00746-17)

We have now replaced “essential” by “important” in our sentence to tone down slightly the requirement of FimH:

Expression of chaperone-usher pathway (CUP) type I pili tipped with the adhesin FimH is important for colonization, invasion and persistence of UPEC in the mouse bladder and to form persistent reservoirs in the gastrointestinal tract

- Line 102 and line 387 should have references.

References were added in the manuscript

- Line 212 – use of 4 kPa as an “intermediate” stiffness value seems strange in considering the lower (1 kPa) and higher “50 kPa) values used in the paper. Why not use 25 kPa, or was 4 kPa used in anticipation of non-linear effects?

By “intermediate” we did not mean mathematical half but rather an ECM elastic modulus that is not characterized as “soft” but above, and would have a biological relevance in epithelial cells proliferation control, as reported in the literature. The 1 to 4 kPa difference is the order of difference seen in cancer-associated ECM stiffening, as characterized in breast cancer in the reference : Paszek, M. J. *et al.* Tensional homeostasis and the malignant phenotype. *Cancer Cell* **8**, 241–254 (2005).

We have eliminated the adjective “intermediate” to avoid confusion.

- The font size used for Fig 2C makes the text very hard to read, even with zooming. The CNF1 nodes (yellow with orange edges) could be made more visible to help readers.

The visualization of this panel has been improved.

Reviewer #3 (Remarks to the Author):

The manuscript by Petracchini and colleagues explores the role of ECM stiffness during bacterial invasion. The authors identify a protein, OPTN, which is upregulated by stiffness. OPTN has been previously shown to interact with the Ubiquitin ligase HACE1, which regulates the ubiquitylation and degradation of active Rac1. Silencing OPTN promotes Rac1 activation and cells that show more and larger focal adhesions, but surprisingly a decreased traction force.

The manuscript tries to connect the bacterial toxin CNF1 with Rac1 activation, enhanced stiffness with increased invasion, and OPTN mediated Rac1 inactivation to mechanotransduction. The individual observations presented in this manuscript are interesting and suggest a concerted mechanism connecting CNF1 to mechanotransduction to regulation of Rac1 activity. However, in my opinion, these connections are not properly validated.

The individual claims are well supported by the results (e.g. OPTN expression is regulated by cellular stiffness). However, the connections between the independent results are not clearly supported. For example, a causal link between Rac1 activation and the phenotypes observed is not demonstrated in this article. It is not even tested that some of the phenotypes observed, such as increased FA number, or decreased traction force, are mediated by the increase in Rac1 activation observed in the absence of OPTN.

Most of the phenotypes observed suggest an involvement of other Rho proteins, especially RhoA. Changes in RhoA activity have not been tested in this article, nor RhoA was ever mentioned in the discussion, which I believe are important omissions, especially when CNF1 can also target RhoA. In my opinion, the key element missing here is to demonstrate that the phenotype changes observed when OPTN is depleted are Rac1 mediated.

This lack of conclusive characterization of the causative roles of each of the key players characterized here make the conclusion read as overinterpreted, missing the caveats or alternative explanations possible.

Major Comments:

- With some exceptions noted below the experiments are sound and well designed.
- The focal adhesions phenotype suggests involvement of RhoA rather than Rac1. This is also true for some of the mechanical response to stiffness observed. Have the authors rule out the involvement of RhoA? This is important, especially since RhoA can also be activated by CNF1.

Specific Comments

- In p6 line 161-162, the activation of Rac1 by CNF1 in different stiffness levels should be shown, at least in the Supp Figures.

We have now added this measure in Supplemental Fig. 1B

Similarly, the OTN mRNA in different stiffnesses (Lines 265-266) should also be shown
 We have now added this experiment in Supplemental Fig. 2E

-In figure 1B-C the increase in the levels of p-PAX, pSrc and p-FAK (to a lesser extent), correlate with the increase in the levels of the corresponding total proteins. It is hard to explain the substantial increase in total levels of these proteins after only 30-60 min of treatment, especially since GAPDH levels do not change the same way. In my opinion, and based on these observations, normalizing to the levels of GAPDH is not the best option, and the results should be shown as a ratio of phospho/total levels. I wonder, based on the shape of the WB lanes (the pattern is identical), if some of these phospho and total blots were reblotted (compare total PAX to p-PAX or total Src vs p-Src).

-Related to the previous point, the statement in lines 179-180 that the signal reaches max levels in 30 min may also be affected by the total levels, which are drastically lower at 60 min (same argument as above applies)

The blots we were presenting in the original figure were indeed blotted first against the phospho-proteins (Top of membrane) and the loading control GAPDH (Bottom of membrane). Top membranes were then stripped and re-probed against each of the total proteins. We agree with reviewer 3 that this generated artefactual variations in the level of total proteins, because stripping did not erase totally the previous antibody. This indeed motivated our choice to quantify phospho-signals relative to GAPDH. We have re-run blots of the experiments corresponding to the ones shown/used in Figure 1B and 1C to generate new panel Supplemental 1C and verify our quantifications. Our quantifications are correct by virtue of total protein levels being invariable, as shown in the graph below representing total FAK, Src, p130CAS and PAX expression levels relative to GAPDH densitometric measures on each of the migrations.

We added the sentence in the figure legend:

“Expression of FAK, PAX, Src and p130CAS are not modified in our experimental conditions.”

-Representative WB for S1D should be shown (again related to my comments above)

Representative western blots are now provided in the Supplemental 1F panel together with the quantifications.

-In p11 line 310, the authors state that “OPTN positively controls HACE1 activity towards Rac1”. This has actually not been directly measured and the increase of ubiquitylated Rac1 may just reflect an increase in binding as shown, but not an increase in activity.

Reviewer 3 is absolutely correct. The sentence has now been changed to:

“In conclusion, OPTN positively controls the ubiquitylation of Rac1 by HACE1 E3 ubiquitin ligase, thereby limiting Rac1 activity in cells.”

-The focal adhesion phenotype observed in Figure 5 is more representative of active RhoA than Rac1 (which when hyperactivated stimulates the formation of focal contacts rather than mature adhesions), and despite correlation between siOPTN and Rac1 activity, these results do not actually connect the activation of Rac1 with the phenotype observed. Since CNF1 can also activate RhoA, it is important to rule out that this effect is not mediated by RhoA.

To address this major criticism, we have included measures of RhoA-ROCK activity, assessed by p-MLC monitoring, in response to fibronectin attachment, and to CNF1 stimulation. We could see that cells knocked down for OPTN do not present an over-activation of the RhoA pathway (Supplemental Fig. 3C

and Supplemental Fig. 6C), which could have explained the focal adhesions phenotype. If any difference, a tendency was towards less p-MLC in OPTN-KD cells, although non-significant. We have also tested the effect of Rac inhibition on cell spreading, and could measure that the OPTN-KD cells were relying on the Rac pathway to a higher extent than control cells (Supplemental Fig. 5E, 69% inhibition of spreading vs 38% respectively).

-The decrease in traction force observed in siOPTN is intriguing. Analyzing stress fibers and pMLC could shed some light on the mechanism by which this is occurring. Also, looking at focal adhesions turnover would be informative too, as turnover defects have been shown to promote phenotypes similar to the one observed in this manuscript.

We agree with Rev3 on the compelling nature of this phenotype. A deep and comprehensive understanding of OPTN role in the connection between adhesion structures and force buildup would need the setup of specialized techniques and further extensive researches that are beyond the scope of this study. They would per se uphold another publication.

We have added the following sentence in the discussion section : “Alternative mechanisms like a reduced turnover of FAs in OPTN knockdown cells could also contribute to their increase in size. Further researches are needed to investigate such hypothesis and the underlying molecular pathways”

Minor Comments:

-Fig 1D and 1E show the same type of measurement. However, the scales are very different in magnitude. Are the results in 1D normalized?

Yes, results in 1D represent CFU/mL for each SiRNA transfection normalized to CFU/mL obtained in the SiCtrl condition. To avoid confusion, the y axis legend was changed from “ Internalized bact (Norm. CFU/mL) ” in the initial graph to “Internalized bact (fold change to SiCtrl; Norm. CFU/mL)

-Low quality on several blots (S1E, GAPDH for example among others)

We provided better western blots for S1E (now S1G) and 1B total proteins (new S1C); and included several new western blots that we wish to be of good quality for figure panels S1F, Fig 3A GAPDH, S3C, Fig 4B, S6A, S6B, S6C.

-In S3A it looks like expressing HACE1 and RacQ61 may have a negative effect on OPTN ubiquitylation. I disagree with the text conclusion that there is no change.

Reviewer 3 is correct. We intended in this approach to determine if OPTN was a target of the HACE1 E3 ligase activity. We have replaced “not modified” by “not induced”; the sentence is as follows:

“Although OPTN presented a ubiquitylation profile, the latter was not induced by the presence of HACE1 or the Rac1Q61L active mutant (Sup Fig. 3A).”

REVIEWERS' COMMENTS

Reviewer #1 (Remarks to the Author):

In the present revised version of this manuscript, all my previous concerns and comments have been addressed by the authors. Therefore, I believe that the current version of the manuscript is appropriate for publication and I have no further comments to add.

Reviewer #2 (Remarks to the Author):

The authors nicely addressed my concerns. I believe this is an interesting and impactful study, and that it is ready for publication. Great job by the authors!